# Enabling Pareto-Stationarity Exploration in Multi-Objective Reinforcement Learning: A Weighted-Chebyshev Multi-Objective Actor-Critic Approach

## Abstract

In many multi-objective reinforcement learning (MORL) applications, being able to systematically explore the Pareto-stationary solutions under multiple non-convex reward objectives with theoretical finite-time sample complexity guarantee is an important and yet under-explored problem. This motivates us to take the first step and fill the important gap in MORL. Specifically, in this paper, we propose a weighted-Chebyshev multi-objective actor-critic (WC-MOAC) algorithm for MORL, which uses multi-temporal-difference (TD) learning in the critic step and judiciously integrates the weighted-Chebychev (WC) and multi-gradient descent techniques in the actor step to enable systematic Pareto-stationarity exploration with finite-time sample complexity guarantee. Our proposed WC-MOAC algorithm achieves a sample complexity of $\tilde{\mathcal{O}}(\epsilon^{-2}p_{\min}^{-2})$ in finding an $\epsilon$-Pareto-stationary solution, where $p_{\min}$ denotes the minimum entry of a given weight vector $p$ in the WC-scarlarization. This result not only implies a state-of-the-art sample complexity that is independent of objective number $M$, but also brand-new dependence result in terms of the preference vector $p$. Furthermore, simulation studies on a large KuaiRand offline dataset, show that the performance of our WC-MOAC algorithm significantly outperforms other baseline MORL approaches.

## 1 Introduction

**1) Motivation:** As a foundational machine learning paradigm for sequential decision-making, reinforcement learning (RL) has found an enormous success in many applications (e.g., healthcare (Petersen et al., 2019; Raghu et al., 2017b), financial recommendation (Theocharous et al., 2015), ranking system (Wen et al., 2023), resources management (Mao et al., 2016) robotics (Levine et al., 2016; Raghu et al., 2017a), and recently in generative AI (Franceschelli & Musolesi, 2024)). Also, as more complex applications emerge, RL has increasingly evolved from single-objective to multi-objective settings. For instance, in RL-driven short video streaming platforms (Cai et al., 2023), the system sequentially displays short videos to optimize multiple rewards at the same time, including but not limited to "WatchTime", "Subscribe", "Like", "Dislike", "Comment", etc. As another example, to attract diverse customers and maximize long-term total benefits, an e-commerce recommender system sequentially ranks and displays products by balancing the conflicting preferences of different user groups (e.g., some prefer low prices and can tolerate slow delivery, while others prefer quick delivery over low prices). All of these applications entail the need for *multi-objective reinforcement learning* (MORL) (Stamenkovic et al., 2022; Ge et al., 2022; Chen et al., 2021a).

Mathematically, an $M$-objective MORL problem can be formulated as finding an optimal policy $\pi_{\boldsymbol{\theta}}$, which is parameterized by $\boldsymbol{\theta}$, to maximize multi-dimensional long-term accumulative rewards, i.e.,

$$\max_{\boldsymbol{\theta} \in \mathbb{R}^{d_1}} \mathbf{J}(\boldsymbol{\theta}) := \left[ J^1(\boldsymbol{\theta}), J^2(\boldsymbol{\theta}), \ldots, J^M(\boldsymbol{\theta}) \right]^\top, \tag{1}$$

where $J^i(\boldsymbol{\theta})$ is the expected accumulative reward for the $i$-th objective under policy $\pi_{\boldsymbol{\theta}}$, $i \in [M]$[1]. For the MORL problem in (1), since it is often infeasible to find a common policy parameter $\boldsymbol{\theta}$

---

[1]In this paper, we use shorthand notation $[M]$ to denote the set $\{1, \cdots, M\}$.

that can simultaneously maximize all objectives in (1), a more appropriate goal in MORL is to find a Pareto-optimal solution for all objectives (i.e., no objective can be further improved unilaterally without decaying any other objective). However, due to the fact that Pareto-optimal solutions are not unique in general, it is important to be able to *systematically* and *efficiently* explore the set of all Pareto-optimal solutions (also known as the Pareto front), based on which one can then pick the most desirable Pareto-optimal solutions. Unfortunately, due to the NP-hardness resulting from non-convex objectives in most MORL problems (Danilova et al., 2022; Yang et al., 2024), finding Pareto-optimal solutions is intractable in general and even developing algorithms that converge to a weaker *Pareto-stationary solution* (a necessary condition for being Pareto-optimal, more on this later) with *low sample complexity* is already highly non-trivial and remains under-explored in this literature thus far. This motivates us to take the first step and fill this important gap in the MORL literature.

In light of the fact that MORL is a special class of multi-objective optimization (MOO) problems, in this paper, we propose a weighted-Chebyshev multi-objective actor-critic (WC-MOAC) method by drawing inspirations and insights from the MOO literature. More specifically, to enable systematic Pareto-front exploration with low sample complexity in MORL, our proposed WC-MOAC method uses temporal-difference (TD) learning in the critic component and judiciously integrates the weighted-Chebyshev (WC) and multi-gradient-descent algorithmic (MGDA) techniques in the actor component. The rationale behind our approach is three-fold: (i) Combining the strengths of value-based and policy-based RL approaches, the actor-critic framework has been shown to offer state-of-the-art performance in RL; (ii) in the MOO literature, it has been shown that an optimal solution under the WC-based scalarization approach (also known as hypervolume scalarization) provably achieves the Pareto front even when the Pareto front is non-convex (Zhang & Golovin, 2020); and (iii) for MOO problems, the MGDA method is an efficient approach for finding a Pareto-stationary solution (Désidéri, 2012). [2] Finally, the connection between gradient information in optimization and TD-error in RL leads us to generalize the WC and MGDA approaches from MOO to our WC-MOAC method for MORL.

**2) Challenges:** However, to show that WC-MOAC enjoys systematic Pareto-stationarity exploration with provable low finite-time sample complexity remains highly non-trivial due to multiple challenges:

1) In the MOO literature, WC- and MGDA-based techniques are developed with very different goals in mind: facilitating Pareto-front exploration and achieving Pareto-stationarity, respectively. To date, it remains unclear how to combine them to achieve systematic Pareto-stationarity exploration with low finite-time sample complexity simultaneously even for general MOO problems, not to mention generalizing them to the more specially structured MORL problems and the associated theoretical performance analysis. Indeed, to our knowledge, there is no such result in the literature on integrating WC- and MGDA- techniques for designing MORL policies.

2) In WC-MOAC, the critic and actor components evaluate and improve the policies, respectively, with an intricate dependence between these two components. Such a complex dependence between actor and critic further renders standard convergence analysis in MOO irrelevant to our proposed WC-MOAC methods. Thus, it remains an open question whether one can design a multi-objective actor-critic algorithm to facilitate Pareto-stationarity exploration with a provable finite-time sample complexity guarantee.

3) In WC-MOAC, both critic and actor components update their parameters through stochastic TD-errors based on directions guided by a WC-scalarization weight vector and finite-length state-action trajectories. All of these inject cumulative biases in policy parameter updates. If not handled properly, such biases could significantly affect the performance of our WC-MOAC method for MORL or could even lead to a divergence of policy parameter updates.

**3) Key Contributions:** In this paper, we overcome the aforementioned challenges and propose a weighted-Chebyshev multi-objective actor-critic algorithmic framework with provable finite-time Pareto-stationary convergence and sample complexity guarantees. Collectively, our results provide the first building block toward a theoretical foundation for MORL. Our main contributions are summarized as follows:

- We propose a weighted-Chebyshev multi-objective actor-critic algorithmic framework (WC-MOAC) based on MGDA-style policy-gradient update for both (heterogeneous) discounted

---

[2]MGDA can be viewed as an extension of the standard gradient descent method to MOO, which dynamically performs a linear combination of all objectives' gradients in each iteration to identify a common descent direction for all objectives. Also, the finite-time convergence rate of MGDA has recently been established under different MOO settings, including convex and non-convex objective functions (Liu & Vicente, 2021; Fernando et al., 2022) and decentralized data (Yang et al., 2024), etc.

and average reward settings in MORL. Our WC-MOAC policy framework offers finite-time convergence and sample complexity of $\tilde{\mathcal{O}}(\epsilon^{-2}p_{\min}^{-2})$ for achieving an $\epsilon$-Pareto stationary solution, where $p_{\min}$ denotes the minimum entry of a given weight vector $\mathbf{p}$ in the WC scalarization. To our knowledge, no such finite-time convergence and sample complexity results with respect to the WC-scalarization parameter exist in the MORL literature.

- To mitigate the cumulative systematic bias injected from the WC-scalarization weight direction and finite-length state-action trajectories, we propose a momentum-based mechanism in WC-MOAC. Somewhat surprisingly, we show that this momentum approach in WC-MOAC enjoys a convergence rate and sample complexity that are *independent* of the number of objectives. This is fundamentally different from general MOO, where the scaling laws of the convergence results could be linear (Fernando et al., 2022) or even cubic (Zhou et al., 2022) with respect to $M$.

- We show that, with the proposed momentum mechanism and an appropriate schedule of the momentum coefficient, WC-MOAC can automate the initialization of the weights of individual policy gradients from data samples in the environment, which avoids cumbersome manual initialization. This significantly improves the practicality and robustness of the algorithm.

- We conduct empirical studies on a large-scale KauiRand offline dataset, to show our WC-MOAC algorithm significantly outperforms other baseline MORL approaches that adopt linear scalarization and other heuristic ideas.

## 2 RELATED WORK

In this section, we provide an overview on three closely related areas, namely multi-objective optimization, multi-objective reinforcement learning, and RL problems with multiple rewards, thereby putting our work in comparative perspectives.

**1) Multi-Objective Optimization (MOO):** Generally speaking, MOO approaches can be broadly classified into four main categories (Miettinen, 1999): 1) no-preference methods, 2) a priori methods, 3) a posteriori methods, and 4) interactive methods. While the latter three categories all involve preference weight information from a decision maker either directly or indirectly, the first category does not require any preference information. A line of work (Fliege et al., 2019; Liu & Vicente, 2021; Zhou et al., 2022; Sener & Koltun, 2018; Yang et al., 2024; Fernando et al., 2022; Xiao et al., 2023) has utilized the MGDA (Désidéri, 2012) technique to characterize the finite-time convergence/sample complexity of MOO problems, including one recent work on no-preference MORL (Zhou et al., 2024). However, in this paper, we are concerned with the finite-time convergence and effectiveness in practical MORL setting that comes with given preference weight information and further enabling Pareto-stationarity exploration. A closely related work in MOO can be found in (Momma et al., 2022), where the authors studied MOO problem with pre-defined preference weight incorporated by proposing a WC-based MGDA approach to align the Pareto solution with the preference direction. However, this work only showed the empirical effectiveness and did not provide finite-time convergence results. Another closely related work in (Xiao et al., 2024) proposed a direction-oriented MOO algorithm based on a weighted sum of the MGDA and the linear scalarization approaches. This is in stark contrast to the WC-scalarization technique in our approach. Extensive empirical comparisons are provided in Section 5 to show the superiority of our WC-MOAC method over the RL counterpart of (Xiao et al., 2024).

**2) Multi-Objective Reinforcement Learning (MORL):** MORL is a type of sequential decision-making problems endowed with multiple rewards. Different from conventional RL problems with scalar-valued rewards (e.g., Sutton & Barto (2018); Konda & Tsitsiklis (1999); Xu et al. (2020); Guo et al. (2021)), MORL is concerned with optimizing vector-valued rewards, either directly or through various types of scalarization. Although the studies on MORL are not new (see, e.g., Gábor et al. (1998); Parisi et al. (2016); Van Moffaert & Nowé (2014); Abels et al. (2019); Yang et al. (2019); Abdolmaleki et al. (2020); Reymond et al. (2023); Roijers et al. (2013); Ru$a$dulescu et al. (2020); Hayes et al. (2022)), finite-time convergence results for multi-objective actor-critic (MOAC) algorithms remain quite limited. To our knowledge, the first MOAC algorithm was proposed in (Chen et al., 2021a), which is based on deterministic policy gradients. Subsequently, a two-stage constrained actor-critic algorithm was proposed in (Cai et al., 2023), where the MORL formulation is different from ours and takes an $\epsilon$-constrained scalarization approach (i.e., all except one objective are reformulated as $\epsilon$-constraints and the only remaining objective is set as the system objective). Also, *none* of the above MORL works offers finite-time convergence rate or sample complexity results.

**3) RL Problems with Multi-Reward Scalarization:** We note that several RL paradigms bear some similarities with MORL in the sense of having multiple rewards. The first such RL paradigm is cooperative multi-agent reinforcement learning (MARL) (Zhang et al., 2018; Chen et al., 2021b; Hairi et al., 2022), where each agent has a scalar-valued reward. However, the global objective of cooperative MARL is a static weighted sum of all agents' rewards. Similarly, many MORL problems are often scalarized to enable the use of single-objective RL techniques (e.g., linear scalarization in (Stamenkovic et al., 2022)). Another multi-reward RL paradigm is the constrained (also known as safe) RL Cai et al. (2023), which balances multiple RL objectives with a set of predefined parameters associated with the constraints to indicate the constraint levels. Due to different problem structures, these multi-reward RL problems are often concerned with other goals rather than Pareto-stationarity.

## 3 MORL PROBLEM FORMULATION

In this section, we first introduce the preliminaries and problem formulation of MORL problems.

**1) Multi-Objective Markov Decision Process:** Similar to its single-objective counterpart, an MORL problem can be formulated as a multi-objective Markov decision process (MOMDP), which is characterized by a quadruple $(\mathcal{S}, \mathcal{A}, P, \mathbf{r})$, where $\mathcal{S}$ and $\mathcal{A}$ denote the state and action space of the agent, respectively. For any given $(s, a) \in (\mathcal{S}, \mathcal{A})$, $P(\cdot|s, a) : \mathcal{S} \times \mathcal{A} \times \mathcal{S} \mapsto [0, 1]$ is the transition kernel that maps a probability measure on $\mathcal{S}$, and $\mathbf{r}(s, a) \in \mathbb{R}^M$ denotes an $M$-dimensional vector-valued reward function. In this paper, we assume $\mathcal{S}$ and $\mathcal{A}$ to be finite. The instantaneous reward $r^i(s, a)$ for each objective $i \in [M]$ is deterministic given state $s$ and action $a$.[3] In MOMDP, consider a $\boldsymbol{\theta}$-parameterized stationary policy defined as $\pi_{\boldsymbol{\theta}} : \mathcal{S} \times \mathcal{A} \mapsto [0, 1]$, with $\pi_{\boldsymbol{\theta}}(a_t|s_t)$ denotes the probability of taking action $a_t \in \mathcal{A}$ in state $s_t \in \mathcal{S}$ in time $t$. Next, we introduce the following standard assumptions on $\pi_{\boldsymbol{\theta}}(a|s)$, which imposes smoothness and guarantees, for the underlying Markov process, the existence of a unique steady state distribution for any given stationary policy, and boundedness on rewards.

**Assumption 1** (MOMDP). For any state $s \in \mathcal{S}$, action $a \in \mathcal{A}$, policy parameter $\boldsymbol{\theta} \in \mathbb{R}^{d_1}$, the given MOMDP satisfies the following:

(a) The policy function $\pi_{\boldsymbol{\theta}}(a|s) \geq 0$ is continuously differentiable with respect to the parameter $\boldsymbol{\theta}$;

(b) The Markov chain $\{s_t\}_{t \geq 0}$ induced by the policy $\pi_{\boldsymbol{\theta}}$ is irreducible and aperiodic, with the transition matrix $P_{\boldsymbol{\theta}}(s'|s) = \sum_{a \in \mathcal{A}} \pi_{\boldsymbol{\theta}}(a|s) \cdot P(s'|s, a), \forall s, s' \in \mathcal{S}$;

(c) Each instantaneous reward $r_t^i$ is non-negative and uniformly bounded by a constant $r_{\max} > 0$.

Assumption 1 (a) allows the smoothness of the parameterized policy $\pi_{\boldsymbol{\theta}}$, which can be easily satisfied with policies like soft-max; (b) guarantees that there exists a unique stationary distribution $d_{\boldsymbol{\theta}}(\cdot)$ over $s \in \mathcal{S}$ for the Markov chain induced by any stationary policy $\pi_{\boldsymbol{\theta}}$; Also, (c) is common in the literature (e.g., Zhang et al. (2018); Xu et al. (2020); Doan et al. (2019)) and easy to be satisfied in many practical MOMDP models with finite state and action spaces.

**2) Learning Goal and Optimality in MORL:** We define the reward objective function $J^i(\boldsymbol{\theta})$ for the $i$-th objective to be the expected accumulative reward under policy $\pi_{\boldsymbol{\theta}}$ over all possible initial states and trajectories. In this paper, we consider both accumulated discounted and average rewards in the infinite time horizon setting defined as follows:

*2-1) Discounted Reward:* For each objective $i \in [M]$, the reward objective function under the discounted reward setting is defined as $J^i(\boldsymbol{\theta}) := \mathbb{E}[\sum_{t=1}^{\infty} (\gamma^i)^t r_t^i(s_t, a_t)]$, where $\gamma^i \in (0, 1)$ is the discount factor associated with objective $i$.

*2-2) Average Reward:* For each objective $i \in [M]$, the reward objective function under the average reward setting is defined as: $J^i(\boldsymbol{\theta}) := \lim_{T \to \infty} \mathbb{E}[\frac{1}{T} \sum_{t=1}^{T} r_t^i(s_t, a_t)]$.

The goal of MORL is to find an optimal policy $\pi_{\boldsymbol{\theta}*}$ with parameters $\boldsymbol{\theta}^*$ to jointly maximize all the objective's long-term rewards in the sense of Pareto-optimality (to be defined next). Specifically, we want to learn a policy $\pi_{\boldsymbol{\theta}}$ that maximizes the following vector-valued objective:

$$\max_{\boldsymbol{\theta} \in \mathbb{R}^{d_1}} \mathbf{J}(\boldsymbol{\theta}) := [J^1(\boldsymbol{\theta}), \ldots, J^M(\boldsymbol{\theta})]^\top.$$

---

[3] For ease of exposition in this paper, we consider the instantaneous rewards as deterministic given state-action pair. However, the results holds similarly for stochastic instantaneous rewards as well.

As mentioned in Section 1, due to the fact that the objectives in MORL are conflicting in general, the more appropriate and relevant learning goal and optimality notions in MORL are the Pareto-optimality and the Pareto front, which are defined as follows:

**Definition 1** ((Weak) Pareto-Optimal Policy and (Weak) Pareto Front). We say that a policy $\pi_{\boldsymbol{\theta}}$ dominates another policy $\pi_{\boldsymbol{\theta'}}$ if and only if $J^i(\boldsymbol{\theta}) \geq J^i(\boldsymbol{\theta'}), \forall i \in [M]$ and $J^i(\boldsymbol{\theta}) > J^i(\boldsymbol{\theta'}), \exists i \in [M]$. A policy $\pi_{\boldsymbol{\theta}}$ is Pareto-optimal if it is not dominated by any other policy. A policy $\pi_{\boldsymbol{\theta}}$ is weak Pareto-optimal if and only if there does not exist a policy $\pi_{\boldsymbol{\theta'}}$ such that $J^i(\boldsymbol{\theta'}) > J^i(\boldsymbol{\theta}), \forall i \in [M]$. Moreover, the image of all (weak) Pareto-optimal policies constitute the (weak) Pareto front.

In plain language, a Pareto-optimal policy identifies an equilibrium where no reward objective can be further increased without reducing another reward objective, while a weak Pareto-optimal policy characterizes a situation where no policy can simultaneously improve the values of all reward objectives (i.e., ties are allowed). However, since MORL problems are often non-convex in practice (e.g., using neural networks for policy modeling or evaluation), finding a weak Pareto-optimal policy is NP-hard. As a result, finding an even weaker Pareto-stationary policy is often pursued in practice. Formally, let $\nabla_{\boldsymbol{\theta}} J^i(\boldsymbol{\theta})$ represent the policy gradient (to be defined later) direction of the $i$-th objective with respect to $\boldsymbol{\theta}$. A Pareto-stationary policy is defined as follows:

**Definition 2** (Pareto-Stationary Policy). A policy $\pi_{\boldsymbol{\theta}}$ is said to be Pareto-stationary if there exists no common ascent direction $\mathbf{d} \in \mathbb{R}^{d_2}$ such that $\mathbf{d}^{\top} \nabla_{\boldsymbol{\theta}} J^i(\boldsymbol{\theta}) > 0$ for all $i \in [M]$.

Since MORL is a special-structured MOO problem, it follows from the MOO literature that Pareto stationarity is a necessary condition for a policy to be Pareto-optimal(Désidéri, 2012). Note that in convex MORL settings where all objective functions are convex functions, Pareto-stationary solutions imply Pareto-optimal solutions.

## 4 WC-MOAC: Algorithm Design and Theoretical Results

In this section, we will propose our WC-MOAC algorithmic framework for solving MORL problems. As mentioned in Section 1, our WC-MOAC algorithm is motivated by two key observations: (i) actor-critic approaches combine the strengths of both value-based and policy-based approaches to offer the state-of-the-art RL performances; and (ii) an optimal solution under the WC-based scalarization provably achieves the Pareto front even for non-convex MOO problems. In what follows, we will first introduce some preliminaries of WC-MOAC in Section 4.1, which are needed to present our WC-MOAC algorithmic design in Section 4.2. Lastly, we will present the finite-time Pareto-stationary convergence and sample complexity results of WC-MOAC in Section 4.3.

### 4.1 Preliminaries for the Proposed WC-MOAC Algorithm

Similar to conventional single-objective actor-critic methods, the critic component in WC-MOAC evaluates the current policy by applying TD learning for all objectives. However, the novelty of WC-MOAC stems from the actor component, which applies policy-gradient updates by judiciously combining 1) WC-scalarization and 2) MGDA-style updates motivated from the MOO literature.

**1) Weighted-Cheybshev Scalarization:** The WC-scarlization is a scarlization method in MOO that converts a vector-valued MOO problem into a scalar-valued optimization problem, which is more amenable for algorithm design. Specifically, let $\Delta_M$ represent the $M$-dimensional probability simplex. For a multi-objective loss minimizaiton problem $\min_{\mathbf{x}} \mathbf{F}(\mathbf{x}) := [f_1(\mathbf{x}), \ldots, f_M(\mathbf{x})]^{\top} \in \mathbb{R}_+^M$, the WC-scalarization with a weight vector $\mathbf{p} \in \Delta_M$ is defined in the following min-max form:

$$\mathsf{WC}_{\mathbf{p}}(\mathbf{F}(\mathbf{x})) := \min_{\mathbf{x}} \max_i \{p_i f_i(\mathbf{x})\}_{i=1}^M = \min_{\mathbf{x}} \| \mathbf{p} \odot \mathbf{F}(\mathbf{x}) \|_{\infty}, \tag{2}$$

where $\odot$ denotes the Hadamard product. The use of WC-scalarization in our WC-MOAC algorithmic design is inspired by the following fact in MOO (Golovin & Zhang, 2020; Qiu et al., 2024):

**Lemma 1.** A solution $\mathbf{x}^*$ is weakly Pareto-optimal to the problem $\min_{\mathbf{x}} \mathbf{F}(\mathbf{x})$ if and only if $\mathbf{x}^* \in \arg\min_{\mathbf{x}} \mathsf{WC}_{\mathbf{p}}(\mathbf{F}(\mathbf{x}))$ for some $\mathbf{p} \in \Delta_M$.

Lemma 1 suggests that, by adopting WC-scalarization in MORL algorithm design (since MORL is a special class of MOO problems), we can systematically obtain all weakly Pareto-optimal policies

(i.e., exploring the weak Pareto front) by enumerating the WC-scalarization weight vector $\mathbf{p}$ if the WC-scalarization problem can be solved optimally. As will be seen later, this motivates our WC-MOAC design in Section 4.2.

**2) Policy Gradient for MORL:** Since the actor component in our WC-MOAC algorithm is a policy-gradient approach, it is necessary to formally define policy gradients for MORL. Toward this end, we first define the advantage function for each reward objective $i \in [M]$: $\mathrm{Adv}_{\boldsymbol{\theta}}^i(s, a) = Q_{\boldsymbol{\theta}}^i(s, a) - V_{\boldsymbol{\theta}}^i(s)$, where $Q^i(s, a)$ and $V^i(s)$ are the Q-function and value function for the $i$-th objective (cf. the Appendix for detailed definitions). Let $\boldsymbol{\psi}_{\boldsymbol{\theta}}(s, a) := \nabla_{\boldsymbol{\theta}} \log \pi_{\boldsymbol{\theta}}(a|s)$ be the score function for state-action pair $(s, a)$. Then, the gradient policy of the $i$-th objective can be computed as follows:

**Lemma 2** (Policy Gradient Theorem). *Let $\pi_{\boldsymbol{\theta}} : \mathcal{S} \times \mathcal{A} \to [0, 1]$ be any policy and $J^i(\boldsymbol{\theta})$ be the accumulated reward function for the $i$-th objective. Then, the policy-gradient of $J^i(\boldsymbol{\theta})$ with respect to policy parameter $\boldsymbol{\theta}$ is:* $\nabla_{\boldsymbol{\theta}} J^i(\boldsymbol{\theta}) = \mathbb{E}_{s \sim d_{\boldsymbol{\theta}}(\cdot), a \sim \pi_{\boldsymbol{\theta}}(\cdot|s)}[\boldsymbol{\psi}_{\boldsymbol{\theta}}(s, a) \cdot \mathrm{Adv}_{\boldsymbol{\theta}}^i(s, a)]$.

We note that Lemma 2 is a straightforward adaptation of the policy gradient theorem in conventional RL Sutton et al. (1999) to each individual objective $i \in [M]$ in the MORL setting.

**3) Function Approximation:** Similar to single-objective actor-critic methods, our WC-MOAC algorithm adopts linear function approximation. Toward this end, we have the following assumptions:

**Assumption 2** (Function Approximation). The value function of each objective $i$ can be approximated by a linear function: $V^i(s) \approx \boldsymbol{\phi}(s)^\top \mathbf{w}^i, i \in [M]$, where $\mathbf{w}^i \in \mathbb{R}^{d_2}$ with $d_2 \leq |\mathcal{S}|$ is a parameter to be learnt, and $\boldsymbol{\phi}(s) \in \mathbb{R}^{d_2}$ is the feature mapping associated with state $s \in \mathcal{S}$ that satisfies:

(a) All features are bounded. Without loss of generality, we further assume $\|\boldsymbol{\phi}(s)\|_2 \leq 1, \forall s \in \mathcal{S}$;

(b) The feature matrix $\Phi \in \mathbb{R}^{|\mathcal{S}| \times d_2}$ is full rank.

Assumption 2 is standard and has been widely used in the RL literature (e.g., (Tsitsiklis & Van Roy, 1999; Zhang et al., 2018; Qiu et al., 2021)). We note that linear representation includes tabular setting as a special case by letting $\boldsymbol{\phi}(s)$ be an appropriate unit vector when $d_2 = |\mathcal{S}|$. For simplicity, in this paper, we assume that the same feature mapping is shared among all objectives.

## 4.2 THE PROPOSED WC-MOAC ALGORITHM FRAMEWORK

With the preliminaries in Section 4.1, we are in a position to present our WC-MOAC algorithm. For ease of exposition, we will structure our WC-MOAC algorithm design in two main derivation steps.

**Step 1) Multiple-TD Learning in the Critic Component:** As stated in Assumption 2, the critic component (i.e., policy evaluation) in WC-MOAC maintains value-function approximation parameters $\mathbf{w}^i$ for each objective $i \in [M]$. For the current policy $\pi_{\boldsymbol{\theta}_t}$, the critic component in WC-MOAC updates the value function parameters $\mathbf{w}_k^i, i \in [M]$ in parallel via TD learning with mini-batch Markovian samples. The TD-error $\delta_{k,\tau}^i$ for objective $i$ in iteration $k$ using sample $\tau$ can be computed as:

- *Average Reward Setting:* 
$$\mu_{k,\tau}^i = (1 - \beta)\mu_{k,\tau-1}^i + \beta r_{k,\tau}^i, \tag{3}$$
$$\delta_{k,\tau}^i = r_{k,\tau}^i - \mu_{k,\tau}^i + \boldsymbol{\phi}^\top(s_{k,\tau+1})\mathbf{w}_k^i - \boldsymbol{\phi}^\top(s_{k,\tau})\mathbf{w}_k^i, \tag{4}$$

where the $\mu^i$-values are to keep track of the $J^i(\boldsymbol{\theta}_t)$-information in the average reward setting.

- *Discounted Reward Setting:* $\delta_{k,\tau}^i = r_{k,\tau}^i + \gamma^i \boldsymbol{\phi}^\top(s_{k,\tau+1})\mathbf{w}_k^i - \boldsymbol{\phi}^\top(s_{k,\tau})\mathbf{w}_k^i. \tag{5}$

Subsequently, each parameter $\mathbf{w}^i$ is updated in a batch fashion in parallel using the following TD-learning step: $\mathbf{w}_k^i = \mathbf{w}_{k-1}^i + (\beta/D)\sum_{\tau=1}^D \delta_{k,\tau}^i \cdot \boldsymbol{\phi}(s_{k,\tau})$. Once the critic component executes $N$ rounds, the parameters $\{\mathbf{w}^i\}_{i \in [M]}$ can be used in the actor component for policy evaluation.

**Step 2) The WC-MGDA-Type Policy Gradient in the Actor Component:** As mentioned earlier, the actor component in WC-MOAC is a "multi-gradient" extension of the policy gradient approach in MORL, which determines a *common policy improvement direction* for all reward objectives by dynamically weighting the individual policy gradients. Toward this end, we will further organize the common policy improvement direction derivations in two key steps as follows:

*Step 2-a) WC-Guided Common Policy Improvement Direction:* First, we compute a dynamic weighting vector $\hat{\boldsymbol{\lambda}}_t^*$ in each iteration $t$ that balances two key aspects: 1) find a common policy improvement

direction based on multi-TD learning to converge to a Pareto-stationary solution; and 2) follow the guidance of a WC-scalarization weight vector $\mathbf{p}$. To adopt an MGDA-type policy improvement update in WC-MOAC, we first convert the original MORL reward maximization problem in Eq. (1) to the following logically equivalent "regret minimization" problem with respect to the Pareto front:

$$\min_{\boldsymbol{\theta} \in \mathbb{R}^{d_1}} (\mathbf{J}_{\text{ub}}^* - \mathbf{J}(\boldsymbol{\theta})) := \left[ J_{\text{ub}}^{1,*} - J^1(\boldsymbol{\theta}), J_{\text{ub}}^{2,*} - J^2(\boldsymbol{\theta}), \dots, J_{\text{ub}}^{M,*} - J^M(\boldsymbol{\theta}) \right]^\top, \quad (6)$$

where $J_{\text{ub}}^{i,*}$ is an estimated upper bound of $J^{i,*} := \max_{\boldsymbol{\theta} \in \mathbb{R}^{d_1}} J^i(\boldsymbol{\theta})$ (i.e., the optimal value of the $i$-th objective under single-objective RL). The rationale behind using $\mathbf{J}_{\text{ub}}^*$ in (6) is to ensure that the polarity of the reformulated problem is conformal to the standard use of WC-scalarization in MOO. Note that, regardless of the choice of the $\mathbf{J}_{\text{ub}}^*$-estimation, there is always a 1-to-1 mapping between the Pareto fronts between Problems (1) and (6). Hence, using the WC-scalarization to explore the Pareto front of Problem (6) is logically equivalent to exploring the Pareto front of Problem (1), and the tightness of the $\mathbf{J}_{\text{ub}}^*$-estimation is not important. Next, since Problem (6) is in the standard MOO form, according to (Désidéri, 2012), the MGDA approach for Problem (6) can be written as:

$$\min \|\mathbf{K}\boldsymbol{\lambda}\|^2 \quad \text{s.t.} \quad \mathbf{1}^\top \boldsymbol{\lambda} = 1, \ \boldsymbol{\lambda} \in \mathbb{R}_+^M, \quad (7)$$

where $\mathbf{K} := \sqrt{\mathbf{G}^\top \mathbf{G}}$ and and $\mathbf{G}$ is the gradient matrix of $\mathbf{J}_{\text{ub}}^* - \mathbf{J}(\boldsymbol{\theta})$. On the other hand, following Eq. (2), the WC-scalarization of Eq. (6) with a given weight vector $\mathbf{p}$ is: $\min_{\boldsymbol{\theta} \in \mathbb{R}^{d_1}} \|\mathbf{p} \odot (\mathbf{J}_{\text{ub}}^* - \mathbf{J}(\boldsymbol{\theta}))\|_\infty$, which can be reformulated as follows by introducing an auxiliary variable $\rho$:

$$\min_{\rho \in \mathbb{R}, \boldsymbol{\theta} \in \mathbb{R}^{d_1}} \rho \quad \text{s.t.} \quad \mathbf{p} \odot (\mathbf{J}_{\text{ub}}^* - \mathbf{J}(\boldsymbol{\theta})) \leq \rho \mathbf{1}. \quad (8)$$

By the KKT stationarity condition on $\rho$ and $\boldsymbol{\theta}$ and associating Lagrangian dual variables $\boldsymbol{\lambda} \in \mathbb{R}_+^M$, it can be readily verified that the Wolfe dual problem of Eq. (8) can be written as (Momma et al., 2022):

$$\max \boldsymbol{\lambda}^\top (\mathbf{p} \odot (\mathbf{J}_{\text{ub}}^* - \mathbf{J}(\boldsymbol{\theta}))), \ \text{s.t.} \ \mathbf{K}_{\mathbf{p}} \boldsymbol{\lambda} = 0, \ \mathbf{1}^\top \boldsymbol{\lambda} = 1, \ \boldsymbol{\lambda} \in \mathbb{R}_+^M, \ \boldsymbol{\theta} \in \mathbb{R}^{d_1}, \quad (9)$$

where $\mathbf{K}_{\mathbf{p}} := \text{diag}(\sqrt{\mathbf{p}}) \sqrt{\mathbf{G}^\top \mathbf{G}} \text{diag}(\sqrt{\mathbf{p}})$. Since the condition $\mathbf{K}_{\mathbf{p}} \boldsymbol{\lambda} = \mathbf{0}$ may not be satisfied at all iterations in an algorithm, we incorporate the minimization of $\|\mathbf{K}_{\mathbf{p}} \boldsymbol{\lambda}\|^2$ in (9) using a parameter $u > 0$ to balance the trade-off with the objective $\boldsymbol{\lambda}^\top (\mathbf{p} \odot (\mathbf{J}_{\text{ub}}^* - \mathbf{J}(\boldsymbol{\theta})))$ to yield:

$$\min \|\mathbf{K}_{\mathbf{p}} \boldsymbol{\lambda}\|^2 - u \boldsymbol{\lambda}^\top (\mathbf{p} \odot (\mathbf{J}_{\text{ub}}^* - \mathbf{J}(\boldsymbol{\theta}))) \quad \text{s.t.} \quad \mathbf{1}^\top \boldsymbol{\lambda} = 1, \ \boldsymbol{\lambda} \in \mathbb{R}_+^M, \boldsymbol{\theta} \in \mathbb{R}^{d_1}. \quad (10)$$

Now, comparing (10) with (7) and (9), it is clear that solving for $\boldsymbol{\lambda}$ in Problem (10) under the current $\boldsymbol{\theta}$-value yields a $\boldsymbol{\lambda}$-weighting of the gradients of $(\mathbf{J}_{\text{ub}}^* - \mathbf{J}(\boldsymbol{\theta}))$, which achieves a balance between Pareto-front exploration and Pareto-stationarity induced by WC and MGDA, respectively. Moreover, upon fixing a $\boldsymbol{\theta}$-value, solving for $\boldsymbol{\lambda}$ in Problem (10) is a convex quadratic program (QP), which can be efficiently solved similar to the standard MGDA (Désidéri, 2012). In iteration $t$, let $\hat{\boldsymbol{\lambda}}_t^*$ be the solution obtained from solving Problem (10) under current policy parameter $\boldsymbol{\theta}_t$. To mitigate the cumulative systematic bias resulting from $\boldsymbol{\lambda}_t$-weighting, we show that (cf. the Appendix) one can update $\boldsymbol{\lambda}_t$ by using a momentum-based approach with momentum coefficient $\eta_t \in [0, 1)$ as follows:

$$\boldsymbol{\lambda}_t = (1 - \eta_t) \boldsymbol{\lambda}_{t-1} + \eta_t \hat{\boldsymbol{\lambda}}_t^*. \quad (11)$$

Next, with the obtained $\boldsymbol{\lambda}_t$ from (11), we can update policy parameters $\boldsymbol{\theta}$ by conducting a gradient-descent-type update in (10) as follows: $\boldsymbol{\theta}_{t+1} = \boldsymbol{\theta}_t - \alpha \mathbf{G}_t (\mathbf{p} \odot \boldsymbol{\lambda}_t)$ with step size $\alpha > 0$.

*Step 2-b) Policy Gradient Computation for Individual Reward Objective:* Although we have derived the WC-MGDA-type update in Step 2-a, it remains to evaluate the gradient matrix $\mathbf{G}$ of $(\mathbf{J}_{\text{ub}}^* - \mathbf{J}(\boldsymbol{\theta}))$. Note that $\mathbf{J}_{\text{ub}}^*$ is a constant, each column $\mathbf{g}_t^i$ in $\mathbf{G}$ is equal to the negative policy gradient of each reward objective $i$. To compute $\mathbf{g}_t^i$, the actor component starts with sampling and TD-error computations. First, from Lemma 2, we compute the score function in the $l$-th actor step as follows:

$$\boldsymbol{\psi}_{t,l} := \nabla_{\boldsymbol{\theta}} \log \pi_{\boldsymbol{\theta}_t}(a_{t,l}|s_{t,l}). \quad (12)$$

Next, similar to the critic component, the actor computes the TD-error for objective $i$ at time $t$ using sample $l$ can be computed as follows:

- *Average Reward Setting:* $\quad \mu_{t,l}^i = (1 - \alpha) \mu_{t,l}^i + \alpha r_{t,l}^i, \quad (13)$

$$\delta_{t,l}^i = r_{t,l}^i - \mu_{t,l}^i + \boldsymbol{\phi}^\top(s_{t,l+1}) \mathbf{w}_t^i - \boldsymbol{\phi}^\top(s_{t,l}) \mathbf{w}_t^i, \quad (14)$$

where the $\mu^i$-values are to keep track of the $J^i(\boldsymbol{\theta}_t)$-information in the average reward setting.

---

**Algorithm 1:** The WC-MOAC Algorithm.

---

**Input** : $s_0$, $\boldsymbol{\theta}_1$, $\Phi$, $\{\mathbf{w}_0^i\}_{i\in[M]}$, $\{\mu_{1,0}^i\}_{i\in[M]}$, $\mathbf{p}$, $\{\eta_t\}_{t\in[T]}$, actor step size $\alpha$, actor iteration $T$,
actor batch size $B$, critic step size $\beta$, critic iteration $N$, critic batch size $D$

**for** $t = 1, \cdots, T$ **do**

    **Critic Component:**
    **for** $k = 1, \cdots, N$ **do**
        $s_{k,1} = s_{k-1,D}$ (when $k = 1$, $s_{1,1} = s_0$)
        **for** $\tau = 1, \cdots, D$ **do**
            execute action $a_{k,\tau} \sim \pi_{\boldsymbol{\theta}_t}(\cdot|s_{k,\tau})$,
            observe state $s_{k,\tau+1}$, reward $\mathbf{r}_{k,\tau+1}$
            **for** $i \in [M]$ **do in parallel**
                ● *Setting I: Average Reward:*
                update $\mu_{k,\tau}^i$, $\delta_{k,\tau}^i$ by Eqs. (3),(4),
                respectively
                ● *Setting II: Discounted Reward:*
                update $\delta_{k,\tau}^i$ by Eq. (5)
        **for** $i \in [M]$ **do in parallel**
            TD update:
            $\mathbf{w}_k^i = \mathbf{w}_{k-1}^i + \frac{\beta}{D} \sum_{\tau=1}^{D} \delta_{k,\tau}^i \cdot \boldsymbol{\phi}(s_{k,\tau})$
    **for** $i \in [M]$ **do in parallel**
        denote $\mathbf{w}_t^i = \mathbf{w}_k^i$

    **Actor Component:**
    **for** $l = 1, \cdots, B$ **do**
        execute action $a_{t,l} \sim \pi_{\boldsymbol{\theta}_t}(\cdot|s_{t,l})$,
        observe state $s_{t,l+1}$, reward $\mathbf{r}_{t,l+1}$
        **for** $i \in [M]$ **do in parallel**
            update $\boldsymbol{\psi}_{t,l}$ by Eq. (12),
            ● *Setting I: Average Reward:* update
            $\mu_{t,l}^i$, $\delta_{t,l}^i$ by Eqs. (13),(14), respectively
            ● *Setting II: Discounted Reward:*
            update $\delta_{t,l}^i$ by Eq. (15)
    **for** $i \in [M]$ **do in parallel**
        $\mathbf{g}_t^i = -\frac{1}{B} \sum_{l=1}^{B} \delta_{t,l}^i \cdot \boldsymbol{\psi}_{t,l}$
    Solve for $\hat{\boldsymbol{\lambda}}_t^*$ in Problem (10) under current $\boldsymbol{\theta}_t$;
    Update $\boldsymbol{\lambda}_t$ by Eq. (11);
    Update $\mathbf{g}_t = \mathbf{G}_t(\mathbf{p} \odot \boldsymbol{\lambda}_t)$;
    Update policy: $\boldsymbol{\theta}_{t+1} = \boldsymbol{\theta}_t - \alpha \cdot \mathbf{g}_t$

**Output** : $\boldsymbol{\theta}_{\hat{T}}$ with $\hat{T}$ chosen uniformly random from $\{1, \cdots, T\}$

---

● *Discounted Reward Setting:* $\quad \delta_{t,l}^i = r_{t,l}^i + \gamma^i \boldsymbol{\phi}^\top(s_{t,l+1})\mathbf{w}_t^i - \boldsymbol{\phi}^\top(s_{t,l})\mathbf{w}_t^i.$     (15)

With the score function in (12) and the TD-error in (13) or (14) depending on the reward setting, one can compute the individual policy gradient as $\mathbf{g}_t^i = -\frac{1}{B} \sum_{l=1}^{B} \delta_{t,l}^i \cdot \boldsymbol{\psi}_{t,l}$ following Lemma 2.

Lastly, to conclude the discussion on the WC-MOAC algorithmic development, we summarize the full WC-MOAC algorithm in Algorithm 1.

### 4.3 THEORETICAL PERFORMANCE OF WC-MOAC

In this section, we analyze WC-MOAC's convergence to a Pareto-stationary solution and the associated sample complexity of the WC-MOAC. Due to space limitations, we relegate all proofs to the Appendix. For finite-time Pareto-stationary convergence analysis, instead of using the original definition in Defition 2, it is more convenient to use the following equivalent near-Pareto stationarity characterization defined as follows (Désidéri, 2012; Sener & Koltun, 2018; Yang et al., 2024):

**Definition 3.** ($\epsilon$-Pareto Stationary Point) For a given $\epsilon > 0$, a solution $\boldsymbol{\theta}$ is $\epsilon$-Pareto stationary if there exists $\boldsymbol{\lambda} \in \mathbb{R}_+^M$ satisfying $\boldsymbol{\lambda} \geq \mathbf{0}$, $\mathbf{1}^\top \boldsymbol{\lambda} = 1$, such that $\min_{\boldsymbol{\lambda}} \|\nabla_{\boldsymbol{\theta}}\mathbf{J}(\boldsymbol{\theta})\boldsymbol{\lambda}\|_2^2 \leq \epsilon$, where

$$\nabla_{\boldsymbol{\theta}}\mathbf{J}(\boldsymbol{\theta}) = \begin{bmatrix} \nabla_{\boldsymbol{\theta}}J^1(\boldsymbol{\theta}) & \nabla_{\boldsymbol{\theta}}J^2(\boldsymbol{\theta}) & \cdots & \nabla_{\boldsymbol{\theta}}J^M(\boldsymbol{\theta}) \end{bmatrix} \in \mathbb{R}^{d_1 \times M}.$$

Next, we state the following assumptions needed for our Pareto-stationary convergence analysis:

**Assumption 3.** For any two policy parameters $\boldsymbol{\theta}, \boldsymbol{\theta}' \in \mathbb{R}^{d_1}$, and any state-action pair $(s, a) \in \mathcal{S} \times \mathcal{A}$, there exist positive constants $C_{\boldsymbol{\psi}}, L > 0$ such that the following hold: (a) $\|\boldsymbol{\psi}_{\boldsymbol{\theta}}(s,a)\|_2 \leq C_{\boldsymbol{\psi}}$; and (b) $\|\nabla_{\boldsymbol{\theta}}J^i(\boldsymbol{\theta}) - \nabla_{\boldsymbol{\theta}}J^i(\boldsymbol{\theta}')\|_2 \leq L_J\|\boldsymbol{\theta} - \boldsymbol{\theta}'\|_2, \forall i \in [M]$.

In Assumption 3, Part (a) requires that the score function is uniformly bounded for any policy and state-action pair and Part (b) requires the gradient of each objective function is Lipschitz with respect to the policy parameter. These assumptions are standard and has been adopted in the analysis of the single-objective actor-critic RL algorithms in (Qiu et al., 2021; Xu et al., 2020). For discounted reward setting, both items can be guaranteed by choosing common policy parameterizations (Xu et al., 2020). For average reward setting, both assumptions can also be satisfied by the popular

class of soft-max policy under Assumption 1 (Guo et al., 2021). The following lemma characterizes the mixing time of the underlying Markov chain and the data sampled in WC-MOAC follows such Markovian chain, which holds under Assumption 1 (Levin & Peres, 2017, Theorem 4.9).

**Lemma 3.** *For any policy $\pi_{\boldsymbol{\theta}}$, consider an MDP with $P(\cdot \mid s, a)$ and stationary distribution $d_{\boldsymbol{\theta}}(\cdot)$. There exist constants $\kappa > 0$ and $\rho \in (0, 1)$ such that $\sup_{s \in \mathcal{S}} \|P(s_t \mid s_0 = s) - d_{\boldsymbol{\theta}}(\cdot)\|_{TV} \leq \kappa \rho^t$.*

We let $\zeta_{\text{approx}} := \max_{i \in [M]} \max_{\boldsymbol{\theta}} \mathbb{E}[|V^i(s) - V^i_{\mathbf{w}^{i,*}}(s)|^2]$ represent the approximation error of the critic component, which is zero if the ground-truth value functions $V^i(\cdot)$, $\forall i$, are in the linear function class; otherwise, $\zeta_{\text{approx}}$ is non-zero due to the expressivity limit of the critics. We now state our main convergence theorem of WC-MOAC to a neighborhood of a Pareto-stationary point as follows:

**Theorem 4.** *Under Assumptions 1-3, set the actor and critic step sizes as $\alpha = \frac{1}{3L_J}$ and $0 < \beta \leq \min\{\frac{\lambda_{\mathbf{A}}}{8C_{\mathbf{A}}^2}, \frac{4}{\lambda_{\mathbf{A}}}\}$, where $C_{\mathbf{A}}$ is a constant depending on the problem setting. Then, the iterations generated by Algorithm 1 satisfy the following finite-time Pareto-stationary convergence error bound:*

$$\mathbb{E}\big[\|\boldsymbol{\lambda}_{\hat{T}}^{*\top} \nabla_{\boldsymbol{\theta}} \mathbf{J}(\boldsymbol{\theta}_{\hat{T}})\|_2^2\big] \leq \frac{16 L_J r_{\max}}{\zeta_1 T} \left(1 + \frac{2}{p_{\min}^2} \sum_{t=1}^{T} \eta_t\right) + \frac{60}{T} \sum_{t=1}^{T} \max_{j \in [M]} \mathbb{E}\left[\left\|\mathbf{w}_t^j - \mathbf{w}_t^{j,*}\right\|_2^2\right]$$

$$\frac{\zeta_2(1 - \rho + 4\kappa\rho)}{(1 - \rho)B} + 60\zeta_{\text{approx}},$$

*where $\hat{T}$ is sampled uniformly among $\{1, \cdots, T\}$ and (i) for average setting $\zeta_1 = 1$ and $\zeta_2 = 240(r_{\max} + R_{\mathbf{w}})^2$; and (ii) for discounted setting $\zeta_1 = 1 - \|\boldsymbol{\gamma}\|_{\infty}$ and $\zeta_2 = 60(r_{\max} + 2R_{\mathbf{w}})^2$.*

Two remarks on Theorem 4 are in order: (1) Theorem 4 depends on the momentum coefficients $\eta_t \in [0, 1]$ in Eq. (11). By letting $\eta_t$ to be iteration-dependent, e.g., $\eta_t = t^{-2}$, then WC-MOAC guarantees convergence to a neighborhood of Pareto-stationarity at a rate of $\mathcal{O}(T^{-1})$. (2) Theorem 4 also suggests that the convergence depends on the the minimum entry $p_{\min}$ of the WC-scalarization weight vector $\mathbf{p}$: the smaller $p_{\min}$, the longer Pareto-stationary convergence time. The following Pareto-stationarity sample complexity result immediately follows from Theorem 4:

**Corollary 5.** *Under the same conditions as in Theorem 4, for any $\epsilon > 0$, by setting $T \geq 16L_J r_{\max}/(C_4\epsilon) \cdot (1 + \frac{2}{p_{\min}^2} \sum_{t=1}^{T} \eta_t)$, $\mathbb{E}[\|\mathbf{w}_t^i - \mathbf{w}_t^{i,*}\|_2^2] \leq \epsilon/12, \forall i \in [M]$, and $B \geq C_5(1 - \rho + 4\kappa\rho)/(\epsilon(1 - \rho))$, we have $\mathbb{E}\big[\|\boldsymbol{\lambda}_{\hat{T}}^{*\top} \nabla_{\boldsymbol{\theta}} \mathbf{J}(\boldsymbol{\theta}_{\hat{T}})\|_2^2\big] \leq \epsilon + \mathcal{O}(\zeta_{\text{approx}})$, with total sample complexity of $\mathcal{O}(\epsilon^{-2} p_{\min}^{-2} \log(\epsilon^{-1}))$. Further, by setting $\eta_t = p_{\min}^2/t^2$, the sample complexity is $\mathcal{O}(\epsilon^{-2} \log(\epsilon^{-1}))$.*

Note that Theorem 4 and Corollary 5 show the convergence rate of WC-MOAC are *independent* of the number of objectives $M$, and the sample complexity of WC-MOAC is the *same* as the state-of-the-art sample complexity for single-objective RL (Xu et al., 2020).

## 5 EXPERIMENTS

In this section, we conduct experiments to evaluate our algorithm and compare it with other related state-of-the-art methods on a large-scale real-world dataset. Due to space limitations, we present the main experimental results here and relegate the full experimental setting details to the Appendix.

**1) Dataset:** We leverage a large-scale real-world dataset from the recommendation logs of the short video streaming mobile app Kuaishou[4]. The dataset includes multiple reward signals, such as "Click", "Like", "Comment", "Dislike", "WatchTime," etc. The full statistics of the dataset is shown in Table 2 in the Appendix. Here, a state corresponds to the event that a video is watched by a user and is formed by concatenating user and video features; an action corresponds to recommending a video to a user.

**2) Baselines:** In this experiment, we leverage the following state-of-the-art methods as baselines:

- **Behavior-Clone**: A supervised behavior-cloning policy $\pi_{\beta}$ to mimic the recommendation policy in the dataset, which takes the user states as inputs and the video IDs as outputs.
- **TSCAC** (Cai et al., 2023): An $\epsilon$-constrained actor-critic approach that optimizes a single objective (i.e., "WatchTime"), while treating other objectives as constraints bounded by some $\epsilon > 0$.

---

[4]https://kuairand.com/

Table 1: Comparison of WC-MOAC with baseline methods given a weight vector.

| Objective weights | Click↑ 0.2 | Like↑(e-2) 0.2 | Comment↑(e-3) 0.2 | Dislike↓(e-4) 0 | WatchTime↑ 0.4 |
|---|---|---|---|---|---|
| Behavior-Clone | 5.338 | 1.231 | 3.225* | 2.304 | 1.285 |
| TSCAC | 5.485 2.75% | 1.328 7.88% | 2.877 −10.80% | 1.177 −48.92% | 1.365 6.23% |
| SDMGrad | 5.434 1.79% | 1.279 3.87% | 3.136 −2.77% | 1.166* −49.41%* | 1.329 3.46% |
| WC-MOAC (Ours) | **5.550** **3.97%** | **1.329** **7.96%** | 3.092 −4.12% | 1.339 −41.88% | **1.375** **7.00%** |

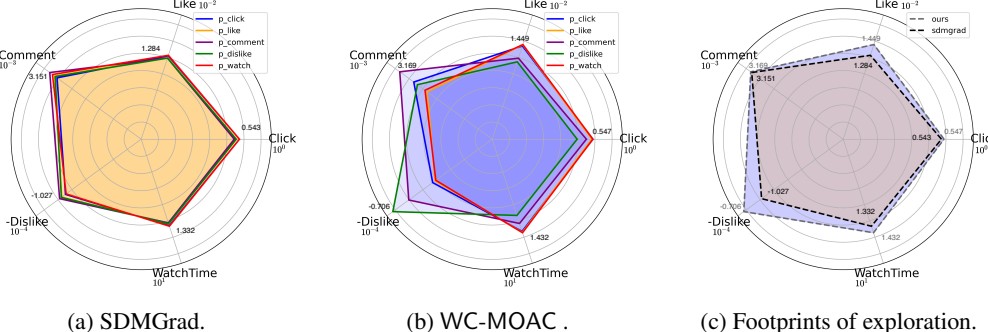

(a) SDMGrad.     (b) WC-MOAC .     (c) Footprints of exploration.

Figure 1: Comparison of WC-MOAC and SDMGrad with five one-hot weight vectors.

- **SDMGrad** (Xiao et al., 2024): A weight/direction vector $\mathbf{p}$ oriented stochastic gradient descent algorithm, which is shown to find an $\epsilon$-accurate Pareto stationary point.

Due to the fact that the Kuaishou dataset is a static offline dataset and all baselines are off-policy, for fair comparisons, we also adapt WC-MOAC to the off-policy setting. We adopt normalised capped importance sampling (NCIS), a standard evaluation approach for off-policy RL algorithms (Zou et al., 2019) to evaluate all methods. By definition, a larger NCIS score implies a better policy for reward maximization. The definition of NCIS is provided in Section A.1.

**3) Results and Observations:** We summarize the performance of all methods based on a given weight vector in Table 1, and only illustrate the comparison between WC-MOAC and SDMGrad (since TSCAC cannot explore Pareto front) in Fig. 1. In Table 1, we set the weight vector $\mathbf{p}$ to be $(0.2, 0.2, 0.2, 0, 0.4)^{\top}$ for "Click", "Like", "Comment", "Dislike", and "WatchTime", respectively. Note that TSCAC does not require a weight vector since it only optimizes "WatchTime". All methods start with the same critic and actor parameters initialized for policies that perform worse than Behavior-Clone. From Table 1, we observe that WC-MOAC outperforms SDMGrad and TSCAC in three out of four objectives, i.e., "Click", "Like", and "WatchTime", implying that WC-MOAC is more aligned with the weighted objectives. In Fig. 1, we set the weight vector to be one-hot vectors with "Click", "Like", "Comment", "Dislike", and "WatchTime" as the only objective, respectively. All figures are plotted in the same scale. Comparing Fig. 1a and Fig. 1b, we observe that i) WC-MOAC is more aligned with weight vector in all directions; ii) among all the weight vector directions, WC-MOAC possesses a larger footprint in the radar chart than SDMGrad (see Fig. 1c), which shows that WC-MOAC is closer to being Pareto-optimal and has a better Pareto front exploration performance.

# 6 CONCLUSION

In this paper, we proposed a weighted Chebyshev multi-objective actor-critic (WC-MOAC) algorithm for multi-objective reinforcement learning (MORL). Our proposed WC-MOAC method judiciously integrates weighted Chebyshev and multi-policy-gradient techniques to facilitate systematic Pareto-stationary solution exploration with provable finite-time sample complexity guarantee. Our numerical experiments with real-world datasets also verified the theoretical results of our WC-MOAC method and its practical effectiveness.

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
