## A    EXPERIMENTAL SETUP AND COMPLEMENTARY RESULTS

### A.1    REAL-WORLD DATA

**Environment and Setup.** The data statistics are provided in the Table 2. In the dataset, logs provided by the same user are concatenated to form a trajectory in one episode, and a batch of tuple $\{s_t, a_t, \boldsymbol{r}_t, s_{t+1}\}$ are sampled at each iteration. For all the methods, we leverage ADAM to optimize the parameters. We only experiment on discounted total reward for fair comparison. For our method, we set the momentum coefficient of gradient weight by $\eta_t = 1/t$ (without pre-specifying values, the gradient weights are initialized by the solution to a QP problem regarding the average gradients of the first batch of samples), and set the same gradient weight initialization for all the other methods.

Table 2: Data statistic. The reward data is imbalanced, with a density of over $98\%$ for the sum of Click and WatchTime.

| State: 1218 | Action: 150 | | | | |
| --- | --- | --- | --- | --- | --- |
| | | Reward | | | |
| | Click | Like | Comment | Dislike | WatchTime |
| Amount | 254940 | 5190 | 1438 | 213 | 199122 |
| Density | 55.25% | 1.125% | 0.312% | 0.046% | 43.15% |

**Evaluation Metric.** Specifically, NCIS score is defined as follows:

$$N(\pi) = \frac{\sum_{s,a \in D} w(s,a) r(s,a)}{\sum_{s,a \in D} w(s,a)}, \quad w(s,a) = \min\left\{C, \frac{\pi(a \mid s)}{\pi_\beta(a \mid s)}\right\},$$

where $D$ is the dataset, $C$ is a positive constant, and $\pi_\beta$ is a behavior policy.

### A.2    ADDITIONAL EMPIRICAL RESULTS

In this subsection, we provide additional empirical results for WC-MOAC under varying weight vectors $\mathbf{p}$. Specifically, in addition to the 5 one-hot vectors, we have chosen the weight vectors to be as follows in Table 3. The corresponding results in radar chart are provided in Figure 2. In Figure

Table 3: Additional Weight Vectors **p**

| radar result | click | like | comment | dislike | watchtime |
|:---:|:---:|:---:|:---:|:---:|:---:|
| abl1 | 0.85 | 0.05 | 0.05 | 0 | 0.05 |
| abl2 | 0.7 | 0.1 | 0.1 | 0 | 0.1 |
| abl3 | 0.55 | 0.15 | 0.15 | 0 | 0.15 |
| abl4 | 0.4 | 0.2 | 0.2 | 0 | 0.2 |
| abl5 | 0.05 | 0.05 | 0.85 | 0.0001 | 0.05 |
| abl6 | 0.10 | 0.10 | 0.70 | 0.0001 | 0.10 |
| abl7 | 0.15 | 0.15 | 0.55 | 0.0001 | 0.15 |

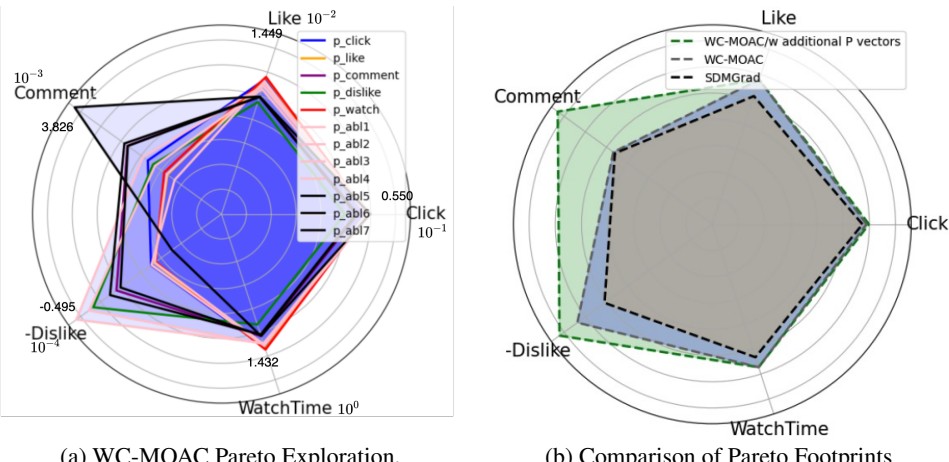

(a) WC-MOAC Pareto Exploration.    (b) Comparison of Pareto Footprints

Figure 2: Comparison of WC-MOAC and SDMGrad with additional weight vectors.

2a, we show the Pareto solutions explored by the 7 ablation **p** vectors in addition to those from the one-hot vectors. In Figure 2b, we further show the footprint of exploration that includes the additional **p** vectors.

From the empirical results in Figure 2a, we can see that with additional weight vectors **p**, WC-MOAC is exploring more Pareto stationary solutions compared to WC-MOAC with only one-hot vectors as the weight vectors. In Figure 2b, it further shows that with more **p** vectors, WC-MOAC explores even wider Pareto footprints. This further confirms our theoretical prediction as well as strengthens the empirical observation that, with increasing number of weight/explore vectors **p**, WC-MOAC possess the potential to explore more Pareto stationary points.

## B SUPPORTING DEFINITIONS, LEMMAS AND CRITIC RESULTS

### B.1 DEFINITIONS AND ADDITIONAL ASSUMPTIONS

Here, we first define some standard terms and reiterate Assumption 2 for clarity.

For each objective $i \in [M]$, we define the state-action value function as follows: (i) for average total reward: $Q_{\boldsymbol{\theta}}^i(s,a) := \mathbb{E}\left[\sum_{t=0}^{\infty} r^i(s_t, a_t) - J^i(\boldsymbol{\theta})|s_0 = s, a_0 = a\right]$, and (ii) for discounted total reward: $Q_{\boldsymbol{\theta}}^i(s,a) = \mathbb{E}\left[\sum_{t=0}^{\infty} (\gamma^i)^t r^i(s_t, a_t)|s_0 = s, a_0 = a\right]$. It then follows that the value function satisfies: $V_{\boldsymbol{\theta}}^i(s) = \sum_{a \in \mathcal{A}} Q_{\boldsymbol{\theta}}^i(s,a) \cdot \pi_{\boldsymbol{\theta}}(a|s)$. We define the advantage function as follows: $\text{Adv}_{\boldsymbol{\theta}}^i(s,a) = Q_{\boldsymbol{\theta}}^i(s,a) - V_{\boldsymbol{\theta}}^i(s), \forall i \in [M]$.

**Assumption 4** (Reiteration of Assumption 2). The value function of each objective $i$ is approximated by a linear function: $V^i(s) \approx \phi(s)^\top \mathbf{w}^i, i \in [M]$, where $\mathbf{w}^i \in \mathbb{R}^{d_2}$ with $d_2 \leq |\mathcal{S}|$ is a parameter to be learnt, and $\phi(s) \in \mathbb{R}^{d_2}$ is the feature associated with state $s \in \mathcal{S}$, which satisfies:
(a) All features are normalized, i.e., $\|\phi(s)\|_2 \leq 1, \forall s \in \mathcal{S}$;

(b) The feature matrix $\Phi \in \mathbb{R}^{|\mathcal{S}| \times d_2}$ is full rank;

(c) For any $u \in \mathbb{R}^{d_2}$, $\Phi u \neq \mathbf{1}$, where $\mathbf{1} \in \mathbb{R}^{d_2}$;

(d) Let $\mathbf{A}_{\boldsymbol{\theta}} := \mathbb{E}_{s \sim d_{\theta}(\cdot), s' \sim P(\cdot|s)}[(\phi(s') - \phi(s))\phi^{\top}(s)]$ if in average reward setting. Otherwise, if in discounted reward setting, let $\mathbf{A}_{\boldsymbol{\theta}} := \mathbb{E}_{s \sim d_{\theta}(\cdot), s' \sim P(\cdot|s)}\left[(\gamma\phi(s') - \phi(s))\phi^{\top}(s)\right]$. Then, there exists a constant $\lambda_{\mathbf{A}} > 0$ such that $\lambda_{\max}(\mathbf{A}_{\boldsymbol{\theta}} + \mathbf{A}_{\boldsymbol{\theta}}^{\top}) \leq -\lambda_{\mathbf{A}}$ for all $\theta \in \mathcal{R}^{d_1}$, where $\lambda_{\max}(\mathbf{A})$ is the largest eigenvalue of the matrix $\mathbf{A}$.

Assumption 2 item (c) and item (d), which are used for average reward setting, imply that for any policy $\pi_{\boldsymbol{\theta}}$, the inequality $\mathbf{w}^{\top}\mathbf{A}_{\boldsymbol{\theta}}\mathbf{w} < 0$ holds for any $\mathbf{w} \neq 0$, and $\mathbf{A}_{\pi_{\boldsymbol{\theta}}}$ is invertible with $\lambda_{\max}(\mathbf{A}_{\boldsymbol{\theta}} + \mathbf{A}_{\boldsymbol{\theta}}^{\top}) \leq 0$. This ensures that the optimal approximation $\mathbf{w}_{\boldsymbol{\theta}}^{i,*}$ for any given policy $\pi_{\boldsymbol{\theta}}$ and $i \in [M]$ is uniformly bounded. Assumption 4 has been widely use in the literature (e.g., Tsitsiklis & Van Roy (1999); Zhang et al. (2018); Qiu et al. (2021)).

## B.2 SUPPORTING LEMMAS

**Lemma 6** (Average reward setting). *Given a policy $\pi_{\boldsymbol{\theta}}$, for any objective $i \in [M]$, the TD fixed point for average reward setting $\mathbf{w}_{\boldsymbol{\theta}}^{i,*}$ is uniformly bounded, specifically, there exists constant $R_{\mathbf{w}} = 4r_{\max}/\lambda_A > 0$ such that*

$$\|\mathbf{w}_{\boldsymbol{\theta}}^{i,*}\| \leq R_{\mathbf{w}}, \forall i \in [M].$$

*Proof.*
$$\|\mathbf{w}_{\boldsymbol{\theta}}^{i,*}\|_2 = \| - A_{\pi_{\boldsymbol{\theta}}}^{-1}\mathbf{b}_{\pi_{\boldsymbol{\theta}}}^i\|_2$$

$$= \| - \mathbb{E}_{s \sim d_{\boldsymbol{\theta}}(s), s' \sim P(\cdot|s)}[(\phi(s') - \phi(s))\phi^T(s)]^{-1} \cdot \mathbb{E}_{s \sim d_{\boldsymbol{\theta}}, a \sim \pi_{\boldsymbol{\theta}}}\left[\phi(s)\left(r^i(s,a) - J^i(\boldsymbol{\theta})\right)\right]\|_2$$

$$\leq \| - \mathbb{E}_{s \sim d_{\boldsymbol{\theta}}(s), s' \sim P(\cdot|s)}[(\phi(s') - \phi(s))\phi^T(s)]^{-1}\|_2 \cdot \|\mathbb{E}_{s \sim d_{\boldsymbol{\theta}}, a \sim \pi_{\boldsymbol{\theta}}}\left[\phi(s)\left(r^i(s,a) - J^i(\boldsymbol{\theta})\right)\right]\|_2$$

$$\overset{\text{(i)}}{=} \frac{\|\mathbb{E}_{s \sim d_{\boldsymbol{\theta}}, a \sim \pi_{\boldsymbol{\theta}}}\left[\phi(s)\left(r^i(s,a) - J^i(\boldsymbol{\theta})\right)\right]\|_2}{\sigma_{\min}\left(\| - \mathbb{E}_{s \sim d_{\boldsymbol{\theta}}(s), s' \sim P(\cdot|s)}[(\phi(s') - \phi(s))\phi^T(s)]\|_2\right)}$$

$$\overset{\text{(ii)}}{\leq} \frac{2\|\mathbb{E}_{s \sim d_{\boldsymbol{\theta}}, a \sim \pi_{\boldsymbol{\theta}}}\left[\phi(s)\left(r^i(s,a) - J^i(\boldsymbol{\theta})\right)\right]\|_2}{\lambda_A\left(-A_{\pi_{\boldsymbol{\theta}}} - A_{\pi_{\boldsymbol{\theta}}}^{\top}\right)}$$

$$\leq \frac{2 \cdot \mathbb{E}_{s \sim d_{\boldsymbol{\theta}}, a \sim \pi_{\boldsymbol{\theta}}}\left[\|\phi(s)\|_2 \cdot \left(|r^i(s,a)| + |J^i(\boldsymbol{\theta})|\right)\right]}{\lambda_A}$$

$$= \frac{4r_{\max}}{\lambda_A},$$

where (i) follows from the fact $\|A^{-1}\| = 1/\sigma_{\min}(A)$, and (ii) follows from Bhatia (2013) (Proposition III 5.1). $\square$

**Lemma 7** (Discounted reward setting). *Given a policy $\pi_{\boldsymbol{\theta}}$, for any objective $i \in [M]$, the value function approximation parameter $\mathbf{w}_{\boldsymbol{\theta}}^{i,*}$ is uniformly bounded, specifically, there exists constant $R_{\mathbf{w}} = 2r_{\max}/\lambda_A > 0$ such that*

$$\|\mathbf{w}_{\boldsymbol{\theta}}^{i,*}\| \leq R_{\mathbf{w}}, \forall i \in [M].$$

*Proof.*
$$\|\mathbf{w}_{\boldsymbol{\theta}}^{i,*}\|_2 = \| - A_{\pi_{\boldsymbol{\theta}}}^{-1}\mathbf{b}_{\pi_{\boldsymbol{\theta}}}^i\|_2$$

$$= \| - \mathbb{E}_{s \sim d_{\boldsymbol{\theta}}(s), s' \sim P(\cdot|s)}\left[(\gamma\phi(s') - \phi(s))\phi^T(s)\right]^{-1} \cdot \mathbb{E}_{s \sim d_{\boldsymbol{\theta}}, a \sim \pi_{\boldsymbol{\theta}}}\left[r^i(s,a)\phi(s)\right]\|_2$$

$$\leq \| - \mathbb{E}_{s \sim d_{\boldsymbol{\theta}}(s), s' \sim P(\cdot|s)}\left[(\gamma\phi(s') - \phi(s))\phi^T(s)\right]^{-1}\|_2 \cdot \|\mathbb{E}_{s \sim d_{\boldsymbol{\theta}}, a \sim \pi_{\boldsymbol{\theta}}}\left[r^i(s,a)\phi(s)\right]\|_2$$

$$= \frac{\|\mathbb{E}_{s \sim d_{\boldsymbol{\theta}}, a \sim \pi_{\boldsymbol{\theta}}}\left[r^i(s,a)\phi(s)\right]\|_2}{\| - \mathbb{E}_{s \sim d_{\boldsymbol{\theta}}(s), s' \sim P(\cdot|s)}\left[(\gamma\phi(s') - \phi(s))\phi^T(s)\right]\|_2}$$

$$\leq \frac{2\|\mathbb{E}_{s \sim d_{\boldsymbol{\theta}}, a \sim \pi_{\boldsymbol{\theta}}}\left[r^i(s,a)\phi(s)\right]\|_2}{\lambda_A\left(-A_{\pi_{\boldsymbol{\theta}}} - A_{\pi_{\boldsymbol{\theta}}}^{\top}\right)}$$

$$\leq \frac{2 \cdot \mathbb{E}_{s \sim d_{\boldsymbol{\theta}}, a \sim \pi_{\boldsymbol{\theta}}}\left[\|\phi(s)\|_2 \cdot |r^i(s,a)|\right]}{\lambda_A}$$

$$= \frac{2r_{\max}}{\lambda_A}.$$

$\square$

**Lemma 8.** *(Hairi et al. (2022) Lemma 2) Let $\nu_{\boldsymbol{\theta}}$ denote the stationary distribution of the state-action pairs given policy $\pi_{\boldsymbol{\theta}}$, there exists constants $\kappa > 0$ and $\rho \in (0,1)$ such that*

$$\sup_{s \in \mathcal{S}} \| P(s_t, a_t \mid s_0 = s) - \nu_{\boldsymbol{\theta}} \|_{TV} \leq \kappa \rho^t.$$

**Lemma 9.** *(Hairi et al. (2022) Lemma 3) Suppose Assumption 2 holds. Given a policy $\pi_{\boldsymbol{\theta}}$, we have the following:*

$$(-\mathbf{w}_{\boldsymbol{\theta}}^{i,*})^\top \mathbf{A}_{\pi_{\boldsymbol{\theta}}} (-\mathbf{w}_{\boldsymbol{\theta}}^{i,*}) \leq -\frac{\lambda_{\mathbf{A}}}{2} \| \mathbf{w}_{\boldsymbol{\theta}}^{i,*} \|_2^2.$$

**Lemma 10.** *(Xu et al. (2020) Theorem 4) For any $i \in [M]$, consider mini-batch linear stochastic approximation on $\mathbf{A}_{\pi_{\boldsymbol{\theta}}}$, $\mathbf{b}_{\boldsymbol{\theta}}^{\prime i}$ (discounted setting), and $\mathbf{b}_{\boldsymbol{\theta}}^i$ (average setting). Let $C_{\mathbf{A}} > \| \mathbf{A}_{\pi_{\boldsymbol{\theta}}} \|_F$ and $C_{\mathbf{b}}$ denote the upper bound for $\| \mathbf{b}_{\boldsymbol{\theta}}^i \|_2$ and $\| \mathbf{b}_{\boldsymbol{\theta}}^{\prime i} \|_2$, then by setting $\beta \leq \min\{\frac{\lambda_{\mathbf{A}}}{8C_{\mathbf{A}}^2}, \frac{4}{\lambda_{\mathbf{A}}}\}$ and $D \geq \left( \frac{2}{\lambda_{\mathbf{A}}} + 2\beta \right) \frac{192 C_{\mathbf{A}}^2 [1 + \rho(\kappa - 1)]}{(1-\rho)\lambda_{\mathbf{A}}}$ and we have*

$$\mathbb{E}\big[\| \mathbf{w}_N^i - \mathbf{w}_{\boldsymbol{\theta}}^{i,*} \|_2^2\big] \leq \left( 1 - \frac{\beta \lambda_{\mathbf{A}}}{8} \right)^N \cdot \| \mathbf{w}_0^i - \mathbf{w}_{\boldsymbol{\theta}}^{i,*} \|_2^2 + \left( \frac{2}{\lambda_{\mathbf{A}}} + 2\beta \right) \frac{192 \left( C_{\mathbf{A}}^2 R_{\mathbf{w}}^2 + C_{\mathbf{b}}^2 \right) [1 + \rho(\kappa - 1)]}{(1-\rho)\lambda_{\mathbf{A}} D}.$$

*Further, setting $N \geq \frac{8}{\beta \lambda_{\mathbf{A}}} \log \left( 2\| \mathbf{w}_0^i - \mathbf{w}_{\boldsymbol{\theta}}^{i,*} \|_2^2 / \epsilon \right)$ and $D \geq \left( \frac{2}{\lambda_{\mathbf{A}}} + 2\beta \right) \frac{192 \left( C_{\mathbf{A}}^2 R_{\mathbf{w}}^2 + C_{\mathbf{b}}^2 \right) [1 + \rho(\kappa - 1)]}{\epsilon(1-\rho)\lambda_{\mathbf{A}}}$, we have $\mathbb{E}\big[\| \mathbf{w}_N^i - \mathbf{w}_{\boldsymbol{\theta}}^{i,*} \|_2^2\big] \leq \epsilon$ with total sample complexity $ND = \mathcal{O}\left( \epsilon^{-1} \log\left( \epsilon^{-1} \right) \right)$.*

### B.3 Theoretical Results of the Critic of WC-MOAC

The critic component of WC-MOAC outputs $M$ value function approximation parameters based on the same sequences of Markovian samplings. In the average reward setting, given a policy parameter $\boldsymbol{\theta}$, define vector $\mathbf{b}_{\boldsymbol{\theta}}^i := \mathbb{E}_{s \sim d_{\boldsymbol{\theta}}, a \sim \pi_{\boldsymbol{\theta}}} \left[ \left( r^i(s,a) - J^i(\boldsymbol{\theta}) \right) \phi(s) \right], \forall i \in [M]$. Then the fixed point of TD-learning for objective $i$ is $\mathbf{w}_{\boldsymbol{\theta}}^{i,*} = -\mathbf{A}_{\pi_{\boldsymbol{\theta}}}^{-1} \mathbf{b}_{\boldsymbol{\theta}}^i$, where $\mathbf{A}_{\pi_{\boldsymbol{\theta}}}$ is defined in Assumption 2(d). Similarly, in the discounted reward setting, define vector $\mathbf{b}_{\boldsymbol{\theta}}^{\prime i} := \mathbb{E}_{s \sim d_{\boldsymbol{\theta}}, a \sim \pi_{\boldsymbol{\theta}}} \left[ r^i(s,a) \phi(s) \right]$ and we have $\mathbf{w}_{\boldsymbol{\theta}}^{i,*} = -\mathbf{A}_{\pi_{\boldsymbol{\theta}}}^{-1} \mathbf{b}_{\boldsymbol{\theta}}^{\prime i}, \forall i \in [M]$. Let constant $C_{\mathbf{A}} > \| \mathbf{A}_{\pi_{\boldsymbol{\theta}}} \|_F$, where $\| \cdot \|_F$ denotes the Frobenius Norm. We now state the convergence of the critic step of WC-MOAC as follows:

**Theorem 11.** Under Assumptions 1-3, for both average and discounted settings, let the critic step size $\beta \leq \min\{\frac{\lambda_{\mathbf{A}}}{8C_{\mathbf{A}}^2}, \frac{4}{\lambda_{\mathbf{A}}}\}$. Then, for any objective $i \in [M]$, the iterations generated by Algorithm 1 satisfy the following finite-time convergence error bound:

$$\mathbb{E}\big[\| \mathbf{w}_N^i - \mathbf{w}_{\boldsymbol{\theta}}^{i,*} \|_2^2\big] \leq C_1 \left( 1 - \frac{\beta \lambda_{\mathbf{A}}}{8} \right)^N + \frac{C_2 C_3 \left( \frac{2}{\lambda_{\mathbf{A}}} + 2\beta \right)}{\lambda_{\mathbf{A}} D}, \tag{16}$$

where $C_1 = \| \mathbf{w}_0^i - \mathbf{w}_{\boldsymbol{\theta}}^{i,*} \|_2^2$, $C_2 = [1 + (\kappa - 1)\rho]/(1 - \rho)$, and $C_3 > 0$ is a constant depending on $\mathbf{A}_{\pi_{\boldsymbol{\theta}}}$, $\mathbf{b}_{\boldsymbol{\theta}}^i$, and $\mathbf{b}_{\boldsymbol{\theta}}^{\prime i}$.

*Proof.* The results of Theorem 11 follows directly from Lemma 10, by setting $\mathbf{A}_{\pi_{\boldsymbol{\theta}}} := \mathbb{E}_{s \sim d_{\boldsymbol{\theta}}(s), s' \sim P(\cdot|s)}[(\phi(s') - \phi(s))\phi^\top(s)]$ and $\mathbf{b}_{\boldsymbol{\theta}}^i := \mathbb{E}_{s \sim d_{\boldsymbol{\theta}}, a \sim \pi_{\boldsymbol{\theta}}} \left[ \left( r^i(s,a) - J^i(\boldsymbol{\theta}) \right) \phi(s) \right], \forall i \in [M]$ for the average reward setting, and by setting $\mathbf{A}_{\pi_{\boldsymbol{\theta}}} := \mathbb{E}_{s \sim d_{\boldsymbol{\theta}}(s), s' \sim P(\cdot|s)} \left[ (\gamma \phi(s') - \phi(s)) \phi^T(s) \right]$ and $\mathbf{b}_{\boldsymbol{\theta}}^{\prime i} := \mathbb{E}_{s \sim d_{\boldsymbol{\theta}}, a \sim \pi_{\boldsymbol{\theta}}} \left[ r^i(s,a) \phi(s) \right], \forall i \in [M]$ for the discounted reward setting.

For clarity, we present Theorem 11 with some terms simplified as constants, where $C_1 = \| \mathbf{w}_0^i - \mathbf{w}_{\boldsymbol{\theta}}^{i,*} \|_2^2$, $C_2 = [1 + (\kappa - 1)\rho]/(1 - \rho)$, and $C_3 = 192 \left( C_{\mathbf{A}}^2 R_{\mathbf{w}}^2 + C_{\mathbf{b}}^2 \right)$. $\square$

Theorem 11 states that critic component of Algorithm 1 will evaluate and maintain a value function parameter $w_{\boldsymbol{\theta}}^i$ each objective $i \in [M]$ for the given policy $\pi_{\boldsymbol{\theta}}$. Compared to many existing works

Lakshminarayanan & Szepesvari (2018); Doan et al. (2018); Zhang et al. (2021) in RL algorithm finite-time convergence analysis, the samples in our method are correlated (i.e., Markovian noise) instead of i.i.d. noise, which is equivalent to $\rho = 0$. Despite the fact that Markovian noise introduces extra bias error seen from term $C_2$, our batching approach with size $D > 1$ offer two-fold benefits: 1) Part of the convergence error can be controlled with increasing $D$ (cf. the second term on the RHS in Eq. (16); 2) it allows the use of *constant* step size, leading to a better sample complexity comparing to non-batch approach Srikant & Ying (2019); Qiu et al. (2021); Hairi et al. (2024) and faster convergence in practice in general.

Theorem 11 immediately implies the following sample complexity results for the critic component in WC-MOAC:

**Corollary 12.** *For both average and discounted settings, let $N \geq \frac{8}{\beta \lambda_{\mathbf{A}}} \log(2C_1/\epsilon)$ and $D \geq C_2 C_3 \left( \frac{2}{\lambda_{\mathbf{A}}} + 2\beta \right)/(\epsilon \lambda_{\mathbf{A}})$. It then holds that $\mathbb{E}\left[ \|\mathbf{w}_N^i - \mathbf{w}_{\boldsymbol{\theta}}^{i,*}\|_2^2 \right] \leq \epsilon, i \in [M]$, which implies a sample complexity of $\mathcal{O}(\epsilon^{-1} \log(\epsilon^{-1}))$.*

## C  PROOF OF THEOREM 4

We first present the proof in average reward setting, then we show how to obtain the results in discounted reward setting.

*Proof.* For any given $\boldsymbol{\theta}$ and its associated policy $\pi_{\boldsymbol{\theta}}$, we denote the gradient matrix to be

$$\nabla_{\boldsymbol{\theta}} \mathbf{J}(\boldsymbol{\theta}) = \begin{bmatrix} \nabla_{\boldsymbol{\theta}} J^1(\boldsymbol{\theta}) & \nabla_{\boldsymbol{\theta}} J^2(\boldsymbol{\theta}) & \cdots & \nabla_{\boldsymbol{\theta}} J^M(\boldsymbol{\theta}) \end{bmatrix} \in \mathbb{R}^{d_1 \times M}.$$

Given $\boldsymbol{\theta} \in \mathbb{R}^{d_1}$, $\mathbf{w} \in \mathbb{R}^{d_2}$, for $t \geq 0$ and for any $i \in [M]$, by Lipschitzness in Assumption 3, we have

$$J^i(\boldsymbol{\theta}_{t+1}) \geq J^i(\boldsymbol{\theta}_t) + \langle \nabla_{\boldsymbol{\theta}} J^i(\boldsymbol{\theta}_t), \boldsymbol{\theta}_{t+1} - \boldsymbol{\theta}_t \rangle - \frac{L_J}{2} \|\boldsymbol{\theta}_{t+1} - \boldsymbol{\theta}_t\|^2. \tag{17}$$

Note that $J^i(\boldsymbol{\theta})$ is an expected value taken, where the expectation is taken over steady-state distribution induced by policy $\pi_{\boldsymbol{\theta}}$. We use $\boldsymbol{\lambda}_t^*$ to denote solution for $\boldsymbol{\lambda} \geq \mathbf{0}, \mathbf{1}^\top \boldsymbol{\lambda} = 1$, such that $\min_{\boldsymbol{\lambda}} \|\nabla_{\boldsymbol{\theta}} \mathbf{J}(\boldsymbol{\theta}_t) \boldsymbol{\lambda}\|_2$. In comparison, $\boldsymbol{\lambda}_t$ is the QP solution with momentum in Equation (11) for using $\{\mathbf{g}_t^i\}_{i \in [M]}$ as in Algorithm 1.

Let $\boldsymbol{q}_t := \frac{\boldsymbol{\lambda}_t \odot \mathbf{p}}{\langle \boldsymbol{\lambda}_t, \mathbf{p} \rangle}$, $l_t := \langle \boldsymbol{\lambda}_t, \mathbf{p} \rangle$ and $p_{\min} := \min_{i \in [M]} \mathbf{p}_i$. Note that $p_{\min} \leq l_t \leq 1$. For $t > 0$, $\boldsymbol{q}_t$ serves as a pseudo-weight for the actor convergence analysis and $l_t$ measures the length of it.

Taking $\boldsymbol{q}_t$ weighted summation over Eq. (17), we have

$$\boldsymbol{q}_t^\top \boldsymbol{J}(\boldsymbol{\theta}_{t+1}) \geq \boldsymbol{q}_t^\top \boldsymbol{J}(\boldsymbol{\theta}_t) + \langle \nabla_{\boldsymbol{\theta}} \boldsymbol{J}(\boldsymbol{\theta}_t) \boldsymbol{q}_t, \boldsymbol{\theta}_{t+1} - \boldsymbol{\theta}_t \rangle - \frac{L_J}{2} \|\boldsymbol{\theta}_{t+1} - \boldsymbol{\theta}_t\|_2^2$$

$$= \boldsymbol{q}_t^\top \boldsymbol{J}(\boldsymbol{\theta}_t) + \alpha l_t \left\langle \nabla_{\boldsymbol{\theta}} \boldsymbol{J}(\boldsymbol{\theta}_t) \boldsymbol{q}_t, \sum_{j=1}^{M} q_t^j \mathbf{g}_t^j \right\rangle - \frac{\alpha^2 L_J}{2} \|\mathbf{g}_t\|_2^2$$

$$= \boldsymbol{q}_t^\top \boldsymbol{J}(\boldsymbol{\theta}_t) + \alpha l_t \left\langle \nabla_{\boldsymbol{\theta}} \boldsymbol{J}(\boldsymbol{\theta}_t) \boldsymbol{q}_t, \sum_{j=1}^{M} q_t^j \cdot \left( \mathbf{g}_t^j - \nabla_{\boldsymbol{\theta}} J^j(\boldsymbol{\theta}_t) + \nabla_{\boldsymbol{\theta}} J^j(\boldsymbol{\theta}_t) \right) \right\rangle - \frac{\alpha^2 L_J}{2} \|\mathbf{g}_t\|_2^2$$

$$= \boldsymbol{q}_t^\top \boldsymbol{J}(\boldsymbol{\theta}_t) + \alpha l_t \left\langle \nabla_{\boldsymbol{\theta}} \boldsymbol{J}(\boldsymbol{\theta}_t) \boldsymbol{q}_t, \sum_{j=1}^{M} q_t^j \nabla_{\boldsymbol{\theta}} J^j(\boldsymbol{\theta}_t) \right\rangle$$

$$+ \alpha l_t \left\langle \nabla_{\boldsymbol{\theta}} \boldsymbol{J}(\boldsymbol{\theta}_t) \boldsymbol{q}_t, \sum_{j=1}^{M} q_t^j \cdot \left( \mathbf{g}_t^j - \nabla_{\boldsymbol{\theta}} J^j(\boldsymbol{\theta}_t) \right) \right\rangle - \frac{\alpha^2 L_J}{2} \|\mathbf{g}_t\|_2^2$$

$$= \boldsymbol{q}_t^\top \boldsymbol{J}(\boldsymbol{\theta}_t) + \alpha l_t \|\nabla_{\boldsymbol{\theta}} \boldsymbol{J}(\boldsymbol{\theta}_t) \boldsymbol{q}_t\|_2^2 + \alpha l_t \left\langle \nabla_{\boldsymbol{\theta}} \boldsymbol{J}(\boldsymbol{\theta}_t) \boldsymbol{q}_t, \sum_{j=1}^{M} q_t^j \cdot \left( \mathbf{g}_t^j - \nabla_{\boldsymbol{\theta}} J^j(\boldsymbol{\theta}_t) \right) \right\rangle - \frac{\alpha^2 L_J}{2} \|\mathbf{g}_t\|_2^2$$

$$\overset{(i)}{\geq} \boldsymbol{q}_t^\top \boldsymbol{J}(\boldsymbol{\theta}_t) + \frac{\alpha l_t}{2} \|\nabla_{\boldsymbol{\theta}} \boldsymbol{J}(\boldsymbol{\theta}_t) \boldsymbol{q}_t\|_2^2 - \frac{\alpha l_t}{2} \left\| \sum_{j=1}^{M} q_t^j \cdot \left( \nabla_{\boldsymbol{\theta}} J^j(\boldsymbol{\theta}_t) - \mathbf{g}_t^j \right) \right\|_2^2 - \frac{\alpha^2 L_J}{2} \|\mathbf{g}_t\|_2^2$$

$$
= \boldsymbol{q}_t^\top \boldsymbol{J}(\boldsymbol{\theta}_t) + \frac{\alpha l_t}{2} \|\nabla_{\boldsymbol{\theta}} \boldsymbol{J}(\boldsymbol{\theta}_t) \boldsymbol{q}_t\|_2^2 - \frac{\alpha l_t}{2} \left\| \sum_{j=1}^M q_t^j \cdot \left( \nabla_{\boldsymbol{\theta}} J^j(\boldsymbol{\theta}_t) - \mathbf{g}_t^j \right) \right\|_2^2
$$

$$
- \frac{\alpha^2 l_t^2 L_J}{2} \left\| \sum_{j=1}^M q_t^j \cdot \left( \mathbf{g}_t^j - \nabla_{\boldsymbol{\theta}} J^j(\boldsymbol{\theta}_t) + \nabla_{\boldsymbol{\theta}} J^j(\boldsymbol{\theta}_t) \right) \right\|_2^2
$$

$$
\overset{\text{(ii)}}{\geq} \boldsymbol{q}_t^\top \boldsymbol{J}(\boldsymbol{\theta}_t) + \left( \frac{\alpha l_t}{2} - \alpha^2 l_t^2 L_J \right) \|\nabla_{\boldsymbol{\theta}} \boldsymbol{J}(\boldsymbol{\theta}_t) \boldsymbol{q}_t\|_2^2 - \left( \frac{\alpha l_t}{2} + \alpha^2 l_t^2 L_J \right) \left\| \sum_{j=1}^M q_t^j \cdot \left( \nabla_{\boldsymbol{\theta}} J^j(\boldsymbol{\theta}_t) - \mathbf{g}_t^j \right) \right\|_2^2,
$$

$$
\tag{18}
$$

where inequality (i) follows from

$$
\left\langle \nabla_{\boldsymbol{\theta}} \boldsymbol{J}(\boldsymbol{\theta}_t) \boldsymbol{q}_t, \sum_{j=1}^M q_t^j \cdot \left( \mathbf{g}_t^j - \nabla_{\boldsymbol{\theta}} J^j(\boldsymbol{\theta}_t) \right) \right\rangle \geq -\frac{1}{2} \|\nabla_{\boldsymbol{\theta}} \boldsymbol{J}(\boldsymbol{\theta}_t) \boldsymbol{q}_t\|_2^2 - \frac{1}{2} \left\| \sum_{j=1}^M q_t^j \cdot \left( \nabla_{\boldsymbol{\theta}} J^j(\boldsymbol{\theta}_t) - \mathbf{g}_t^j \right) \right\|_2^2,
$$

and inequality (ii) follows from

$$
\left\| \sum_{j=1}^M q_t^j \cdot \left( \mathbf{g}_t^j - \nabla_{\boldsymbol{\theta}} J^j(\boldsymbol{\theta}_t) + \nabla_{\boldsymbol{\theta}} J^j(\boldsymbol{\theta}_t) \right) \right\|_2^2 \leq 2 \|\nabla_{\boldsymbol{\theta}} \boldsymbol{J}(\boldsymbol{\theta}_t) \boldsymbol{q}_t\|_2^2 + 2 \left\| \sum_{j=1}^M q_t^j \cdot \left( \nabla_{\boldsymbol{\theta}} J^j(\boldsymbol{\theta}_t) - \mathbf{g}_t^j \right) \right\|_2^2.
$$

Taking expectation on both sides of Eq. (18) and conditioning on $\mathcal{F}_t$, we have

$$
\mathbb{E} \left[ \|\nabla_{\boldsymbol{\theta}} \boldsymbol{J}(\boldsymbol{\theta}_t) \boldsymbol{q}_t\|_2^2 \mid \mathcal{F}_t \right] \leq \frac{2 \left( \mathbb{E} \left[ \boldsymbol{q}_t^\top \boldsymbol{J}(\boldsymbol{\theta}_{t+1}) | \mathcal{F}_t \right] - \boldsymbol{q}_t^\top \boldsymbol{J}(\boldsymbol{\theta}_t) \right)}{\alpha l_t - 2\alpha^2 l_t^2 L_J} + \frac{\alpha + 2\alpha^2 l_t L_J}{\alpha - 2\alpha^2 l_t L_J} \mathbb{E} \left[ \left\| \sum_{j=1}^M q_t^j \left( \nabla_{\boldsymbol{\theta}} J^j(\boldsymbol{\theta}_t) - \mathbf{g}_t^j \right) \right\|_2^2 \mid \mathcal{F}_t \right].
$$

By the definitions of $\boldsymbol{\lambda}_t^*$ and $\boldsymbol{q}_t$, for any time $t$, we have

$$
\mathbb{E} \left[ \|\nabla_{\boldsymbol{\theta}} \boldsymbol{J}(\boldsymbol{\theta}_t) \boldsymbol{\lambda}_t^*\|_2^2 \mid \mathcal{F}_t \right] \leq \mathbb{E} \left[ \|\nabla_{\boldsymbol{\theta}} \boldsymbol{J}(\boldsymbol{\theta}_t) \boldsymbol{q}_t\|_2^2 \mid \mathcal{F}_t \right].
$$

Therefore, we have

$$
\mathbb{E} \left[ \|\nabla_{\boldsymbol{\theta}} \boldsymbol{J}(\boldsymbol{\theta}_t) \boldsymbol{\lambda}_t^*\|_2^2 \mid \mathcal{F}_t \right] \leq \frac{2 \left( \mathbb{E} \left[ \boldsymbol{q}_t^\top \boldsymbol{J}(\boldsymbol{\theta}_{t+1}) | \mathcal{F}_t \right] - \boldsymbol{q}_t^\top \boldsymbol{J}(\boldsymbol{\theta}_t) \right)}{\alpha l_t - 2\alpha^2 l_t^2 L_J} + \frac{\alpha + 2\alpha^2 l_t L_J}{\alpha - 2\alpha^2 l_t L_J} \mathbb{E} \left[ \left\| \sum_{j=1}^M q_t^j \left( \nabla_{\boldsymbol{\theta}} J^j(\boldsymbol{\theta}_t) - \mathbf{g}_t^j \right) \right\|_2^2 \mid \mathcal{F}_t \right].
$$

$$
\tag{19}
$$

### C.1 For the 2nd term on RHS of Eq. (19)

Define a notation: $\Delta_{\boldsymbol{\theta}_t, \mathbf{w}_t^*}^j = \mathbb{E}_{d_{\boldsymbol{\theta}}} \left[ \mathbb{E}_{P_{\boldsymbol{\theta}}} \left[ \delta_{t,l}^j(\mathbf{w}_t^{j,*}) \mid (a_{t,l}, s_{t,l}) \right] \cdot \psi_{t,l}^{\boldsymbol{\theta}} \right]$. We first bound the last term on the right hand side of Eq. (19) as follows:

$$
\mathbb{E} \left[ \left\| \sum_{j=1}^M \lambda_t^j \left( \nabla_{\boldsymbol{\theta}} J^j(\boldsymbol{\theta}_t) - \mathbf{g}_t^j \right) \right\|_2^2 \Big| \mathcal{F}_t \right]
$$

$$
\leq \mathbb{E} \left[ \left( \sum_{j=1}^M \lambda_t^j \left\| \nabla_{\boldsymbol{\theta}} J^j(\boldsymbol{\theta}_t) - \mathbf{g}_t^j \right\|_2 \right)^2 \Big| \mathcal{F}_t \right]
$$

$$
\leq \mathbb{E} \left[ \left( \sum_{j=1}^M \lambda_t^j \left( \left\| \nabla_{\boldsymbol{\theta}} J^j(\boldsymbol{\theta}_t) - \Delta_{\boldsymbol{\theta}_t, \mathbf{w}_t^*}^j \right\|_2 + \left\| \Delta_{\boldsymbol{\theta}_t, \mathbf{w}_t^*}^j - \mathbf{g}_{\boldsymbol{\theta}_t^*}^j \right\|_2 + \left\| \mathbf{g}_{\boldsymbol{\theta}_t^*}^j - \mathbf{g}_t^j \right\|_2 \right) \right)^2 \Big| \mathcal{F}_t \right]
$$

$$\leq 3\mathbb{E}\left[\left(\sum_{j=1}^{M}\lambda_t^j\left\|\nabla_{\boldsymbol{\theta}}J^j(\boldsymbol{\theta}_t)-\Delta_{\boldsymbol{\theta}_t,\mathbf{w}_t^*}^j\right\|_2\right)^2\middle|\mathcal{F}_t\right]+3\mathbb{E}\left[\left(\sum_{j=1}^{M}\lambda_t^j\left\|\mathbf{g}_{\boldsymbol{\theta}_t^*}^j-\mathbf{g}_t^j\right\|_2\right)^2\middle|\mathcal{F}_t\right]$$

$$+3\mathbb{E}\left[\left(\sum_{j=1}^{M}\lambda_t^j\cdot\left\|\Delta_{\boldsymbol{\theta}_t,\mathbf{w}_t^*}^j-\mathbf{g}_{\boldsymbol{\theta}_t^*}^j\right\|_2\right)^2\middle|\mathcal{F}_t\right], \tag{20}$$

where

$$\left\|\nabla_{\boldsymbol{\theta}}J^j(\boldsymbol{\theta}_t)-\Delta_{\boldsymbol{\theta}_t,\mathbf{w}_t^*}^j\right\|_2^2=\left\|\mathbb{E}_{d_{\boldsymbol{\theta}}}\left[\mathbb{E}_{P_{\boldsymbol{\theta}}}\left[\delta_{t,l}^j\mid(a_{t,l},s_{t,l})\right]\cdot\boldsymbol{\psi}_{t,l}^{\boldsymbol{\theta}}\right]-\mathbb{E}_{d_{\boldsymbol{\theta}}}\left[\mathbb{E}_{P_{\boldsymbol{\theta}}}\left[\delta_{t,l}^j(\mathbf{w}_t^{j,*})\mid(a_{t,l},s_{t,l})\right]\cdot\boldsymbol{\psi}_{t,l}^{\boldsymbol{\theta}}\right]\right\|_2^2$$

$$=\left\|\mathbb{E}_{d_{\boldsymbol{\theta}}}\left[\left(\mathbb{E}_{P_{\boldsymbol{\theta}}}\left[\delta_{t,l}^j\mid(a_{t,l},s_{t,l})\right]-\mathbb{E}_{P_{\boldsymbol{\theta}}}\left[\delta_{t,l}^j(\mathbf{w}_t^{j,*})\mid(a_{t,l},s_{t,l})\right]\right)\cdot\boldsymbol{\psi}_{t,l}^{\boldsymbol{\theta}}\right]\right\|_2^2$$

$$\leq\mathbb{E}_{d_{\boldsymbol{\theta}}}\left[\left\|\left(\mathbb{E}_{P_{\boldsymbol{\theta}}}\left[\delta_{t,l}^j\mid(a_{t,l},s_{t,l})\right]-\mathbb{E}_{P_{\boldsymbol{\theta}}}\left[\delta_{t,l}^j(\mathbf{w}_t^{j,*})\mid(a_{t,l},s_{t,l})\right]\right)\cdot\boldsymbol{\psi}_{t,l}^{\boldsymbol{\theta}}\right\|_2^2\right]$$

$$\leq\mathbb{E}_{d_{\boldsymbol{\theta}}}\left[\left|\mathbb{E}_{P_{\boldsymbol{\theta}}}\left[\delta_{t,l}^j\mid(a_{t,l},s_{t,l})\right]-\mathbb{E}_{P_{\boldsymbol{\theta}}}\left[\delta_{t,l}^j(\mathbf{w}_t^{j,*})\mid(a_{t,l},s_{t,l})\right]\right|^2\right]$$

$$=\mathbb{E}_{d_{\boldsymbol{\theta}}}\left[\left|\mathbb{E}\left[V_{\boldsymbol{\theta}}^j(s_{t,l+1})-V_{\boldsymbol{\theta}}^j(s_{t,l+1};\mathbf{w}_t^{j,*})\mid(a_{t,l},s_{t,l})\right]+V_{\boldsymbol{\theta}}^j(s_{t,l})-V_{\boldsymbol{\theta}}^j(s_{t,l};\mathbf{w}_t^{j,*})\right|^2\right]$$

$$\leq 4\zeta_{\text{approx}}.$$

We note that $\delta_{t,l}^j$ denotes the TD error for objective $j\in[M]$ using the ground truth value functions. We also remark that the above inequality holds for all $j\in[M]$. As a result, for the first term on the RHS of Eq. (20), we have

$$\mathbb{E}\left[\left(\sum_{j=1}^{M}\lambda_t^j\left\|\nabla_{\boldsymbol{\theta}}J^j(\boldsymbol{\theta}_t)-\Delta_{\boldsymbol{\theta}_t,\mathbf{w}_t^*}^j\right\|_2\right)^2\middle|\mathcal{F}_t\right]\leq\mathbb{E}\left[\left(\sum_{j=1}^{M}\lambda_t^j2\sqrt{\zeta_{\text{approx}}}\right)^2\middle|\mathcal{F}_t\right]=4\zeta_{\text{approx}}$$

Furthermore, we have

$$\left\|\mathbf{g}_{\boldsymbol{\theta}_t^*}^j-\mathbf{g}_t^j\right\|_2=\left\|\frac{1}{B}\sum_{l=0}^{B-1}\left(\delta_{t,l}^j(\mathbf{w}_t^j)-\delta_{t,l}^j(\mathbf{w}_t^{j,*})\right)\cdot\boldsymbol{\psi}_{t,l}^{\boldsymbol{\theta}}\right\|_2$$

$$=\left\|\frac{1}{B}\sum_{l=0}^{B-1}(\boldsymbol{\phi}(s_{t,l+1})-\boldsymbol{\phi}(s_{t,l}))^{\top}\left(\mathbf{w}_t^j-\mathbf{w}_t^{j,*}\right)\cdot\boldsymbol{\psi}_{t,l}^{\boldsymbol{\theta}}\right\|_2$$

$$\leq\left\|\frac{1}{B}\sum_{l=0}^{B-1}(\boldsymbol{\phi}(s_{t,l+1})-\boldsymbol{\phi}(s_{t,l}))^{\top}\left(\mathbf{w}_t^j-\mathbf{w}_t^{j,*}\right)\right\|_2$$

$$\leq\max_{l\in\{0,\ldots,B-1\}}\left\|(\boldsymbol{\phi}(s_{t,l+1})-\boldsymbol{\phi}(s_{t,l}))^{\top}\left(\mathbf{w}_t^j-\mathbf{w}_t^{j,*}\right)\right\|_2$$

$$\leq 2\cdot\left\|\mathbf{w}_t^j-\mathbf{w}_t^{j,*}\right\|_2.$$

As a result, for the second term on the RHS of Eq. (20), we have

$$\mathbb{E}\left[\left(\sum_{j=1}^{M}\lambda_t^j\left\|\mathbf{g}_{\boldsymbol{\theta}_t^*}^j-\mathbf{g}_t^j\right\|_2\right)^2\middle|\mathcal{F}_t\right]\leq\mathbb{E}\left[\left(\sum_{j=1}^{M}\lambda_t^j2\left\|\mathbf{w}_t^j-\mathbf{w}_t^{j,*}\right\|_2\right)^2\middle|\mathcal{F}_t\right]\leq 4\max_{i\in[M]}\mathbb{E}\left[\left\|\mathbf{w}_t^i-\mathbf{w}_t^{i,*}\right\|_2^2\middle|\mathcal{F}_t\right]. \tag{21}$$

For the second inequality above, it holds because

$$\mathbb{E}\left[\left(\sum_{j=1}^{M}\lambda_t^j\left\|\mathbf{w}_t^j-\mathbf{w}_t^{j,*}\right\|_2\right)^2\middle|\mathcal{F}_t\right]$$

$$=\mathbb{E}\left[\sum_{j=1}^{M}(\lambda_t^j)^2\left\|\mathbf{w}_t^j-\mathbf{w}_t^{j,*}\right\|_2^2+2\sum_{i\neq j}\lambda_t^i\lambda_t^j\left\|\mathbf{w}_t^i-\mathbf{w}_t^{i,*}\right\|_2\cdot\left\|\mathbf{w}_t^j-\mathbf{w}_t^{j,*}\right\|_2\middle|\mathcal{F}_t\right]$$

$$=\sum_{j=1}^{M}(\lambda_t^j)^2\mathbb{E}\left[\left\|\mathbf{w}_t^j-\mathbf{w}_t^{j,*}\right\|_2^2\middle|\mathcal{F}_t\right]+2\sum_{i\neq j}\lambda_t^i\lambda_t^j\mathbb{E}\left[\left\|\mathbf{w}_t^i-\mathbf{w}_t^{i,*}\right\|_2\cdot\left\|\mathbf{w}_t^j-\mathbf{w}_t^{j,*}\right\|_2\middle|\mathcal{F}_t\right]$$

$$=\sum_{j=1}^{M}(\lambda_t^j)^2\mathbb{E}\left[\left\|\mathbf{w}_t^j-\mathbf{w}_t^{j,*}\right\|_2^2\middle|\mathcal{F}_t\right]+2\sum_{i\neq j}\lambda_t^i\lambda_t^j\mathbb{E}\left[\left\|\mathbf{w}_t^i-\mathbf{w}_t^{i,*}\right\|_2\middle|\mathcal{F}_t\right]\cdot\mathbb{E}\left[\left\|\mathbf{w}_t^j-\mathbf{w}_t^{j,*}\right\|_2\middle|\mathcal{F}_t\right]$$

$$\leq\sum_{j=1}^{M}(\lambda_t^j)^2\mathbb{E}\left[\left\|\mathbf{w}_t^j-\mathbf{w}_t^{j,*}\right\|_2^2\middle|\mathcal{F}_t\right]+2\sum_{i\neq j}\lambda_t^i\lambda_t^j\sqrt{\mathbb{E}\left[\left\|\mathbf{w}_t^i-\mathbf{w}_t^{i,*}\right\|_2^2\middle|\mathcal{F}_t\right]}\cdot\sqrt{\mathbb{E}\left[\left\|\mathbf{w}_t^j-\mathbf{w}_t^{j,*}\right\|_2^2\middle|\mathcal{F}_t\right]}$$

$$\leq\left(\sum_{j=1}^{M}(\lambda_t^j)^2+2\sum_{i\neq j}\lambda_t^i\lambda_t^j\right)\max_{i\in[M]}\mathbb{E}\left[\left\|\mathbf{w}_t^i-\mathbf{w}_t^{i,*}\right\|_2^2\middle|\mathcal{F}_t\right]$$

$$=(\sum_{j=1}^{M}\lambda_t^j)^2\max_{i\in[M]}\mathbb{E}\left[\left\|\mathbf{w}_t^i-\mathbf{w}_t^{i,*}\right\|_2^2\middle|\mathcal{F}_t\right]$$

$$=\max_{i\in[M]}\mathbb{E}\left[\left\|\mathbf{w}_t^i-\mathbf{w}_t^{i,*}\right\|_2^2\middle|\mathcal{F}_t\right],$$

where the third equality is due to the conditional independence of objective $i$ and $j$ given filtration $\mathcal{F}_t$ and the first inequality is because of $(\mathbb{E}[X])^2\leq\mathbb{E}[X^2]$ for a random variable $X$. Similarly, for the last term in Eq. (20), we have

$$\mathbb{E}\left[\left(\sum_{j=1}^{M}\lambda_t^j\cdot\left\|\Delta_{\boldsymbol{\theta}_t,\mathbf{w}_t^*}^j-\mathbf{g}_{\boldsymbol{\theta}_t^*}^j\right\|_2\right)^2\middle|\mathcal{F}_t\right]\leq\max_{i\in[M]}\mathbb{E}\left[\left(\sum_{j=1}^{M}\lambda_t^j\cdot\left\|\Delta_{\boldsymbol{\theta}_t,\mathbf{w}_t^*}^i-\mathbf{g}_{\boldsymbol{\theta}_t^*}^i\right\|_2\right)^2\middle|\mathcal{F}_t\right]=\max_{i\in[M]}\mathbb{E}\left[\left\|\Delta_{\boldsymbol{\theta}_t,\mathbf{w}_t^*}^i-\mathbf{g}_{\boldsymbol{\theta}_t^*}^i\right\|_2^2\middle|\mathcal{F}_t\right].$$

In addition, for any $j\in[M]$, we have

$$\mathbb{E}\left[\left\|\Delta_{\boldsymbol{\theta}_t,\mathbf{w}_t^*}^j-\mathbf{g}_{\boldsymbol{\theta}_t^*}^j\right\|_2^2\middle|\mathcal{F}_t\right]$$

$$=\mathbb{E}\left[\left\|\frac{1}{B}\sum_{l=0}^{B-1}\delta_{t,l}^j(\mathbf{w}_t^{j,*})\cdot\boldsymbol{\psi}_{t,l}^{\boldsymbol{\theta}}-\Delta_{\boldsymbol{\theta}_t,\mathbf{w}_t^*}^j\right\|_2^2\middle|\mathcal{F}_t\right]$$

$$=\mathbb{E}\left[\left\langle\frac{1}{B}\sum_{l_1=0}^{B-1}\delta_{t,l_1}^j(\mathbf{w}_t^{j,*})\cdot\boldsymbol{\psi}_{t,l_1}^{\boldsymbol{\theta}}-\Delta_{\boldsymbol{\theta}_t,\mathbf{w}_t^*}^j,\frac{1}{B}\sum_{l_2=0}^{B-1}\delta_{t,l_2}^j(\mathbf{w}_t^{j,*})\cdot\boldsymbol{\psi}_{t,l_2}^{\boldsymbol{\theta}}-\Delta_{\boldsymbol{\theta}_t,\mathbf{w}_t^*}^j\right\rangle\middle|\mathcal{F}_t\right]$$

$$=\mathbb{E}\left[\frac{1}{B^2}\sum_{l=0}^{B-1}\left\|\delta_{t,l}^j(\mathbf{w}_t^{j,*})\boldsymbol{\psi}_{t,l}^{\boldsymbol{\theta}}-\Delta_{\boldsymbol{\theta}_t,\mathbf{w}_t^*}^j\right\|_2^2+\frac{1}{B^2}\sum_{l_1\neq l_2}\left\langle\delta_{t,l_1}^j(\mathbf{w}_t^{j,*})\cdot\boldsymbol{\psi}_{t,l_1}^{\boldsymbol{\theta}}-\Delta_{\boldsymbol{\theta}_t,\mathbf{w}_t^*}^j,\delta_{t,l_2}^j(\mathbf{w}_t^{j,*})\cdot\boldsymbol{\psi}_{t,l_2}^{\boldsymbol{\theta}}-\Delta_{\boldsymbol{\theta}_t,\mathbf{w}_t^*}^j\right\rangle\middle|\mathcal{F}_t\right]$$

$$\overset{(i)}{\leq}\frac{16}{B}(r_{\max}+R_{\mathbf{w}})^2+\frac{1}{B^2}\sum_{l_1\neq l_2}\mathbb{E}\left[\left\langle\delta_{t,l_1}^j(\mathbf{w}_t^{j,*})\cdot\boldsymbol{\psi}_{t,l_1}^{\boldsymbol{\theta}}-\Delta_{\boldsymbol{\theta}_t,\mathbf{w}_t^*}^j,\delta_{t,l_2}^j(\mathbf{w}_t^{j,*})\cdot\boldsymbol{\psi}_{t,l_2}^{\boldsymbol{\theta}}-\Delta_{\boldsymbol{\theta}_t,\mathbf{w}_t^*}^j\right\rangle\middle|\mathcal{F}_t\right]$$

$$=\frac{16}{B}(r_{\max}+R_{\mathbf{w}})^2+\frac{2}{B^2}\sum_{l_1<l_2}\mathbb{E}\left[\left\langle\delta_{t,l_1}^j(\mathbf{w}_t^{j,*})\cdot\boldsymbol{\psi}_{t,l_1}^{\boldsymbol{\theta}}-\Delta_{\boldsymbol{\theta}_t,\mathbf{w}_t^*}^j,\delta_{t,l_2}^j(\mathbf{w}_t^{j,*})\cdot\boldsymbol{\psi}_{t,l_2}^{\boldsymbol{\theta}}-\Delta_{\boldsymbol{\theta}_t,\mathbf{w}_t^*}^j\right\rangle\middle|\mathcal{F}_t\right]$$

$$=\frac{16}{B}(r_{\max}+R_{\mathbf{w}})^2+\frac{2}{B^2}\sum_{l_1<l_2}\mathbb{E}\left[\left\langle\delta_{t,l_1}^j(\mathbf{w}_t^{j,*})\cdot\boldsymbol{\psi}_{t,l_1}^{\boldsymbol{\theta}}-\Delta_{\boldsymbol{\theta}_t,\mathbf{w}_t^*}^j,\mathbb{E}\left[\delta_{t,l_2}^j(\mathbf{w}_t^{j,*})\cdot\boldsymbol{\psi}_{t,l_2}^{\boldsymbol{\theta}}\middle|\mathcal{F}_{t,l_1}\right]-\Delta_{\boldsymbol{\theta}_t,\mathbf{w}_t^*}^j\right\rangle\middle|\mathcal{F}_t\right]$$

$$\leq\frac{16}{B}(r_{\max}+R_{\mathbf{w}})^2+\frac{2}{B^2}\sum_{l_1<l_2}\mathbb{E}\left[\left\|\delta_{t,l_1}^j(\mathbf{w}_t^{j,*})\cdot\boldsymbol{\psi}_{t,l_1}^{\boldsymbol{\theta}}-\Delta_{\boldsymbol{\theta}_t,\mathbf{w}_t^*}^j\right\|_2\cdot\left\|\mathbb{E}\left[\delta_{t,l_2}^j(\mathbf{w}_t^{j,*})\cdot\boldsymbol{\psi}_{t,l_2}^{\boldsymbol{\theta}}\middle|\mathcal{F}_{t,l_1}\right]-\Delta_{\boldsymbol{\theta}_t,\mathbf{w}_t^*}^j\right\|_2\middle|\mathcal{F}_t\right]$$

$$\leq\frac{16}{B}(r_{\max}+R_{\mathbf{w}})^2+\frac{2}{B^2}\sum_{l_1<l_2}4\left(r_{\max}+R_{\mathbf{w}}\right)\mathbb{E}\left[\left\|\mathbb{E}\left[\delta_{t,l_2}^j(\mathbf{w}_t^{j,*})\cdot\boldsymbol{\psi}_{t,l_2}^{\boldsymbol{\theta}}\middle|\mathcal{F}_{t,l_1}\right]-\Delta_{\boldsymbol{\theta}_t,\mathbf{w}_t^*}^j\right\|_2\middle|\mathcal{F}_t\right]$$

$$\overset{(ii)}{\leq} \frac{16}{B}(r_{\max} + R_{\mathbf{w}})^2 + \frac{2}{B^2}\sum_{l_1 < l_2} 16(r_{\max} + R_{\mathbf{w}})^2 \kappa \rho^{l_2 - l_1},$$

where (i) follows from the facts that

$$|\delta_{t,l}^j(\mathbf{w}_t^{j,*})| = |r_{t,l+1}^j - \mu_{t,l}^j + \boldsymbol{\phi}(s_{t,l+1})^\top \mathbf{w}_t^j - \boldsymbol{\phi}(s_{t,l})^\top \mathbf{w}_t^j|_1$$
$$\leq |r_{t,l+1}^j| + |\mu_{t,l}^j| + \|\boldsymbol{\phi}(s_{t,l+1}) - \boldsymbol{\phi}(s_{t,l})\|_2 \cdot \|\mathbf{w}_t^j\|_2$$
$$\leq 2r_{\max} + 2R_{\mathbf{w}},$$

thus, $\|\delta_{t,l}^j(\mathbf{w}_t^{j,*})\boldsymbol{\psi}_{t,l}^{\boldsymbol{\theta}}\|_2 \leq 2r_{\max} + 2R_{\mathbf{w}}$, and $\Delta_{\boldsymbol{\theta}_t, \mathbf{w}_t^*}^j = \mathbb{E}_{d_{\boldsymbol{\theta}}}\left[\mathbb{E}_{P_{\boldsymbol{\theta}}}\left[\delta_{t,l}^j(\mathbf{w}_t^{j,*}) \mid (a_{t,l}, s_{t,l})\right] \cdot \boldsymbol{\psi}_{t,l}^{\boldsymbol{\theta}}\right] \leq 2r_{\max} + 2R_{\mathbf{w}}$, and (ii) follows from

$$\left\|\mathbb{E}\left[\delta_{t,l_2}^j(\mathbf{w}_t^{j,*}) \cdot \boldsymbol{\psi}_{t,l_2}^{\boldsymbol{\theta}} \big| \mathcal{F}_{t,l_1}\right] - \Delta_{\boldsymbol{\theta}_t, \mathbf{w}_t^*}^j\right\|_2$$

$$= \left\|\mathbb{E}\left[\delta_{t,l_2}^j(\mathbf{w}_t^{j,*}) \cdot \boldsymbol{\psi}_{t,l_2}^{\boldsymbol{\theta}} \big| \mathcal{F}_{t,l_1}\right] - \mathbb{E}_{d_{\boldsymbol{\theta}}}\left[\mathbb{E}_{P_{\boldsymbol{\theta}}}\left[\delta_{t,l}^j(\mathbf{w}_t^{j,*}) \mid (s_{t,l}, a_{t,l})\right] \cdot \boldsymbol{\psi}_{t,l}^{\boldsymbol{\theta}}\right]\right\|_2$$

$$= \left\|\sum_{(s_{t,l_2}, a_{t,l_2})} \mathbb{E}_{P_{\boldsymbol{\theta}}}\left[\delta_{t,l_2}^j(\mathbf{w}_t^{j,*}) \mid (s_{t,l_2}, a_{t,l_2})\right] \cdot \boldsymbol{\psi}_{t,l}^{\boldsymbol{\theta}} \cdot P(s_{t,l_2}, a_{t,l_2} \mid \mathcal{F}_{t,l_1})\right.$$

$$\left. - \sum_{(s_{t,l}, a_{t,l})} \mathbb{E}_{P_{\boldsymbol{\theta}}}\left[\delta_{t,l}^j(\mathbf{w}_t^{j,*}) \mid (s_{t,l}, a_{t,l})\right] \cdot \boldsymbol{\psi}_{t,l}^{\boldsymbol{\theta}} \cdot \nu_{\boldsymbol{\theta}_t}(s_{t,l}, a_{t,l})\right\|_2$$

$$\leq \sum_{(s_{t,l}, a_{t,l})} \left\|\mathbb{E}_{P_{\boldsymbol{\theta}}}\left[\delta_{t,l}^j(\mathbf{w}_t^{j,*}) \mid (s_{t,l}, a_{t,l})\right] \cdot \boldsymbol{\psi}_{t,l}^{\boldsymbol{\theta}}\right\|_2 \cdot \left|P^{l_2 - l_1}(s_{t,l}, a_{t,l} \mid \mathcal{F}_{t,l_1}) - \nu_{\boldsymbol{\theta}_t}(s_{t,l}, a_{t,l})\right|$$

$$\overset{(i)}{\leq} 4(r_{\max} + R_{\mathbf{w}}) \cdot \left\|P^{l_2 - l_1}(s, a \mid \mathcal{F}_{t,l_1}) - \nu_{\boldsymbol{\theta}_t}(s, a)\right\|_{TV}$$

$$\leq 4(r_{\max} + R_{\mathbf{w}})\kappa \rho^{l_2 - l_1},$$

where (i) follows from Lemma 8.

Therefore, for the last term in Eq. (20), we have

$$\mathbb{E}\left[\left(\sum_{j=1}^M \lambda_t^j \cdot \left\|\Delta_{\boldsymbol{\theta}_t, \mathbf{w}_t^*}^j - \mathbf{g}_{\boldsymbol{\theta}_t}^j\right\|_2\right)^2 \bigg| \mathcal{F}_t\right] \leq \frac{16}{B}(r_{\max} + R_{\mathbf{w}})^2 + \frac{32}{B^2}\sum_{l_1 < l_2}(r_{\max} + R_{\mathbf{w}})^2 \kappa \rho^{l_2 - l_1}$$

$$\leq \frac{16}{B}(r_{\max} + R_{\mathbf{w}})^2 + \frac{32}{B^2}(r_{\max} + R_{\mathbf{w}})^2 \frac{2\kappa\rho B}{1 - \rho}$$

$$= \frac{16(r_{\max} + R_{\mathbf{w}})^2(1 - \rho + 4\kappa\rho)}{(1 - \rho)B}. \tag{22}$$

Substituting Eqs. (21), (21), (22) into Eq. (20) yields the expected gradient bias as follows

$$\mathbb{E}\left[\left\|\sum_{j=1}^M \lambda_t^j\left(\nabla_{\boldsymbol{\theta}} J^j(\boldsymbol{\theta}_t) - \mathbf{g}_t^j\right)\right\|_2^2 \bigg| \mathcal{F}_t\right]$$

$$\leq 12\zeta_{\text{approx}} + 12\mathbb{E}\left[\left\|w_t^i - w_t^{i,*}\right\|_2^2 \bigg| \mathcal{F}_t\right] + \frac{48(r_{\max} + R_{\mathbf{w}})^2(1 - \rho + 4\kappa\rho)}{(1 - \rho)B}. \tag{23}$$

By letting $\alpha = \frac{1}{3L_J}$, we have

$$\frac{2}{\alpha l_t - 2\alpha^2 l_t^2 L_J} = \frac{18L_J}{-2l_t^2 + 3l_t} \leq 16L_J$$

due to the facts $p_{\min} \leq l_t \leq 1$ and $p_{\min} \leq \frac{1}{M} \leq \frac{3}{4} = \arg\min_{l_t} -2l_t^2 + 3l_t$. Similarly, we also have

$$\frac{\alpha + 2\alpha^2 l_t L_J}{\alpha - 2\alpha^2 l_t L_J} = \frac{3 + 2l_t}{3 - 2l_t} \leq 5.$$

Further, Substituting Eq. (23) into Eq. (19) and taking expectation of $\mathcal{F}_t$ yield

$$\mathbb{E}\left[\|\nabla_{\boldsymbol{\theta}}\boldsymbol{J}(\boldsymbol{\theta}_t)\boldsymbol{\lambda}_t^*\|_2^2\right] \le 16L_J\left(\mathbb{E}\left[\boldsymbol{q}_t^\top \boldsymbol{J}(\boldsymbol{\theta}_{t+1})\right] - \boldsymbol{q}_t^\top \boldsymbol{J}(\boldsymbol{\theta}_t)\right) + 60\zeta_{\text{approx}} + 60\max_{j\in[M]}\mathbb{E}\left[\left\|\mathbf{w}_t^j - \mathbf{w}_t^{j,*}\right\|_2^2\right]$$
$$+ \frac{240(r_{\max} + R_{\mathbf{w}})^2(1 - \rho + 4\kappa\rho)}{(1-\rho)B}. \tag{24}$$

C.2   FOR THE 1ST TERM ON RHS OF EQ. (19)

Let $\hat{T}$ denote a random variable that takes value uniformly random among $\{1, \ldots, T\}$, then taking average of Eq. (24) over $T$ and we have

$$\mathbb{E}\left[\|\nabla_{\boldsymbol{\theta}}\boldsymbol{J}(\boldsymbol{\theta}_{\hat{T}})\boldsymbol{\lambda}_{\hat{T}}^*\|_2^2\right] = \frac{1}{T}\sum_{t=1}^T \mathbb{E}\left[\|\nabla_{\boldsymbol{\theta}}\boldsymbol{J}(\boldsymbol{\theta}_t)\boldsymbol{\lambda}_t^*\|_2^2\right]$$
$$\le \frac{16L_J}{T}\sum_{t=1}^T\left(\mathbb{E}\left[\boldsymbol{q}_t^\top \boldsymbol{J}(\boldsymbol{\theta}_{t+1})\right] - \boldsymbol{q}_t^\top \boldsymbol{J}(\boldsymbol{\theta}_t)\right) + \frac{60}{T}\sum_{t=1}^T\max_{j\in[M]}\mathbb{E}\left[\left\|\mathbf{w}_t^j - \mathbf{w}_t^{j,*}\right\|_2^2\right]$$
$$+ \frac{240(r_{\max} + R_{\mathbf{w}})^2(1 - \rho + 4\kappa\rho)}{(1-\rho)B} + 60\zeta_{\text{approx}}.$$

Specifically,

$$\sum_{t=1}^T\left(\mathbb{E}\left[\boldsymbol{q}_t^\top \boldsymbol{J}(\boldsymbol{\theta}_{t+1})\right] - \boldsymbol{q}_t^\top \boldsymbol{J}(\boldsymbol{\theta}_t)\right) = \mathbb{E}\left[\sum_{t=1}^{T-1}(-\boldsymbol{q}_{t+1} + \boldsymbol{q}_t)^\top \boldsymbol{J}(\boldsymbol{\theta}_{t+1}) - \boldsymbol{q}_1^\top \boldsymbol{J}(\boldsymbol{\theta}_1) + \boldsymbol{q}_T^\top \boldsymbol{J}(\boldsymbol{\theta}_{T+1})\right]$$
$$\overset{(i)}{\le} \mathbb{E}\left[\sum_{t=1}^{T-1}|\boldsymbol{q}_{t+1} - \boldsymbol{q}_t|_1\|\boldsymbol{J}(\boldsymbol{\theta}_{t+1})\|_\infty + \|\boldsymbol{q}_T\|_1\|\boldsymbol{J}(\boldsymbol{\theta}_{T+1})\|_\infty\right]$$
$$\le r_{\max} + r_{\max}\sum_{t=1}^T\mathbb{E}\left[|\boldsymbol{q}_{t+1} - \boldsymbol{q}_t|_1\right]$$
$$\le r_{\max}\left(1 + \frac{2}{p_{\min}}\sum_{t=1}^T\eta_t\right),$$

where (i) follows from Hölder's Inequality since $1/1 + 1/\infty = 1$. Meanwhile, the above result also used the facts

$$\boldsymbol{q}_{t+1} - \boldsymbol{q}_t = \frac{\boldsymbol{\lambda}_{t+1}\odot\mathbf{p}}{l_{t+1}} - \frac{\boldsymbol{\lambda}_t\odot\mathbf{p}}{l_t}$$
$$= \left(\frac{\boldsymbol{\lambda}_{t+1}}{l_{t+1}} - \frac{\boldsymbol{\lambda}_t}{l_t}\right)\odot\mathbf{p}$$

and

$$\frac{\boldsymbol{\lambda}_{t+1}}{l_{t+1}} - \frac{\boldsymbol{\lambda}_t}{l_t} = \frac{(1-\eta_t)\boldsymbol{\lambda}_t + \eta_t\hat{\boldsymbol{\lambda}}_t^*}{l_{t+1}} - \frac{\boldsymbol{\lambda}_t}{l_t}$$
$$= \frac{\left[(1-\eta_t)\boldsymbol{\lambda}_t + \eta_t\hat{\boldsymbol{\lambda}}_t^*\right]\langle\boldsymbol{\lambda}_t, \mathbf{p}\rangle - (1-\eta_t)\boldsymbol{\lambda}_t\langle\boldsymbol{\lambda}_t, \mathbf{p}\rangle - \eta_t\boldsymbol{\lambda}_t\langle\hat{\boldsymbol{\lambda}}_t^*, \mathbf{p}\rangle}{l_{t+1}l_t}$$
$$= \frac{\eta_t\left(\hat{\boldsymbol{\lambda}}_t^*\langle\boldsymbol{\lambda}_t, \mathbf{p}\rangle - \boldsymbol{\lambda}_t\langle\hat{\boldsymbol{\lambda}}_t^*, \mathbf{p}\rangle\right)}{l_{t+1}l_t}.$$

By the above, we have

$$|\boldsymbol{q}_{t+1} - \boldsymbol{q}_t|_1 \le \left|\frac{\eta_t\left(\hat{\boldsymbol{\lambda}}_t^*\langle\boldsymbol{\lambda}_t, \mathbf{p}\rangle - \boldsymbol{\lambda}_t\langle\hat{\boldsymbol{\lambda}}_t^*, \mathbf{p}\rangle\right)}{l_{t+1}l_t}\right|_1$$

$$\leq \frac{\eta_t}{p_{\min}^2} \left( \left| \hat{\boldsymbol{\lambda}}_t^* \langle \boldsymbol{\lambda}_t, \mathbf{p} \rangle \right|_1 + \left| \boldsymbol{\lambda}_t \langle \hat{\boldsymbol{\lambda}}_t^*, \mathbf{p} \rangle \right| \right)$$

$$\leq \frac{2\eta_t}{p_{\min}^2}. \tag{25}$$

This facilitates the analysis to be $M$-independent in the telescoping process. Then, we have

$$\mathbb{E} \left[ \left\| \nabla_{\boldsymbol{\theta}} \boldsymbol{J}(\boldsymbol{\theta}_{\hat{T}}) \boldsymbol{\lambda}_{\hat{T}}^* \right\|_2^2 \right] \leq \frac{16 L_J r_{\max}}{T} \left( 1 + \frac{2}{p_{\min}^2} \sum_{t=1}^{T} \eta_t \right) + \frac{60}{T} \sum_{t=1}^{T} \max_{j \in [M]} \mathbb{E} \left[ \left\| \mathbf{w}_t^j - \mathbf{w}_t^{j,*} \right\|_2^2 \right]$$

$$+ \frac{240(r_{\max} + R_{\mathbf{w}})^2(1 - \rho + 4\kappa\rho)}{(1 - \rho)B} + 60\zeta_{\text{approx}}.$$

## C.3 FINAL RESULT FOR AVERAGE REWARD SETTING

Recalling that $\alpha = \frac{1}{3L_J}$ and by letting $T \geq \frac{48 L_J r_{\max}}{\epsilon} \cdot (1 + \frac{2}{p_{\min}^2} \sum_{t=1}^{T} \eta_t)$, $\mathbb{E} \left[ \left\| \mathbf{w}_t^j - \mathbf{w}_t^{j,*} \right\|_2^2 \right] \leq$

$\frac{\epsilon}{180}$ for any objective $j \in [M]$, and $B \geq \frac{720(r_{\max} + R_{\mathbf{w}})^2(1 - \rho + 4\kappa\rho)}{\epsilon}$ yields

$$\mathbb{E} \left[ \left\| \boldsymbol{\lambda}_{\hat{T}}^\top \nabla_{\boldsymbol{\theta}} \boldsymbol{J}(\boldsymbol{\theta}_{\hat{T}}) \right\|_2^2 \right] \leq \epsilon + 60\zeta_{\text{approx}},$$

with a total sample complexity given by

$$(B + ND)T = \mathcal{O} \left( \left( \frac{1}{\epsilon} + \frac{1}{\epsilon} \log \frac{1}{\epsilon} \right) \frac{1}{\epsilon p_{\min}^2} \right) = \mathcal{O} \left( \frac{1}{\epsilon^2 p_{\min}^2} \log \frac{1}{\epsilon} \right).$$

## C.4 FINAL RESULT FOR DISCOUNTED REWARD SETTING

Similar to the proof in average reward setting, we have

$$\mathbb{E} \left[ \left\| \nabla_{\boldsymbol{\theta}} \boldsymbol{J}(\boldsymbol{\theta}_t) \boldsymbol{\lambda}_t^* \right\|_2^2 \mid \mathcal{F}_t \right] \leq \frac{2 \left( \mathbb{E} \left[ \boldsymbol{\lambda}_t^\top \boldsymbol{J}(\boldsymbol{\theta}_{t+1}) | \mathcal{F}_t \right] - \boldsymbol{\lambda}_t^\top \boldsymbol{J}(\boldsymbol{\theta}_t) \right)}{\alpha - 2\alpha^2 L_J} + \frac{\alpha + 2\alpha^2 L_J}{\alpha - 2\alpha^2 L_J} \mathbb{E} \left[ \left\| \sum_{j=1}^{M} \lambda_t^j \left( \nabla_{\boldsymbol{\theta}} J^j(\boldsymbol{\theta}_t) - \mathbf{g}_t^j \right) \right\|_2^2 \mid \mathcal{F}_t \right], \tag{26}$$

where the last term on the right hand side is bounded by

$$\mathbb{E} \left[ \left\| \sum_{j=1}^{M} \lambda_t^j \left( \nabla_{\boldsymbol{\theta}} J^j(\boldsymbol{\theta}_t) - \mathbf{g}_t^j \right) \right\|_2^2 \mid \mathcal{F}_t \right]$$

$$\leq 3\mathbb{E} \left[ \left( \sum_{j=1}^{M} \lambda_t^j \left\| \nabla_{\boldsymbol{\theta}} J^j(\boldsymbol{\theta}_t) - \Delta_{\boldsymbol{\theta}_t, \mathbf{w}_t^*}^j \right\|_2 \right)^2 \mid \mathcal{F}_t \right]$$

$$+ 3\mathbb{E} \left[ \left( \sum_{j=1}^{M} \lambda_t^j \left\| \mathbf{g}_{\boldsymbol{\theta}_t^*}^j - \mathbf{g}_t^j \right\|_2 \right)^2 \mid \mathcal{F}_t \right] + 3\mathbb{E} \left[ \left( \sum_{j=1}^{M} \lambda_t^j \cdot \left\| \Delta_{\boldsymbol{\theta}_t, \mathbf{w}_t^*}^j - \mathbf{g}_{\boldsymbol{\theta}_t^*}^j \right\|_2 \right)^2 \mid \mathcal{F}_t \right]. \tag{27}$$

Considering the discounted factor $\gamma$, we have

$$\left\| \nabla_{\boldsymbol{\theta}} J^j(\boldsymbol{\theta}_t) - \Delta_{\boldsymbol{\theta}_t, \mathbf{w}_t^*}^j \right\|_2 \leq 2\sqrt{\zeta_{\text{approx}}}, \tag{28}$$

and

$$\left\| \mathbf{g}_{\boldsymbol{\theta}_t^*}^j - \mathbf{g}_t^j \right\|_2 \leq 2 \cdot \left\| \mathbf{w}_t^j - \mathbf{w}_t^{j,*} \right\|_2. \tag{29}$$

For the last term in Eq. (27), we have

$$\mathbb{E} \left[ \left| \sum_{j=1}^{M} \lambda_t^j \cdot \left\| \Delta_{\boldsymbol{\theta}_t, \mathbf{w}_t^*}^j - \mathbf{g}_{\boldsymbol{\theta}_t^*}^j \right\|_2 \right|^2 \mid \mathcal{F}_t \right] \leq \frac{4(r_{\max} + 2R_{\mathbf{w}})^2(1 - \rho + 4\kappa\rho)}{(1 - \rho)B}, \tag{30}$$

since the facts

$$|\delta_{t,l}^j(\mathbf{w}_t^{j,*})| = |r_{t,l+1}^j + \gamma\boldsymbol{\phi}(s_{t,l+1})^\top\mathbf{w}_t^j - \boldsymbol{\phi}(s_{t,l})^\top\mathbf{w}_t^j|_1$$
$$\leq |r_{t,l+1}^j| + \|\gamma\boldsymbol{\phi}(s_{t,l+1}) - \boldsymbol{\phi}(s_{t,l})\|_2 \cdot \|\mathbf{w}_t^j\|_2$$
$$\leq r_{\max} + 2R_{\mathbf{w}},$$

thus, $\|\delta_{t,l}^j(\mathbf{w}_t^{j,*})\boldsymbol{\psi}_{t,l}^{\boldsymbol{\theta}}\|_2 \leq r_{\max} + 2R_{\mathbf{w}}$, and $\Delta_{\boldsymbol{\theta}_t,\mathbf{w}_t^*}^j = \mathbb{E}_{d_{\boldsymbol{\theta}}}\left[\mathbb{E}_{P_{\boldsymbol{\theta}}}\left[\delta_{t,l}^j(\mathbf{w}_t^{j,*}) \mid (a_{t,l}, s_{t,l})\right] \cdot \boldsymbol{\psi}_{t,l}^{\boldsymbol{\theta}}\right] \leq r_{\max} + 2R_{\mathbf{w}}$.

Substituting Eqs. (28), (29), (30) into Eq. (27), we have

$$\mathbb{E}\left[\left\|\sum_{j=1}^M \lambda_t^j\left(\nabla_{\boldsymbol{\theta}}J^j(\boldsymbol{\theta}_t) - \mathbf{g}_t^j\right)\right\|_2^2 \Big| \mathcal{F}_t\right] \leq 12\zeta_{\text{approx}} + 12\max_{j\in[M]}\mathbb{E}\left[\left\|\mathbf{w}_t^j - \mathbf{w}_t^{j,*}\right\|_2^2 \Big| \mathcal{F}_t\right] + \frac{12(r_{\max} + 2R_{\mathbf{w}})^2(1 - \rho + 4\kappa\rho)}{(1 - \rho)B}.$$
$$(31)$$

Substituting Eq. (31) into Eq. (26), letting $\alpha = \dfrac{1}{3L_J}$, taking expectation of $\mathcal{F}_t$, and taking average of Eq. (26) over $T$ yields

$$\mathbb{E}\left[\|\nabla_{\boldsymbol{\theta}}\boldsymbol{J}(\boldsymbol{\theta}_{\hat{T}})\boldsymbol{\lambda}_{\hat{T}}^*\|_2^2\right] = \frac{1}{T}\sum_{t=1}^T\mathbb{E}\left[\|\nabla_{\boldsymbol{\theta}}\boldsymbol{J}(\boldsymbol{\theta}_t)\boldsymbol{\lambda}_t^*\|_2^2\right]$$

$$\leq \frac{16L_J}{T}\sum_{t=1}^T\left(\mathbb{E}\left[\boldsymbol{\lambda}_t^\top\boldsymbol{J}(\boldsymbol{\theta}_{t+1})\right] - \boldsymbol{\lambda}_t^\top\boldsymbol{J}(\boldsymbol{\theta}_t)\right) + \frac{60}{T}\sum_{t=1}^T\max_{j\in[M]}\mathbb{E}\left[\left\|\mathbf{w}_t^j - \mathbf{w}_t^{j,*}\right\|_2^2\right]$$

$$+ \frac{60(r_{\max} + 2R_{\mathbf{w}})^2(1 - \rho + 4\kappa\rho)}{(1 - \rho)B} + 60\zeta_{\text{approx}},$$

where

$$\sum_{t=1}^T\left(\mathbb{E}\left[\boldsymbol{q}_t^\top\boldsymbol{J}(\boldsymbol{\theta}_{t+1})\right] - \boldsymbol{q}_t^\top\boldsymbol{J}(\boldsymbol{\theta}_t)\right) = \mathbb{E}\left[\sum_{t=1}^{T-1}(-\boldsymbol{q}_{t+1} + \boldsymbol{q}_t)^\top\boldsymbol{J}(\boldsymbol{\theta}_{t+1}) - \boldsymbol{q}_1^\top\boldsymbol{J}(\boldsymbol{\theta}_1) + \boldsymbol{q}_T^\top\boldsymbol{J}(\boldsymbol{\theta}_{T+1})\right]$$

$$\leq \mathbb{E}\left[\sum_{t=1}^{T-1}|\boldsymbol{q}_{t+1} - \boldsymbol{q}_t|_1\|\boldsymbol{J}(\boldsymbol{\theta}_{t+1})\|_\infty + |\boldsymbol{q}_T|_1\|\boldsymbol{J}(\boldsymbol{\theta}_{T+1})\|_\infty\right]$$

$$\leq \sum_{t=1}^{T-1}\left(\frac{2\eta_t}{p_{\min}^2} \cdot \frac{r_{\max}}{1 - \|\boldsymbol{\gamma}\|_\infty}\right) + \frac{r_{\max}}{1 - \|\boldsymbol{\gamma}\|_\infty}$$

$$\leq \frac{r_{\max}}{1 - \|\boldsymbol{\gamma}\|_\infty}\left(1 + \frac{2}{p_{\min}^2}\sum_{t=1}^T\eta_t\right),$$

where the 2nd from the last inequality, we used inequality 25 for discounted setting. Then, we have

$$\mathbb{E}\left[\|\nabla_{\boldsymbol{\theta}}\boldsymbol{J}(\boldsymbol{\theta}_{\hat{T}})\boldsymbol{\lambda}_{\hat{T}}\|_2^2\right] \leq \frac{16L_J r_{\max}}{T(1 - \|\boldsymbol{\gamma}\|_\infty)}\left(1 + \frac{2}{p_{\min}^2}\sum_{t=1}^T\eta_t\right) + \frac{60}{T}\sum_{t=1}^T\max_{j\in[M]}\mathbb{E}\left[\left\|\mathbf{w}_t^j - \mathbf{w}_t^{j,*}\right\|_2^2\right]$$

$$+ \frac{60(r_{\max} + 2R_{\mathbf{w}})^2(1 - \rho + 4\kappa\rho)}{(1 - \rho)B} + 60\zeta_{\text{approx}}.$$

By letting $T \geq \dfrac{48L_J r_{\max}}{\epsilon(1 - \|\boldsymbol{\gamma}\|_\infty)} \cdot (1 + \frac{2}{p_{\min}^2}\sum_{t=1}^T\eta_t)$, $\mathbb{E}\left[\left\|\mathbf{w}_t^j - \mathbf{w}_t^{j,*}\right\|_2^2\right] \leq \dfrac{\epsilon}{240}$ for any objective $j \in [M]$, and $B \geq \dfrac{240(r_{\max} + 2R_{\mathbf{w}})^2(1 - \rho + 4\kappa\rho)}{\epsilon}$ yields

$$\mathbb{E}\left[\|\boldsymbol{\lambda}_{\hat{T}}^\top\nabla_{\boldsymbol{\theta}}\boldsymbol{J}(\boldsymbol{\theta}_{\hat{T}})\|_2^2\right] \leq \epsilon + 60\zeta_{\text{approx}},$$

with total sample complexity given by

$$(B + ND)T = \mathcal{O}\left(\left(\frac{1}{\epsilon} + \frac{1}{\epsilon}\log\frac{1}{\epsilon}\right)\frac{1}{\epsilon p_{\min}^2}\right) = \mathcal{O}\left(\frac{1}{\epsilon^2 p_{\min}^2}\log\frac{1}{\epsilon}\right).$$

□