# OpenReview forum: "Enabling Pareto-Stationarity Exploration in Multi-Objective Reinforcement Learning: A Weighted-Chebyshev Multi-Objective Actor-Critic Approach"
_ICLR.cc/2025/Conference — Submitted to ICLR 2025_

### Official Review · Reviewer_qYKr · 2024-11-02

**Soundness:** 3
**Presentation:** 4
**Contribution:** 2
**Rating:** 6
**Confidence:** 4

**Summary:**

This paper studies multi-objective reinforcement learning by extending existing multi-objective optimization techniques within the framework of actor-critic methodologies. Central to the discussion is the concept of (local) (weak) Pareto optimality, which serves as the foundational solution criterion for balancing multiple objectives. The proposed training architecture employs a single actor coupled with multiple critics, where each critic is specifically trained to approximate the value function corresponding to a distinct objective. During the training process, the actor is updated using a weighted gradient approach that integrates the estimated values from each critic. These weights are supposed to balance the influence of each objective, guiding the actor towards policies that achieve a desirable trade-off among the competing objectives. The authors rigorously demonstrate that their method is capable of converging to locally Pareto optimal policies. This convergence is guaranteed up to an error bound, which is linked to the accuracy of the critic's value function estimations.

**Strengths:**

1. The manuscript is exceptionally well-written, presenting complex concepts in a clear and accessible manner. It seems that the authors have meticulously structured the paper, which significantly enhances readability. Additionally, the discussion of related works is thorough, providing a comprehensive overview of existing approaches in multi-objective reinforcement learning and actor-critic methodologies.

The experimental component of the study appears to be conducted on a large-scale real-world short video platform. (However, the manuscript lacks detailed descriptions of the experimental setup, making it challenging to fully assess the validity the results. Clarification on the this concern is detailed in the section below.)

**Weaknesses:**

1. The major weakness might lie in the novelty of the proposed method. To the knowledge of the reviewer, Although the theoretical analysis partially answers the challenges mentioned in the introduction, the proposed method seems to largely based on (Désidéri, 2012) and  (Momma et al., 2022), with the multi-objective optimization gradients replaced with policy gradients.

2. See questions below.

**Questions:**

1. In Theorem 4, what is the different between |w*-w| and zeta? Intuitively, both these two terms are related to the error of value functions. The reviewer is confused because the definition of w* also needs clarification. The optimality of the critic network depends on both phi(s) and w. Different state representation phi might correspond to different optimal w. Is phi function fixed?

2. In terms of the proposed method, what is the major novelty compared to previous methods in MOO literature? The reviewer understands the discussions already presented in the paper, but is just wondering whether the proposed method is replacing the gradients in MOO with policy gradients given by different critics.

3. It seems that the detailed experiment description is not specified even in the appendix. What is the horizon of this game? What is the size of the state set?

4. Is the weight vector p predefined or learned?

---

> ### Author Response · Authors · 2024-11-22
> **Response to Reviewer qYKr's Comments**
>
> >**Your Comment 1 (Also Weakness 1):** The major weakness might lie in the novelty of the proposed method. To the knowledge of the reviewer, Although the theoretical analysis partially answers the challenges mentioned in the introduction, the proposed method seems to largely based on (Désidéri, 2012) and (Momma et al., 2022), with the multi-objective optimization gradients replaced with policy gradients.
>
> **Our Response:** Thanks for your comments. We agree with the reviewer that the actor component of the proposed MOAC approach is indeed based on the MGDA approach in (Désidéri, 2012) and (Momma et al. (2022)). However, we do want to emphsize that the adaptation of these techniques to the MORL setting is by no means straightforward due to the unique challenges in MORL, which requires additional new algorithmic techniques (e.g., the momentum-based approach to eliminate systematic biases). Moreover, the theoretical finite-time convergence proof and analysis are new, which cannot be found in (Désidéri, 2012) and (Momma et al. (2022)). In particular, the Pareto stationary convergence rate's dependency on the weight vector $\mathbf{p}$ is a new result in the literature. In addition, the empirical studies in Figure 1 does indicate that our proposed WC-MOAC approach in Algorithm 1 is able to explore a much larger Pareto stationarity front footprint than that of the conventional approach based on linear scalarization.
>
> ________________
> >**Your Comment 2 (Also Question 1):** In Theorem 4, what is the different between |w*-w| and zeta? Intuitively, both these two terms are related to the error of value functions. The reviewer is confused because the definition of w* also needs clarification. The optimality of the critic network depends on both $\phi(s)$ and w. Different state representation phi might correspond to different optimal w. Is $\phi$ function fixed?
>
> **Our Response:** Thank you for your comment. $\mathbf{w}^{i,*} - \mathbf{w}_t^i$ and $\zeta_{\text{approx}}$ have different physical meansings, which are stated as follows:
>
> **1) The meaning of $\zeta_{\text{approx}}$:** The quantity $\zeta_{\text{approx}}$ represents the *fundamental inability* of the linear function approximation apporach for accurately quantifying the $i$-th value function (i.e., $V^{i}(s)\approx\phi(s)^{\top} \mathbf{w}^{i}$). Mathematically, $\zeta_{\text{approx}}=\max_{i\in[M]}V^{i}(\cdot) - \mathbf{w}^{i,\*}\Phi$ characterizes the maximum gap between the ground truth value function $V^{i}$ and best linear approximation given the feature matrix $\Phi$ and the optimal linear approximation solution $\mathbf{w}^{i,*}$ among all $M$ objectives.
>
> **2) The meaning of $\mathbf{w}^{i,\*} - \mathbf{w}_t^i$:** The quantity $\mathbf{w}^{i,\*} - \mathbf{w}^i_t$ represents the finite-time convergence gap compared to the optimal linear function approximation solution $\mathbf{w}^{i,*}$, which itself has a gap $\zeta_{\text{approx}}$ to the ground truth value function $V^i(\cdot)$ of the $i$-th objective.
>
> For ease of explaination, let's use the discounted reward setting in here as a concrete example. The value function $V^{i}(s)$ for the $i$-th objective at $s\in \mathcal{S}$ follows the canonical definition of the discounted reward setting. However, in some MDPs, due to large state space $\mathcal{S}$, approximation is used to cope with the high dimensionality challenge. In this paper, we applied linear approximation assuming the feature vector mapping $\phi(s)$ is *given*. How to choose a suitable feature mapping $\phi(\cdot)$ is an interesting problem by itself, but beyond the scope of this paper. This suggests for each objective $i$, the critic will maintain a $\mathbf{w}^{i}$ parameter to evaluate the $i$-th objective. $\mathbf{w}^{i,\*}$ is used to denote the optimal approximation under such linear approximation. Please see Line 854 in the supplementary material for the precise defition of $\mathbf{w}^{i,\*}$ under the discounted reward setting. In Theorem 4, the term $\mathbf{w}^{i}-\mathbf{w}^{i,*}$ refers to the error in convergence from critic component of Algorithm 1 and the optimal linear approximation parameter for $i$-th objective.
>
> ________________

---

> > ### Author Response · Authors · 2024-11-22
> > **Response to Reviewer qYKr's Comments (Continued)**
> >
> > >**Your Comment 3 (Also Question 2):** In terms of the proposed method, what is the major novelty compared to previous methods in MOO literature? The reviewer understands the discussions already presented in the paper, but is just wondering whether the proposed method is replacing the gradients in MOO with policy gradients given by different critics.
> >
> > **Our Response:** Yes, as we mentioned in our response to your Comment 1, the adaptation of the MOO techniques to the MORL setting calls for many novel algorithmic designs due to the uniques challenges in MORL. In addition to the momentum-based technique in our reponse to your Comment 1, in the actor component of our WC-MOAC approach, the policy gradient is updated through a judious integration of the multi-gradient descent and *weighted-Chebyshev* approahces. Moreover, this new algorithmic design for MORL necessitates new finite-time convergence proof in our theoretical Pareto stationary convergence analysis, which leads to the *new* insights and knowledge of the dependency on the weight vector $\mathbf{p}$ in the WC-MOAC's Pareto stationary convergence.
> >
> > >**Your Comment 4 (Also Question 3):** It seems that the detailed experiment description is not specified even in the appendix. What is the horizon of this game? What is the size of the state set?
> >
> > **Our Response:** Thanks for your question. In this work, we consider infinite time horizon. The state size is 1218, which is indicated in Table 2 in the Appendix.
> > ________________
> >
> > >**Your Comment 5 (Also Question 4):** Is the weight vector $\mathbf{p}$ predefined or learned?
> >
> > **Our Response:** Thanks for your question. The weight vector $\mathbf{p}$ is a given input in each run of Algorithm 1. To further understand the role of $\mathbf{p}$, note first that Lemma 1 implies that there is a one-to-one mapping between a WC-solution and a Pareto-optimal solution and further suggest a systematical way to explore the Pareto-front (assuming the WC-scalaization problem can be solved to optimality) by varying $\mathbf{p}$ as input to Algorithm 1. Thus, when applying Algorithm 1, the decision-maker is supposed to vary and enumerate all possible vectors $\mathbf{p}$ in the $M$-dimensional standard simplex. We hope this clarifies the role of $\mathbf{p}$ in our WC-MOAC approach.
> > ________________

---

### Official Review · Reviewer_JPv7 · 2024-11-04

**Soundness:** 2
**Presentation:** 3
**Contribution:** 2
**Rating:** 3
**Confidence:** 3

**Summary:**

The paper tackles the multiobjective reinforcement learning problem. It builds on top of the actor-critic algorithm using multi-gradient
descent algorithm (MGDA), named multiobjective actor-critic (MOAC), designed by Zhou et al. (2024) and leverages the framework introduced by Momma et al. (2022) to optimize a multiobjective weighted by a preference vector, denoted by p. The difference from Zhou et al. (2024) is the introduction of preference weighting, which allows exploration of the Pareto Stationary front solutions. The proposed WC-MOAC algorithm maintains the sample complexity $\tilde{O}(\epsilon^{-2})$, shown in Zhou et al. (2024), after the addition of the preference weighting. Meanwhile, the algorithm is tested on a large-scale real-world dataset from the recommendation logs of the short video streaming mobile app Kuaisho.

Zhou, T., Hairi, F. N. U., Yang, H., Liu, J., Tong, T., Yang, F., ... & Gao, Y. (2024). Finite-time convergence and sample complexity of actor-critic multi-objective reinforcement learning. ICML.

Momma, M., Dong, C., & Liu, J. (2022, June). A multi-objective/multi-task learning framework induced by pareto stationarity. In International Conference on Machine Learning (pp. 15895-15907). PMLR.

**Strengths:**

The methodology and the algorithm are clearly stated. The assumptions and the main theorem are well presented.

**Weaknesses:**

The contribution can be stated more directly. For example, the abstract mentions that the algorithm fills the gap "to systematically explore the Pareto-stationary solutions". It would be more apparent if the paper explained why it is needed to explore these solutions and how the exploration is achieved.

The paper overstates the contribution without a clear acknowledgement of previous work. The paper should introduce the work by Zhou et al. (2024) with more details, and it is also necessary to make a comparison to clarify the novelty of the work. Some examples of overstated contributions examples are listed below. (1) It is claimed in the abstract that "the performance of our WC-MOAC algorithm significantly outperforms other baseline MORL approaches." However, the performance drops from the behaviour clone baseline for two objectives.  (2) Line 100 states, "Collectively, our results provide the first building block toward a theoretical foundation for MORL." Zhou et al. (2024) gave a sample complexity bound of the same order.  (3) Line 113 states, "To mitigate the cumulative systematic bias injected from the WC-scalarization weight direction and finite-length state-action trajectories, we propose a momentum-based mechanism in WC-MOAC." What is the difference in the mechanism from Zhou et al. (2024)?

The clarity and correctness of writing can also be improved. Please refer to the question section for details.

The experiment design is not fully convincing, and the result is hard to understand. Please refer to the question section for details. More experiments can be designed to show the advantages of using the preference.

**Questions:**

1. Line 113 states, “ To mitigate the cumulative systematic bias injected from the WC-scalarization weight direction and finite-length state-action trajectories, we propose a momentum-based mechanism.” What is this systematic bias? Also, why does this momentum mechanism remove the dependence on the number of objectives M?

2. Where is Lemma 1 cited from? In Qiu et al. (2024) Proposition 4.2, they state that a stochastic policy has to maximize linearized scalarization for all weight vectors p to be a weakly Pareto optimal policy. However, Lemma 1 of the paper only requires maximizing the infinity norm of the scalarization for some weight vector.

3. For the experiment in Table 1, how is the preference vector chosen?

4. For the experiment in Figure 1, would TSCA work better by switching the main objective? More experiment designs to show the necessity of preference would be better.

5. The paper focuses on the finite-time convergence results for multi-objective actor-critic. What is the advantage of using actor-critic algorithms?

Qiu, S., Zhang, D., Yang, R., Lyu, B., & Zhang, T. (2024). Traversing pareto optimal policies: Provably efficient multi-objective reinforcement learning. arXiv preprint arXiv:2407.17466.

---

> ### Author Response · Authors · 2024-11-21
> **Response to Reviewer JPv7's comments**
>
> >**Your Comment 1 (Also Weakness 1):** The contribution can be stated more directly. For example, the abstract mentions that the algorithm fills the gap "to systematically explore the Pareto-stationary solutions". It would be more apparent if the paper explained why it is needed to explore these solutions and how the exploration is achieved.
>
> **Our Response:** When decision-maker aims for optimizing in a multi-objective setting, at the beginning there's not always a clear priori on what the preference for a particular Pareto Optimal solution among the set of Pareto Front(PF). Considering such potential uncertainty in priori, exploring the PF has been an on-going effort on multi-objective problem settings [1]. On the other hand, once PF is entirely characterizes, it allows a Posteriori selection of a preferable Pareto solution, giving better insight to a decision-maker [2]. Towards this effort, we are proposing a WC-MOAC algorithm that enables systematically exploration of Pareto solution in MORL context, which is under-investigated.
>
> [1] Kristof Van Moffaert and Ann Nowé. Multi-objective reinforcement learning using sets of pareto dominating policies. The Journal of Machine Learning Research, 15(1):3483–3512, 2014.
>
> [2] Simone Parisi, Matteo Pirotta, and Marcello Restelli. Multi-objective reinforcement learning through continuous pareto manifold approximation. Journal of Artificial Intelligence Research, 57:187–227, 2016.
>
> ________________
>
> >**Your comment (Also Weakness 2):** The paper overstates the contribution without a clear acknowledgement of previous work. The paper should introduce the work by Zhou et al. (2024) with more details, and it is also necessary to make a comparison to clarify the novelty of the work.
>
> **Our Response:** To highlight the contribution of this work, we provide following comparisons with (Zhou et al. (2024)).
>
> **Objective Differences**: Our work focuses on systematically exploring the Pareto Stationary front using a weighted-Chebyshev formulation. This objective is distinctly different from that of (Zhou et al., 2024), which aims to find a Pareto stationary point rather than exploring the full Pareto stationary front.
>
> **Methodological Innovation**: We have incorporated a weighted-Chebyshev formulation, a technique which is entirely absent in (Zhou et al., 2024). Our paper devotes Section 4.2 to describing this approach in detail, illustrating how we use the weighted-Chebyshev method to achieve systematic exploration. This technique leverages an exploration/weight vector $\mathbf{p}$, extensively explained in our derivations.
>
> **Unique Theoretical Contributions**: Our Theorem 4 and Corollary 5 highlight the effects of the weight vector $\mathbf{p}$, particularly through its minimum entry, $p_{\min}$, on convergence rate to Pareto stationarity and corresponding sample complexities for each exploration. While some order-wise results appear similar, the coefficients in Theorem 4 are different from those in (Zhou et al., 2024).
>
> **Empirical Evidence**: In Figure 1 of Experiment Section, we present the Pareto footprint, i.e., the Pareto fronts explored by our method via varying exploration/weight vector $\mathbf{p}$ and a popular linear scalarization approach. In contrast, (Zhou et al., 2024) includes no such comparative analysis.
>
> ________________
> >**Your Comment 3(Also Weakness 3):** Line 100 states, "Collectively, our results provide the first building block toward a theoretical foundation for MORL." Zhou et al. (2024) gave a sample complexity bound of the same order.
>
> **Our Response:** We are happy to remove this statement and acknowledge the contribution from the previous literature including (Zhou et al. (2024)). For the second point, we want to emphasize that proving the same sample complexity doesn't mean it's not a new result in the context of leveraging the weight-Chebyshev. In particular, reiterating the point in our response to your comment 2, while some order-wise results appear similar, the coefficients in Theorem 4 are different from those in (Zhou et al., 2024).
>
> ________________

---

> > ### Author Response · Authors · 2024-11-21
> > **Response to Reviewer JPv7's comments (Continued)**
> >
> > >**Your Comment 4 (Also Weakness 4):** Line 113 states, "To mitigate the cumulative systematic bias injected from the WC-scalarization weight direction and finite-length state-action trajectories, we propose a momentum-based mechanism in WC-MOAC." What is the difference in the mechanism from Zhou et al. (2024)?
> >
> > **Our Response:** Indeed, momenteum approach-wise is the same approach in this paper and (Zhou et al. (2024)). However, WC inspired Eq.(10) and $\mathbf{g_t}$ update have incurred significant differences in the following senses.
> > The novelty in the analysis stems from involving Hadamard product by formulating the problem in a weighted-Chebyshev problem, where as in (Zhou et al (2024)), there's no such operand. In contrast to pure MGDA approach, where the $\mathbf{\lambda_t}$ is used to weigh multiple gradients from objectives, incorporating WC aspect, due to aforementioned Hadamard product, now requires the analysis to construct a pseudo weight $\mathbf{q_t}:=\frac{\mathbf{\lambda_t}\odot \mathbf{p}}{\langle \mathbf{\lambda_t}, \mathbf{p}\rangle}$ and utilize properties of such pseudo weight. Please see Line 907-970, 1160-1210, where we carefully handled such pseudo weight. As a result of this analysis, we observed that $p_{\min}$ is a cruicial quantity in characterizing the convergence to stationarity. As one can easily observe that coefficients in front of the cruicial terms are significantly different from (Zhou et al. (2024)). For example, the cofficient in front of $1/T$ term is $16L_Jr_{\max}(1+\frac{2}{p_{\min}^{2}}\sum_{t=1}^{T}\eta_t)$ in this work and $18L_Jr_{\max}(1+2\sum_{t=1}^{T}\eta_t)$ in (Zhou et al. (2024)).
> >
> > In fact, the WC-approach proposed in this paper is a more general approach than that of (Zhou et al. (2024)). When $u=0$ and $\mathbf{p}=(1,\cdots,1)^{\top}$ (for simplicity, ignoring the scaling), the results in our work imply those in (Zhou et al. (2024)). To see this, Eq.(10) reduces to the $\min_{\lambda}||\mathbf{K_p}\lambda||^{2}$, which is equivalent to MGDA where the Quadratic programming is to solve $\min_{\lambda}||\mathbf{G}_t\mathbf{\lambda}||^{2}$ where $\mathbf{G}_t=[\mathbf{g}^{i}_t,\cdots, \mathbf{g}^{M}_t]$ and $\mathbf{g}^{i}_t$ is defined in the second column of Line 394. In algorithm 1, the update of $\mathbf{g}_t$ will reduce to $\mathbf{g}_t= \mathbf{G}_t\mathbf{\lambda}_t$.
> >
> > ________________
> > >**Your Comment 5 (Also Question 1):** Line 113 states, “ To mitigate the cumulative systematic bias injected from the WC-scalarization weight direction and finite-length state-action trajectories, we propose a momentum-based mechanism.” What is this systematic bias? Also, why does this momentum mechanism remove the dependence on the number of objectives M?
> >
> > **Our Response:** Thank you for the insightful question. To see the advantage of momentum approch in obtaining an $M$-independent result, first observe the right-hand-side (RHS) of the inequality in Eq. (18), which consists of two terms. For the first term on the right-hand-side of the inequality in Eq. (18), it further goes through a telescoping operation (see Appendix C.2 Line 1173). The so-called systematic bias term is the summand term on the Left-hand-side of in the equation of Line 1173, i.e. $\mathbb{E}[\mathbf{q_t^{\top}}\mathbf{J}(\mathbf{\theta_{t+1}})]-\mathbf{q_{t}}\mathbf{J}(\mathbf{\theta_t})$. Here, the contribution of momentum-based $\bf{\lambda_t}$ as in Eq. (11) helps with providing bound on $\mathbf{q_{t+1}}-\mathbf{q_{t}}$ via Line 1187 and 1192.
> >
> > ________________
> > >**Your Comment 6 (Also Question 2)** Where is Lemma 1 cited from? In Qiu et al. (2024) Proposition 4.2, they state that a stochastic policy has to maximize linearized scalarization for all weight vectors p to be a weakly Pareto optimal policy. However, Lemma 1 of the paper only requires maximizing the infinity norm of the scalarization for some weight vector.
> >
> > **Our Response:** Lemma 1 is cited from Proposition 4.7 and its proof within in (Qiu et al. (2024)). In Page 42 of proof for Proposition 4.7, part 1) and part 2) provide if and only if statement in Lemma 1 of this paper.
> >
> > ________________
> > >**Your Comment 7 (Also Question 3):** For the experiment in Table 1, how is the preference vector chosen?
> >
> > **Our Response:** The particular $\mathbf{p}$ vector we chose for the result in Table 1 is $(0.2,0.2,0.2,0,0.4)^{\top}$. The rationale behind the weight choice is inpired by Kuaishou dataset and its application to emphasize on WatchTime with higher weight, and on the other hand ignoring dislike, essentially making it a 4-objective problem. Then for the remaining objectives, we chose uniform weights to highlight no particular preference within this subset of objectives.
> >
> > ________________

---

> > > ### Author Response · Authors · 2024-11-21
> > > **Response to Reviewer JPv7's comments (Continued)**
> > >
> > > >**Your Comment 8 (Also Question 4):** For the experiment in Figure 1, would TSCA work better by switching the main objective? More experiment designs to show the necessity of preference would be better.
> > >
> > > **Our Response:** In principle, it is possible to switch objects as suggested to get potentially different results for TSCAC algorithm. However, it does require priori knowleges on what should constitute reasonable contraints as it is a constraint actor-critic algorithm. On the other hand, it shows the advantage of our approach, which explores systematically, as it doesn't require additional domain knowledge which serves as constraints.
> > >
> > > ________________
> > > >**Your Comment 9 (Also Question 5):** The paper focuses on the finite-time convergence results for multi-objective actor-critic. What is the advantage of using actor-critic algorithms?
> > >
> > > **Our Response:** If one were to compare actor-critic(AC) with other algorithms, notably Q-learning. We have the following comparisons:
> > > 1. on policy vs off policy: AC generally speaking is an on-policy method. In constrast to Q-learning, which is an off-policy algorithm, the Q-function of the sampling policy doesn't align with the current Q-values being computed. As a result, AC provides a better understanding and intuition for the current policy for both single-objective RL and MORL.
> > > 2. Continuous vs discrete state-action space (Grondman et al. (2012)): As policy for AC is parameterized, it is natural for AC to extend to continuous state-action space in comparison with Q-learning. For Q-learning to work properly, Q functions are typically required to be approximated due to $\max$ operator, which is not well-defined in non-compact continuous space. This should hold true for both single-object and multi-objective scenarios.
> > >
> > > Grondman, Ivo, et al. "A survey of actor-critic reinforcement learning: Standard and natural policy gradients." IEEE Transactions on Systems, Man, and Cybernetics, part C (applications and reviews) 42.6 (2012): 1291-1307.
> > >
> > > 3. Since AC is a policy gradient based algorithm, it is natural to incorporate multiple-gradient based approach, which is a standard approach for finding Pareto stationary points.
> > >
> > > ________________
> > >
> > > We hope our responses have addressed the reviewer's concerns, and we are happy to answer any further questions. If the concerns are addressed, we would highly appreciate a re-rating of our work.

---

> ### Comment · Reviewer_JPv7 · 2024-11-22
> **Misguiding Paper Writing With Over-stated Contributions**
>
> Thanks a lot for your detailed replies! We can now focus on my previous Comments 1, 2, 7 and 8.
>
> The main issue with the paper's organization is that others can strongly overestimate the contributions.
>
> As stated in Reviewer P2zK's summary, "The WC-MOAC algorithm is designed to combine multi-temporal-difference (TD) learning in the critic phase with ... multi-gradient descent in the actor phase, effectively managing the complexities of non-convexity and scalability in multi-objective problems. The primary theoretical result of this work is a finite-time sample complexity bound, which is independent of the number of objectives ... This independence ... is a notable theoretical advance."
>
> Reviewer qYKr02 stated:" Central to the discussion is the concept of (local) (weak) Pareto optimality, which serves as the foundational solution criterion for balancing multiple objectives. The proposed training architecture employs a single actor coupled with multiple critics, where each critic is specifically trained to approximate the value function corresponding to a distinct objective."
>
> None of these are your novel contributions, which Zhou and colleagues have introduced. The paper writing is strongly misleading.
>
> As your reply to the first comment stated, this work "focuses on systematically exploring the Pareto Stationary front using a weighted-Chebyshev formulation." The introduction of the preference weighting is the main contribution.
>
> However, your experiment design does not show the advantage of including such weighting. For the experiment in Table 1, the effect of emphasizing "Comment" or "Dislike" objectives can be more persuasive. As replied to Comments 7 and 8, the preference vector is not trained. What is the systematic way of deciding the weighting in your approach? As TSCAC, does your method also require domain knowledge? Meanwhile, for the experiment shown in Figure 1, concentrating on click, dislike, and watch do not differ much.

---

> > ### Author Response · Authors · 2024-11-26
> > **Response for Reviewer JPv7's Follow-up Comments**
> >
> > Thank you very much for the prompt reply. Here are our point-to-point response for your follow-up comments.
> > >**Follow-up comment 1:**
> > The main issue with the paper's organization is that others can strongly overestimate the contributions.
> > As stated in Reviewer P2zK's summary, "The WC-MOAC algorithm is designed to combine multi-temporal-difference (TD) learning in the critic phase with ... multi-gradient descent in the actor phase, effectively managing the complexities of non-convexity and scalability in multi-objective problems. The primary theoretical result of this work is a finite-time sample complexity bound, which is independent of the number of objectives ... This independence ... is a notable theoretical advance."
> > Reviewer qYKr02 stated:" Central to the discussion is the concept of (local) (weak) Pareto optimality, which serves as the foundational solution criterion for balancing multiple objectives. The proposed training architecture employs a single actor coupled with multiple critics, where each critic is specifically trained to approximate the value function corresponding to a distinct objective."
> > None of these are your novel contributions, which Zhou and colleagues have introduced. The paper writing is strongly misleading.
> >
> > **Our Reseponse:** We agree with the reviewer that the convergence error bound in (Zhou et al.(2024)) in also independent of $M$, the number of objectives. However, we do want to emphasize that these seemingly similar $M$-independent results are for two **different** algorithms. Specifically, in (Zhou et al.(2024)),  $M$-independence result is for a pure *MGDA-type approach* in their the actor component. In contrast, in our work, the $M$-independence result is established for a **hybrid weighted-Chebyshev MGDA** approach for the actor component. Consequently, the proof and theoretical analysis for our $M$-independence result is quite different from that in (Zhou et al. (2024)). Here, we want to outline how we overcome two *new and unique* challenges in our proof of the $M$-independence result for the hybrid weighted-Chebyshev MGDA approach, which stems from two terms on the right-hand-side (RHS) of the inequality in Eq. (18):
> >
> > 1) For the first term on the RHS of the inequality in Eq. (18), we note that it will further go through a telescoping operation (see Appendix C.2 Line 1173). However, due to the WC-component $\lambda^{\top}(\mathbf{p} \odot \left(\mathbf{J_{\text{ub}}}^{*}-\mathbf{J} (\mathbf{\theta}) \right))$ (cf. Problem (10)) of the actor update, it requires a *clever* construction of **pseudo weights** defined as $\mathbf{q_{t}}:=\frac{\mathbf{p}\odot \lambda_t}{\langle \mathbf{p},\lambda_t\rangle}$, as the naive approach for MGDA doesn't apply here. Note that such a pseudo weight construction is *not* needed in (Zhou et al. (2024)). The analysis in this work further requires applying the properties of this pseudo weight. Please see our *proofs in Line 948-1008, 1197-1247*, where we carefully handled such a pseudo weight. As a result of this analysis, we are able to show that $p_{\min}$ is a key quantity in affecting the convergence to Pareto stationarity with $M$-independence.
> >
> > 2) For the second term on the RHS of the inequality in Eq. (18), it requires establishing the $\mathbb{E}[\\|\sum_{i=1}^{M}\lambda^{i}(w_t^{i}-w^{i,\*})\\|^{2}]\le\max_{i\in[M]}\mathbb{E}[\\|w_t^{i}-w^{i,\*}\\|^{2}]$ for any $\sum_{i=1}^{M}\lambda^{i}=1$ and $\lambda^{i}\ge 0$ for all $i\in[M]$. This **proof analysis from Line 1076- Line 1102**, which utlizes the independence of evaluations of objective $i$ and $j$ given trajectory for any $i\neq j$ and $i,j\in[M]$, are new and different from (Zhou et al. (2024)).
> >
> > Regarding Reviewer qYKr02's comments (i.e., *"Central to the discussion is the concept of (local) (weak) Pareto optimality, which serves as the foundational solution criterion for balancing multiple objectives. The proposed training architecture employs a single actor coupled with multiple critics, where each critic is specifically trained to approximate the value function corresponding to a distinct objective."*), we want to clarify that the actor-critic (AC) architecture itself is indeed not our contribution. However, we want to point out that the hybrid weighted Chebyshev MGDA-type technique for dynamic $\mathbf{\lambda}$-weighting in the actor component of our WC-MOAC framework and its subsequent theoretical and empirical analysis are our **new** contributions.
> >
> > With the above clarifications, we are happy to re-position our paper and tone down some of the contribution statements to avoid similar misunderstandings arising. We thank for the reviewer's comments that help us improve the quality of our paper.
> > ______________

---

> > > ### Author Response · Authors · 2024-11-26
> > > **Response for Reviewer JPv7's Follow-up Comments (Continued)**
> > >
> > > >**Follow-up Comment 2:** As your reply to the first comment stated, this work "focuses on systematically exploring the Pareto Stationary front using a weighted-Chebyshev formulation." The introduction of the preference weighting is the main contribution.
> > > However, your experiment design does not show the advantage of including such weighting. For the experiment in Table 1, the effect of emphasizing "Comment" or "Dislike" objectives can be more persuasive.
> > >
> > > **Our Response:** Thanks for your comment and we would like to further clarify on this. The rationale behind our weight choice $[0.2,0.2,0.2,0,0.4]^{\top}$ in Table 1 is inspired by the Kuaishou dataset for short video streaming, which typically emphasize on "WatchTime" with higher weight (the most important performance metric in short video streaming), and on the other hand ignoring "Dislike" (not important for short video streaming in practice). Essentially, this yields a 4-objective MORL problem. For the remaining objectives, we chose equal weight 0.2 for each of them to highlight no particular preference within this subset of objectives.
> > >
> > > In this rebuttal period, to further illustrate our WC-MOAC framework's effectiveness in exploring the Pareto stantionarity front, we have conducted additional experiments with more variations of the $\mathbf{p}$ vector, some of which have placed more emphasis on "Comment" and "Dislike" as you suggested. Please see the new $\mathbf{p}$ vector setting in Table 3 and the associated new plots in Fig. 2 in the revised manuscript. Here, the settings include larger weights for "Comment" and "Dislike" than those in Table 1 following your suggestion, and we thank the reviewer's comment that helps enrich our experiments.
> > >
> > > ______________
> > >
> > > >**Follow-up Comment 3:** As replied to Comments 7 and 8, the preference vector is not trained. What is the systematic way of deciding the weighting in your approach? As TSCAC, does your method also require domain knowledge?
> > >
> > > **Our Response:** Thanks for your comments and questions. Please see our point-to-point responses below:
> > >
> > > 1. **Systematic Way of Weighting:** To systematically explore the Pareto stationary solutions, the decision-maker can simply enumerate and choose the vector $\mathbf{p}$ in the $M$-dimensional standard simplex (i.e., the set $\{ p_i\geq 0, i=1,\ldots,M | \sum_{i=1}^{M} p_i =1\}$). For example, one possible approach will be using a grid search on the $M$-dimensional standard simplex for trying different $\mathbf{p}$ vectors. The chosen $\mathbf{p}$ vector is then taken by Algorithm 1 as input.
> > >
> > > 2. **Not Requiring Domain Knowledge:** Unlike TSCAC, our approach does **not** require domain knowledge, which is actually a *salient feature* of our proposed method. The reason is that our goal is to explore the Pareto stationarity front with as many $\mathbf{p}$-vector choices as possible. This implies that we are at the *"exact oppositie"* of getting domain knolwedge to determine $\mathbf{p}$.
> > >
> > > Additionally, as we mentioned above, we have added new empirical results in this rebuttal period with more weight vectors $\mathbf{p}$ in Appendix A.2 Section. In Figure 1\(c) and Figure 2(b), it can be seen that WC-MOAC can explore Pareto (stationary) solution in that has smaller dislike values (or equivalently, higher negative dislike values) than SDMGrad. Also from Figure 1\(c) and 2(b), in terms of "Comment" objective, WC-MOAC explore Pareto (stationary) solution that is no worse than SDMGrad.
> > > ______________
> > >
> > > >**Follow-up Comment4:** Meanwhile, for the experiment shown in Figure 1, concentrating on click, dislike, and watch do not differ much.
> > >
> > > **Our Response:** Thanks for your comment. However, we are not exactly sure if we fully understand this comment. If our guess is correct, we suspect there may be some misunderstanding in reading Figure 1(a), where the results of "Click," "Dislike," and "WatchTime" are quite similar under different $\mathbf{p}$ vectors. We would like to clarify Figure 1(a) in here. First of all, Figure 1(a) is **not** showing the performance of our proposed WC-MOAC method. Rather, Figure 1(a) illustrates the performance of SDMGrad over varying weight vector $\mathbf{p}$. We note that SDMGrad is a **baseline method** for comparison, which is based on linear scalarization. The fact that the SDMGrad method concentrates (i.e., having a small footprint) around "WatchTime", "Like" and some concentrations on "Click" in Figure 1(a) shows exactly it is **not** suitable for Pareto stationarity front exploration.
> > >
> > > On the other hand, **our proposed WC-MOAC method** is shown in Figure 1(b), which shows that WC-MOAC explores a much **larger** Pareto sationary solution footprint without much concentration in all objectives. For ease of comparison, in Figure 1\(c), we compared the explored footprints from these two methods to compare the Pareto front explorations. Again, it confirms that WC-MOAC explores significantly more Pareto solutions.
> > > ______________

---

### Official Review · Reviewer_P2zK · 2024-11-06

**Soundness:** 2
**Presentation:** 2
**Contribution:** 3
**Rating:** 6
**Confidence:** 3

**Summary:**

This paper presents the weighted-Chebyshev multi-objective actor-critic (WC-MOAC) algorithm to address the challenge of systematically exploring Pareto-stationary solutions in multi-objective reinforcement learning under multiple non-convex reward objectives. The WC-MOAC algorithm is designed to combine multi-temporal-difference (TD) learning in the critic phase with weightedChebyshev scalarization and multi-gradient descent in the actor phase, effectively managing the complexities of non-convexity and scalability in multi-objective problems. The primary theoretical result of this work is a finite-time sample complexity bound, $O\left(\epsilon^{-2} p_{\min }^{-2}\right)$, which is independent of the number of objectives $M$, where $p_{\min }$ denotes the minimum entry of the weight vector $p$ in the scalarization. This independence from $M$ is a notable theoretical advance. The empirical evaluation, conducted on the KuaiRand offline dataset, indicates that the WC-MOAC algorithm surpasses baseline methods in performance, highlighting its robustness and potential for realworld applications in multi-objective reinforcement learning.

**Strengths:**

I feel the paper's originality comes from its novel application of the weighted-Chebyshev scalarization technique within the framework of multi-objective reinforcement learning, accompanied by theoretical guarantees on finite-time sample complexity. This is particularly noteworthy given the limited prior work addressing non-convex reward objectives in this field. The technical rigor is evident, as the authors present a well-founded theoretical analysis that establishes finite-time sample complexity guarantees. This analysis is both rigorous and practically relevant, especially in environments that require systematic exploration of Pareto-stationary solutions. The empirical validation further strengthens the contribution, with experimental results on the KuaiRand dataset showing that the proposed algorithm consistently outperforms baseline approaches. This demonstrates the algorithm's robustness and effectiveness in practical scenarios. Additionally, the research fills an important gap in multi-objective reinforcement learning, with potential applications across various domains requiring multi-objective optimization. Its relevance to real-world problems enhances the significance of the work.

**Weaknesses:**

While the paper provides a comprehensive theoretical foundation, certain methodological aspects could be clarified further. For example, the integration of the multi-gradient descent update, which computes a dynamic weighting vector $\lambda_t$ that balances exploration with convergence, could benefit from a more detailed discussion on its rationale and practical implementation steps. Additionally, the empirical evaluation is limited to a single dataset, the KuaiRand offline dataset, which raises questions about the algorithm's generalizability. Expanding the experimental analysis to include diverse datasets or multi-objective environments would provide deeper insights into the algorithm's robustness across varied applications.

**Questions:**

The paper would benefit from further clarification on several key points. First, could the authors discuss the impact of the preference vector $p$ on the algorithm's performance, especially in environments with highly non-convex reward landscapes? As the convergence rate depends on $p_{\min }$, understanding the selection of $p$ would be useful for real-world applications. Additionally, what limitations, if any, might arise in adapting the proposed algorithm to online settings, given that the study's empirical evaluation is restricted to offline data? The paper also raises questions regarding the sensitivity of the algorithm's performance to variations in $p$ and whether there are recommended heuristics for selecting optimal values. Lastly, in relation to multi-temporaldifference learning, were there specific stability challenges encountered, particularly when combining it with weighted-Chebyshev updates? If so, what mechanisms or parameters were introduced to maintain convergence stability?

**Details Of Ethics Concerns:**

N/A.

---

> ### Author Response · Authors · 2024-11-22
> **Reponse to Reviewer P2zK's Comments:**
>
> >**Your Comment 1 (Also Weakness 1):** While the paper provides a comprehensive theoretical foundation, certain methodological aspects could be clarified further. For example, the integration of the multi-gradient descent update, which computes a dynamic weighting vector $\lambda_t$ that balances exploration with convergence, could benefit from a more detailed discussion on its rationale and practical implementation steps.
>
> **Our Response:** Thanks for your comments. In terms of computing weight vector $\lambda_t$, it is obtained from solving Eq. (10), whose solution $\hat{\lambda}^{*}_t$ is then used in the momentum computation in Eq.(11). In Eq.(10), the $\lambda_t$-solution balances two aspects:
>
> 1) the first term corresponds to multiple-gradient descent approach, which is an approach to ensure achieving Pareto stationarity upon convergence.
>
> 2) The second term $\lambda^{\top}(\mathbf{P}\odot(\mathbf{J}^{*}_{ub}-\mathbf{J}(\theta)))$ corresponds to the weighted-Chebyshev scarlization formulation, which induces Pareto stationarity front exploration as explained in the paper.
>
> 3) The hypber-parameter $u>0$ is used to balance the trade-off between i) achieving a Pareto stationary solution and ii) systematically exploring the Pareto stationarity front.
>
> ________________
> >**Your Comment 2 (Also Weakness 2):** Additionally, the empirical evaluation is limited to a single dataset, the KuaiRand offline dataset, which raises questions about the algorithm's generalizability. Expanding the experimental analysis to include diverse datasets or multi-objective environments would provide deeper insights into the algorithm's robustness across varied applications.
>
> **Our Response:** Thank you for your suggestions. In this rebuttal period, we will provide more experimental settings. We are still working hard on conducting further experiments and will provide you an upate as soon as we are finished. Unfortunately, adding more datasets during this rebuttal period is quite challenging due to i) the limited amount of time and ii) the KuaiRand is the only suitable large-scale dataset for MORL that we can find at the current stage. We hope these resource limitations will not negatively affect the reviewer's evaluation toward our work.
>
> ________________
>
>
> >**Your Comment 3 (Also Question 1):** The paper would benefit from further clarification on several key points. First, could the authors discuss the impact of the preference vector $\mathbf{p}$ on the algorithm's performance, especially in environments with highly non-convex reward landscapes?
>
> **Our Response:** We thank the reviewer for this insightful question. For settings with non-convex accumulated reward landscapes (i.e., $J^{i}(\theta)$ is nonconvex in variable $\theta$ for some reward function $i\in [M]$), our algorithm converges to a Pareto stationary point for *any* given weight vector $\mathbf{p}$ suggested by Theorem 4. Note that Pareto stationarity is a necessary condition for Pareto optimality. Similar to single-objective non-convex optimization for machine learning problems, it is often acceptable to find a Pareto stationary solution in the non-convex multi-objective settings.
>
> ________________
> >**Your Comment 4 (Also Question 2):** As the convergence rate depends on $p_{\min}$, understanding the selection of $p$ would be useful for real-world applications.
>
> **Our Response:** Thanks for your comments and questions. Our Theorem 4 suggests that the smaller  miminum enntry in vector $\mathbf{p}$ (i.e. smaller $p_{\min}$-value), the slower the convergence is. This implies that it might take more iterations to explore the Pareto stationarity front with a smaller $p_{\min}$-value.
> On the other hand, in order to systematically explore the Pareto stationary solutions, the user can vary and enumerate the vector $\mathbf{p}$ in the $M$-dimensional standard simplex, which is then taken by Algorithm 1 as input. For example, one possible approach will be using a grid search for trying different $\mathbf{p}$ vectors.
>
> ________________
> >**Your Comment 5 (Also Question 3):** Additionally, what limitations, if any, might arise in adapting the proposed algorithm to online settings, given that the study's empirical evaluation is restricted to offline data?
>
> **Our Response:** In fact, the proposed Algorithm 1 is an online algorithm. Since KuaiShou dataset used in the empirical studies is an offline dataset, we have adapted our algorithm to the offline MORL setting. In principle, Algorithm 1 should perform better in online settings due to the fact that, in online setting, the algrotihm can sample more diverse data to evaluate and improve upon the current policy. In contrast, in offline MORL, the algorithm needs to work on the online dataset collected by a behavior policy, which may only have a limtied coverage.
>
> ________________

---

> > ### Author Response · Authors · 2024-11-22
> > **Reponse to Reviewer P2zK's Comments (Continued):**
> >
> > >**Your Comment 6 (Also Question 4):** The paper also raises questions regarding the sensitivity of the algorithm's performance to variations in $p$ and whether there are recommended heuristics for selecting optimal values.
> >
> > **Our Response:** Thanks for your comments and suggestions. We are currently working hard on running new simulations. We will attach the new results as soon as we are finished.
> >
> > ________________
> > >**Your Comment 7 (Also Question 5):** Lastly, in relation to multi-temporal difference learning, were there specific stability challenges encountered, particularly when combining it with weighted-Chebyshev updates? If so, what mechanisms or parameters were introduced to maintain convergence stability?
> >
> > **Our Response:** Thanks for your questions. First of all, we want to get a clarification on what the "stability" is referring to in the reviewer's first question. If it's referring to convergence, then we have following response:
> >
> > In each iteration of the critic component, the multiple-temporal difference (TD) learning is evaluating the current *stationary policy* on all $M$ objectives. Due to the fact that given a state-action pair $(s,a)\in\mathcal{S}\times\mathcal{A}$, the reward from objective $i$ (i.e. $r^{i}(s,a)$) and the reward from objective $j$ (i.e., $r^{j}(s,a)$) is independent of each other for any $i,j\in [M]$ and $i\neq j$, we can utilize existing single-objective TD learning analysis, where a negative Lyapunov drift argument can be constructed to ensure convergence (i.e., achieving stability).
> >
> > If the terminology "stability" in the reviwer's question means "robustness under non-stationary policies" (e.g., non-stationary polices required in RL problems under partially observed Markov decision processes (POMDP)), then this remains a very challenging open problem, which deserves a dedicated paper on this topic. We are very interested in this open problem in our future studies and we thank the reviewer for pointing out this research direction.

---

> > > ### Author Response · Authors · 2024-11-26
> > > **Our Follow-up Response to Comment 6**
> > >
> > > >**Your Comment 6 (Also Question 4):** The paper also raises questions regarding the sensitivity of the algorithm's performance to variations in $p$ and whether there are recommended heuristics for selecting optimal values.
> > >
> > > **Our Follow-up Response to Comment 6(Also Question 4):** Thanks for your patience. We have finished additional experiments with more variations of $\mathbf{p}$. We have attached radar charts that include new results in Appendix A.2 Section. Specifically, in addtion to the 5 one-hot vectors, we have included $\mathbf{p}$ vectors that takes the following values:
> > > | click | like | comment | dislike | watchtime |
> > > | -------- | ------- |------- |------- |------- |
> > > | 0.85 | 0.05 | 0.05 | 0 | 0.05 |
> > > | 0.7 | 0.1 | 0.1 | 0 | 0.1 |
> > > | 0.55 | 0.15 | 0.15 | 0 | 0.15 |
> > > | 0.4 | 0.2 | 0.2 | 0 | 0.2 |
> > > |0.05 | 0.05 | 0.85 | 0.0001 | 0.05 |
> > > |0.10 | 0.10 | 0.70 | 0.0001 | 0.10 |
> > > |0.15 | 0.15 | 0.55 | 0.0001 | 0.15 |
> > >
> > > From the empirical results in Figure 2(a) in Appendix A.2 Section, we can see that with additional weight vectors $\mathbf{p}$, WC-MOAC is exploring **more Pareto stationary solutions** compared to WC-MOAC with only one-hot vectors as the weight vectors. In Figure 2(b), it further shows that with more $\mathbf{p}$ vectors, WC-MOAC explores even wider Pareto footprints. This further confirms our theoretical prediction as well as strengthens the empirical observation that, with increasing number of weight/explore vectors $\mathbf{p}$, WC-MOAC possess the potential to explore more Pareto stationary points.
> > >
> > > With the above new experimental results on more variations of $\mathbf{p}$, we would like to add an additional remark that the terminology "sensitivity" in the reviewer's Comment 6 is somewhat irrelavent to the above experiments with additional $\mathbf{p}$ varations. The reason is that sensitivity is typically referring to the cases with a target choice of $\mathbf{p}$, but there could be imperfection/inaccuracy/errors in picking such a $\mathbf{p}$ in practice. Thus, one would like to study how "sensitive" the system will be affected by the deviation of the real choice of the $\mathbf{p}$ from the target choice of $\mathbf{p}$. However, in this work, we do *not* have any target choice of $\mathbf{p}$. Instead, the goal of our WC-MOAC method is to explore Pareto Stationary front *by applying as many different $\mathbf{p}$ vectors in the algorithm as possible*. Hence, our new results above are showing "exploration capabiltiy" rather than "sensitivity."
> > >
> > >
> > > Lastly, regarding how to choose $\mathbf{p}$ to systematically explore the Pareto stationary solutions, the decision-maker can simply enumerate and choose the vector $\mathbf{p}$ in the $M$-dimensional standard simplex (i.e., the set $\{ p_i\geq 0, i=1,\ldots,M | \sum_{i=1}^{M} p_i =1\}$). For example, one possible approach will be using a grid search on the $M$-dimensional standard simplex for trying different $\mathbf{p}$ vectors. The chosen $\mathbf{p}$ vector is then taken by Algorithm 1 as input.
> > >
> > > We hope that our clarifications above have addressed the reviewer's question on the selection of $\mathbf{p}$. Please let us know if any question remains.

---

> > > > ### Comment · Reviewer_P2zK · 2024-11-27
> > > >
> > > > Thank you for your detailed responses and additional experiments. However, I encourage the authors to address the integrity issues raised by Reviewer P7fw and similar concerns of Reviewer JPv7 to ensure a more comprehensive and novel submission.

---

> > > > > ### Author Response · Authors · 2024-11-27
> > > > > **Follow-up Response to Reviewer P2zK**
> > > > >
> > > > > >**Your Comment:** I encourage the authors to address the integrity issues raised by Reviewer P7fw and similar concerns of Reviewer JPv7 to ensure a more comprehensive and novel submission.
> > > > >
> > > > > **Our Response:** Thanks for your comment. Yes, we have already carefully responsed to all the comments to Reviewer P7fw and Reviewer JPv7. As the integrity accusation is a very serious issue, we have taken serious effort to address their comments. We strongly encourage reviewer P2zK to go through our point-to-point responses to gain an independent assessment. Meanwhile, we provide the following summary to the above two reviewers' response in here:
> > > > >
> > > > > The primary concern only revolves around perceived *cosmetic* similarities between our current submission and (Zhou et al., 2024). Specifically, we want to empahsize the **critical differences** between our work and (Zhou et al., 2024) in the following aspects:
> > > > >
> > > > > **1) Differences in Research Goals:** Our research focuses on systematically exploring the Pareto stationarity front in a multi-objective problem using a weighted-Chebyshev formulation. In comparison, the goal of (Zhou et al., 2024) is to achieve only a Pareto stationary solution rather than exploring the full Pareto stationarity front.
> > > > >
> > > > > **2) Differences in Algorithmic Designs:** In this paper, we have integrated a weighted-Chebyshev technique with the multi-gradient descent approach (MGDA) for multi-objective optimization, a technique entirely absent in (Zhou et al., 2024). Our paper devotes Section 4.2 (about 1.5 pages) to describing this approach in detail, illustrating how we use the weighted-Chebyshev method to achieve systematic exploration. This technique leverages an exploration/weight vector $\mathbf{p}$, extensively explained in our derivations. In contrast, (Zhou et al., 2024) is based only on a direct adaptation from MGDA. As a result, there is *nothing* in (Zhou et al., 2024) that corresponds to Section 4.2 in this paper.
> > > > >
> > > > > **3) New and Unique Theoretical Contributions:** Our Theorem 4 and Corollary 5 highlight the effects of the weight vector $\mathbf{p}$, particularly through its minimum entry $p_{\min}$, on convergence rate to Pareto stationarity and corresponding sample complexities for each exploration. While the convergence rate results have similar convergence rate order, the coefficients in Theorem 4 are distinctly different from those in (Zhou et al., 2024). Again, merely having the same convergence rates does **not** mean the algorithms and the analysis are the same. For example, both the gradient descent (GD) method and the SVRG stochastic gradient method have the same $O(1/T)$. Clearly, calling SVRG and GD the same method is absurd. Therefore, we don't believe any sensible researchers would think Theorem 4 and Corollary 5 are copying from (Zhou et al., 2024).
> > > > >
> > > > > **4) Differences in Experimental Results:** In Figure 1 and Figure 2, we have presented the Pareto footprint comparisons between our proposed algorithm and the conventional linear scalarization approach (i.e., comparing the Pareto front explored by our method via varying exploration/weight vector $\mathbf{p}$ with that achieved by the conventional linear scalarization approach). In contrast, there is no such experimental comparistive studies in (Zhou et al., 2024).
> > > > >
> > > > > Given the serious nature of the accusations, we again strongly encourage reviewer P2zK to go through the point-to-point responses to both reviewers to gain an independent assessment. **We are also happy to answer any further questions Reviewer P2zK may have in this aspect**. Thanks!

---

### Official Review · Reviewer_n72g · 2024-11-07

**Soundness:** 3
**Presentation:** 3
**Contribution:** 4
**Rating:** 6
**Confidence:** 4

**Summary:**

This paper investigates multi-objective reinforcement learning (MORL) using a weighted-Chebyshev multi-objective actor-critic (WC-MOAC) framework. The authors implement a multi-temporal-difference (TD) learning method for the critic and apply a weighted-Chebyshev multi-gradient-descent algorithm (MGDA) for the policy gradient in the actor. The algorithm is shown to achieve a sample complexity of $\mathcal{O}(\epsilon^{-2})$ to reach an $\epsilon$-Pareto stationary point. The authors also provide empirical validation using a real-world dataset.

**Strengths:**

The paper is well written with clear motivation and explanation. The authors’ theoretical analysis provides an important contribution to MORL. I would suggest the paper be accepted after minor revision.

**Weaknesses:**

Some of the details are not clear enough and can be improved. I have put them in Questions.

**Questions:**

1.	Page 5 eq (2). On the left of “:=” is a function of x, while on the right you have a scalar. Please make the definition consistent.
2.	Page 5 Lemma 1. You do not provide a proof for Lemma 1. Please reference the exact theorem or proposition from the cited work that supports Lemma 1.
3.	Lemma 1 states existence of the vector p, which is directly related to a weak Pareto-optimal. However, it is not clear how $p$ is determined in practice within the algorithm. Does Theorem 4 hold for arbitrary p or for the specific p provided in Lemma 1? It would be helpful to include a more detailed explanation of p’s role in the algorithm and whether it influences the theoretical guarantees.
4.	Page 7. It will be clearer if you specify what arguments you are optimizing. In eq (7), could you clarify whether the optimization is over $\lambda$, or both $\lambda$ and $\theta$? It appears from my understanding that eq (9) optimizes over both $\lambda$ and $\theta$, whereas eq (10) involves only $\lambda$.
5.	Page 7 line 340, “gradient” should be “Jacobian”.
6.	Page 8 Def 3 line 419. $\lambda$ is already given. Why do you have $\min_{\lambda}$?
7.	Page 9 Theorem 4. Some of the quantities, such as $\lambda_A$, $L_J$, $R_w$, $\bf{\gamma}$ are from the appendix, it is better to add some explanation for these quantities in the statement of the theorem.
8.	Page 9 line 467. You mention that state-of-the-art sample complexity for single-objective RL is $\mathcal{O}(\epsilon^{-2})$ bu Xu et al. And your Corollary 5 achieves the same complexity for MORL, which is a great achievement. Actually, under some special structure like linear quadratic problem, the one can show a sample complexity of $\mathcal{O}(\epsilon^{-1})$, as proved by Zhou and Lu in “Single Timescale Actor-Critic Method to Solve the Linear Quadratic Regulator with Convergence Guarantees”. Please consider adding it as a remark to enhance completeness of the paper.

---

> ### Author Response · Authors · 2024-11-22
> **Response to Reviewer n72g's Comments**
>
> >**Your Comment 1:** Page 5 eq (2). On the left of “:=” is a function of x, while on the right you have a scalar. Please make the definition consistent.
>
> **Our Response:** Thank you for pointing out this typo. We will revise the definition as follows in the revision:
> $$WC_{\mathbf{p}}(\mathbf{F}(\cdot)) := \min_{\mathbf{x}}\max_{i\in[M]}\{ p_i f_i(\mathbf{x})\}= \min_{\mathbf{x}} \\| \mathbf{p} \odot \mathbf{F}(\mathbf{x})  \\|_{\infty}.$$
>
> ________________
> >**Your Comment 2:** Page 5 Lemma 1. You do not provide a proof for Lemma 1. Please reference the exact theorem or proposition from the cited work that supports Lemma 1.
>
> **Our Response:** Thanks for your comments. Lemma 1 is cited from Proposition 4.7 and its proof on Page 42 in (Qiu et al. (2024)). Specifically, on Page 42 of the proof of Proposition 4.7, Part 1) and Part 2) collectively provide an "if-and-only-if" statement in Lemma 1 of this paper. We will provide more clarification on this in our revision.
>
> Qiu, S., Zhang, D., Yang, R., Lyu, B., & Zhang, T. (2024). Traversing pareto optimal policies: Provably efficient multi-objective reinforcement learning. arXiv preprint arXiv:2407.17466.
> ________________
> >**Your Comment 3:** Lemma 1 states existence of the vector $\mathbf{p}$, which is directly related to a weak Pareto-optimal. However, it is not clear how $\mathbf{p}$ is determined in practice within the algorithm. Does Theorem 4 hold for arbitrary p or for the specific p provided in Lemma 1? It would be helpful to include a more detailed explanation of p’s role in the algorithm and whether it influences the theoretical guarantees.
>
> **Our Response:** Thanks for your question. Again, we would like to emphasize that Lemma 1 is an "if-and-only-if" statement, which implies that there is a one-to-one mapping between a WC-solution and a Pareto-optimal solution and further suggest a systematical way to explore the Pareto-front (assuming the WC-scalaization problem can be solved to optimality). As a result, when applying Algorithm 1, one does *not* need to carefully pick the vector $\mathbf{p}$. Rather, the decision-maker is supposed to vary and enumerate the vector $\mathbf{p}$ in the $M$-dimensional standard simplex, which is then taken by Algorithm 1 as input to systematically explore the Pareto front. Also, Theorem 4 holds for arbitrary $\mathbf{p}$ whose entries are strictly positive (but one would typically use $\mathbf{p}$ in the $M$-dimensional standard simplex to avoid arbitrariness in scaling). For those $\mathbf{p}$-vectors with some entries being 0, one can simply discard those zero entries and reformulate the problem in a multi-objective RL problem with lower dimension, which only retains those non-zero entries in the original $\mathbf{p}$.
> ________________
> >**Your Comment 4:** Page 7. It will be clearer if you specify what arguments you are optimizing. In eq (7), could you clarify whether the optimization is over $\mathbf{\lambda}$, or both $\lambda$ and $\mathbf{\theta}$? It appears from my understanding that eq (9) optimizes over both $\lambda$ and $\theta$, whereas eq (10) involves only $\lambda$.
>
> **Our Response:** Thanks for your question. In Eq.(7), we are optimizing over $\lambda$ for any given $\theta$, where $\pi_{\theta}$ denotes the current policy. Similarly, in Eq.(9), we are also optimizing over $\lambda$ for any given $\theta$. The rationale is that, given the current policy evaluation from the critic component, we want to leverage and integrate multi-policy-gradient descent (first term in Eq.(10)) and the weighted-Chebyshev scalarization (second term in Eq.(10)) to guide the policy update direction to achieve a balance between i) converging to a Pareto stationary solution and 2) systematically exploring the Pareto stationarity front.
>
> ________________
> >**Your Comment 5:** Page 7 line 340, “gradient” should be “Jacobian”.
>
> **Our Response:** Thanks for your question. Here, matrix $\mathbf{G}$ we are referring to is
> $$\mathbf{G}=-[\nabla_{\theta} J^{1}(\theta), \cdots, \nabla_{\theta} J^{i}(\theta), \cdots, \nabla_{\theta} J^{M}(\theta)].$$
> In other words, each column $i\in [M]$ is the gradient of $J_{\text{ub}}^{i,\*}-J^{i}(\theta)$ with respect to $\mathbf{\theta}$, where $J_{\text{ub}}^{i,\*}$ is an upper bound constant. Hence, $\mathbf{G}$ is related to but not exactly the Jacobian (more precisely, it is negative Jacobian matrix). We will further clarify this in our revision.
>
> ________________
>
> >**Your Comment 6:** Page 8 Def 3 line 419. $\lambda$ is already given. Why do you have $\min_{\lambda}$?
>
> **Our Response:** Thank you for pointing out this typo out. We will revise it in revision and change it to $\\| \nabla_{\theta} \mathbf{J}_{\theta}(\theta) \lambda \\|_2^2 \leq \epsilon$.
>
> ________________

---

> > ### Author Response · Authors · 2024-11-22
> > **Response to Reviewer n72g's Comments (Continued)**
> >
> > >**Your Comment 7:** Page 9 Theorem 4. Some of the quantities, such as $\lambda_A$, $L_J$, $R_{w}$, $\gamma$ are from the appendix, it is better to add some explanation for these quantities in the statement of the theorem.
> >
> > **Our Response:** Thank you for the suggestion. $\lambda_A$ is defined in Line 771 and this quantity is introduced to ensure the convergence for the critic component for all policies. $R_w$ is defined in Lemma Line 782 for the average reward setting and in Line 805 for the discounted setting, which is an upper bound constant for $\ell_2$ norm for the critic parameters. $L_J$ is the Lipschitz constant defined in Line 425 in Assumption 3. $\mathbf{\gamma}=(\gamma^1,\cdots,\gamma^M)^{\top}$ is a vector that concatenates all discount factors $\gamma^1$ to $\gamma^M$ for all objectives in discounted setting. We will provide these clarifications in our revision.
> >
> > ________________
> > >**Your Comment 8:** Page 9 line 467. You mention that state-of-the-art sample complexity for single-objective RL is  bu Xu et al. And your Corollary 5 achieves the same complexity for MORL, which is a great achievement. Actually, under some special structure like linear quadratic problem, the one can show a sample complexity of , as proved by Zhou and Lu in “Single Timescale Actor-Critic Method to Solve the Linear Quadratic Regulator with Convergence Guarantees”. Please consider adding it as a remark to enhance completeness of the paper.
> >
> > **Our Response:** Thank you for pointing out this excellent reference. We are more than happy to add discussion on sample complexity of the speacial structured RL problems in the revision as you suggested.

---

### Official Review · Reviewer_P7fw · 2024-11-10

**Soundness:** 2
**Presentation:** 3
**Contribution:** 1
**Rating:** 1
**Confidence:** 5

**Summary:**

This paper proposes weighted-Chebyshev multi-objective actor-critic (WC-MOAC) to find Pareto-stationary policies for MORL with sample complexity guarantees. The proposed WC-MOAC combines two techniques, namely the (stochastic) multiple gradient approach and the momentum-based updates of the dual variables. The WC-MOAC has a sample complexity of $\tilde{O}(\epsilon^{-2})$ in finding an epsilon Pareto-stationary policy. Finally, simulation results on an offline dataset is provided.

**Strengths:**

- The obtained sample complexity is independent of the number of objective functions and has a nice scaling with $M$ (However, there is a severe concern stated below).
- This paper handles two formulations, namely, discounted-reward and average-reward MDPs, simultaneously (despite that the algorithm and the analysis are agnostic to the reward setting in principle).
- Overall the paper is well-organized and easy to follow, with the concepts, definitions, and theoretical results clearly explained.

**Weaknesses:**

The major issue with this paper is that it is very similar to the paper “Finite-Time Convergence and Sample Complexity of Actor-Critic Multi-Objective Reinforcement Learning” by (Zhou et al., 2024) published in ICML 2024, in terms of both the algorithm design, paper writing, and the claimed contributions. The flow of the paper exactly follows (Zhou et al., 2024), and most of the paragraphs are just paraphrased versions of (Zhou et al., 2024). For example:

- The pseudo code of WC-MOAC are almost the same (almost verbatim) as those of the MOAC algorithm (cf. Algorithms 1 and 2 in (Zhou et al., 2024)).
- The theoretical result of WC-MOAC in Theorem 4 and Corollary 5 appear almost the same as the Theorem 5 and Corollary 6 in (Zhou et al., 2024).
- The 1st  paragraph of Introduction (Lines 34-46) appears to be paraphrased from the first two paragraphs of the Introduction of (Zhou et al., 2024).
- The 2nd paragraph of Introduction (Lines 47-64) appears to be paraphrased from the third paragraph of the Introduction of (Zhou et al., 2024).
- The paragraphs about the “Key Contributions” (Lines 97-124) appear to directly follow the “Main Contributions” of the Introduction of (Zhou et al., 2024).
- In Section 3, the problem formulation about MOMDP (Lines 181-187) appears very similar to the second paragraph of Section 3.1 of (Zhou et al., 2024).
- The part on “Learning Goal and Optimality in MORL” (Lines 200-215) appears almost the same as the “Problem Statement” in Section 3.1 of (Zhou et al., 2024). Moreover, even the footnote #3 in this paper almost goes verbatim compared to the footnote #1 in (Zhou et al., 2024).
- The preliminaries about the policy gradient for MORAL (Lines 274-296) also largely resembles Section 3.2. Specifically, several sentences about Lemma 2 and Assumption 2 of this paper are exactly the same as those in Lemma 1 and Assumption 2 of (Zhou et al., 2024).
- The description about Assumption 3 and Lemma 3 in this paper (Lines 424-437) appears to exactly follow the Assumption 3 and Lemma of (Zhou et al., 2024).

**Questions:**

In addition to the above, here are some further technical questions:
- Technically: One of my main technical concerns is the motivation for finding a Pareto-stationary policy (under the assumption that the state and action spaces are finite) in the specific context of MORL. Specifically, while it is indeed difficult to find the whole Pareto front in MORL, it is actually not hard to find one or some Pareto-optimal policies by adapting reducing MORL to single-objective RL and finding the convex coverage set (e.g., (Yang et al., 2019)). For example, based on (Chen and Magulari, 2022), one can use a policy-based method with off-policy TD learning (under linear function approximation) to find an epsilon-optimal solution for single-objective RL with a sample complexity of $\tilde{O}(1/\epsilon^2)$. There are also several other recent works like (Lan 2021; Fatkhullin et al., 2023; Liu et al., 2020; Chen et al., 2022) that can find an epsilon-optimal policy with sample complexity guarantees. To adapt these results to MORL, one can use linear scalarization and thereby find one Pareto-optimal policy (specific to some preference vector). As a result, it remains not totally clear why it is theoretically appealing to design an algorithm for finding only a Pareto-stationary policy if we can already find Pareto-optimal policies (despite that Pareto-stationarity is indeed a widely adopted concept in the MOO literature).

- Another concern is the novelty in terms of algorithm and convergence analysis. Specifically, the WC-MOAC algorithm appears to be a direct application of the MOAC algorithm (Zhou et al. 2024) and also similar to the MOO algorithm CR-MOGM of (Zhou et al. 2022), which is the enhanced (stochastic) MGDA method (e.g., (Desideri, 2012; Liu and Vicente, 2021)) with the momentum update of the dual variable vector, to the setting of MORL (with the multi-objective critic learned by standard TD updates). Under a properly learned critic, then the stochastic multiple gradients can have a sufficiently low bias such that it enables similar convergence guarantees as in the general MOO. This is also shown in Theorem 11 (as a direct result of Lemma 10). As a result, the sample complexity and the convergence analysis of WC-MOAC essentially resemble those of CR-MOGM in (Zhou et al. 2022) for the general non-convex case, cf. Theorem 3 and Appendix E of (Zhou et al. 2022).

References:
- (Yang et al., 2019) Runzhe Yang, Xingyuan Sun, and Karthik Narasimhan, “A Generalized Algorithm for Multi-Objective Reinforcement Learning and Policy Adaptation,” NeurIPS 2019.
- (Zhou et al., 2022) Shiji Zhou, Wenpeng Zhang, Jiyan Jiang, Wenliang Zhong, Jinjie Gu, Wenwu Zhu, “On the Convergence of Stochastic Multi-Objective Gradient Manipulation and Beyond,” NeurIPS 2022.
- (Chen and Magulari, 2022) Zaiwei Chen and Siva Theja Maguluri, “Sample Complexity of Policy-Based Methods under Off-Policy Sampling and Linear Function Approximation,” AISTATS 2022.
- (Lan 2021) Guanghui Lan, “Policy Mirror Descent for Reinforcement Learning: Linear Convergence, New Sampling Complexity, and Generalized Problem Classes,” Mathematical programming, 2021.
- (Fatkhullin et al., 2023) Ilyas Fatkhullin, Anas Barakat, Anastasia Kireeva, Niao He, “Stochastic Policy Gradient Methods: Improved Sample Complexity for Fisher-non-degenerate Policies,” ICML 2023.
- (Liu et al., 2020) Yanli Liu, Kaiqing Zhang, Tamer Basar, Wotao Yin, “An Improved Analysis of (Variance-Reduced) Policy Gradient and Natural Policy Gradient Methods,” NeurIPS 2020.
- (Chen et al., 2022) Zaiwei Chen, Sajad Khodadadian, and Siva Theja Maguluri, "Finite-sample analysis of off-policy natural actor–critic with linear function approximation," IEEE Control Systems Letters, 2022.

**Details Of Ethics Concerns:**

One major issue with this paper is that it is very similar to the paper “Finite-Time Convergence and Sample Complexity of Actor-Critic Multi-Objective Reinforcement Learning” by (Zhou et al., 2024) published in ICML 2024, in terms of both the algorithm design, paper writing, and the claimed contributions. The major issue with this paper is that it is very similar to the paper “Finite-Time Convergence and Sample Complexity of Actor-Critic Multi-Objective Reinforcement Learning” by (Zhou et al., 2024) published in ICML 2024, in terms of both the algorithm design, paper writing, and the claimed contributions. The flow of the paper exactly follows (Zhou et al., 2024), and most of the paragraphs are just paraphrased versions of (Zhou et al., 2024). For example:

- The pseudo code of WC-MOAC are almost the same (almost verbatim) as those of the MOAC algorithm (cf. Algorithms 1 and 2 in (Zhou et al., 2024)).
- The theoretical result of WC-MOAC in Theorem 4 and Corollary 5 appear almost the same as the Theorem 5 and Corollary 6 in (Zhou et al., 2024).
- The 1st  paragraph of Introduction (Lines 34-46) appears to be paraphrased from the first two paragraphs of the Introduction of (Zhou et al., 2024).
- The 2nd paragraph of Introduction (Lines 47-64) appears to be paraphrased from the third paragraph of the Introduction of (Zhou et al., 2024).
- The paragraphs about the “Key Contributions” (Lines 97-124) appear to directly follow the “Main Contributions” of the Introduction of (Zhou et al., 2024).
- In Section 3, the problem formulation about MOMDP (Lines 181-187) appears very similar to the second paragraph of Section 3.1 of (Zhou et al., 2024).
- The part on “Learning Goal and Optimality in MORL” (Lines 200-215) appears almost the same as the “Problem Statement” in Section 3.1 of (Zhou et al., 2024). Moreover, even the footnote #3 in this paper almost goes verbatim compared to the footnote #1 in (Zhou et al., 2024).
- The preliminaries about the policy gradient for MORAL (Lines 274-296) also largely resembles Section 3.2. Specifically, several sentences about Lemma 2 and Assumption 2 of this paper are exactly the same as those in Lemma 1 and Assumption 2 of (Zhou et al., 2024).
- The description about Assumption 3 and Lemma 3 in this paper (Lines 424-437) appears to exactly follow the Assumption 3 and Lemma of (Zhou et al., 2024).

Based on the above, there could be some research integrity issues that would require further attention.

---

> ### Author Response · Authors · 2024-11-21
> **Response to Reviewer P7fw**
>
> We appreciate the reviewer's constructive comments and valuable insights. The detailed point-by-point responses are as follows:
>
> -------------
>
> > **Your Comment 1:** The pseudo code of WC-MOAC are almost the same (almost verbatim) as those of the MOAC algorithm (cf. Algorithms 1 and 2 in (Zhou et al., 2024)).
>
> **Our Response:** In this paper and (Zhou et al., 2024), the algorithms both utilized actor-critic framework, which is a widely used and standard RL framework. If merely following the actor-critic framework violates academic integridty, then we are not sure how the reviwer would view the thousands of papers published in the RL literatrue that adopted the actor-critic framework. Further, in terms of the algorithms, the following are the major differences between our work and (Zhou et al., 2024):
> 1) Algorithm 1 in this paper is presented as a single-oracle, which includes both Critic and Actor components; whereas in (Zhou et al., 2024), the critic component is presented as a subroutine, which is called in its main Algorithm 2.
> In this paper, we have leveraged a weight/exploration vector $\mathbf{p}$ to explore the Pareto stationarirty front (a necessary condition for Pareto Front). This is reflected by two major aspects (see next two points), which are not present in (Zhou et al., 2024).
>
> 2) In the second column of Line 395, $\hat{\mathbf{\lambda}}^{*}_t$ is the solution of Eq. (10), which is an objective function that carefully balances the Weighted-Chebyshev exploration controlled by the $\mathbf{p}$-weight vector and the mutpile-gradient descent term. If one were to check Eq. (10) in this paper and Eq. (9) in (Zhou et al. (2014)), it is apparent that they are completely **different**.
>
> 3) Leveraging a weight vector $\mathbf{p}$, the update for $\mathbf{g_t}=\mathbf{G_t}(\mathbf{p}\odot \mathbf{\lambda_t})$ in this paper, which enables the exploration of the Pareto front (also see Figure 1 for the exploration effect). In contrast, there is **no such Hadamard multiplied $\mathbf{p}$ term** for the policy update in (Zhou et al., 2024), which is in the form of $\mathbf{g}_{t}=\mathbf{G}_t(\mathbf{p}\odot \mathbf{\lambda}_t)$. We also note that the solution of Eq. (10) yields a completely different quantity from that of Eq. (9) in (Zhou et al. (2014)). As a reuslt of such a difference, the algorithm in (Zhou et al. (2024)) can only guarantee convergence to a Pareto stationary point, but lacks the capability of systematic Pareto stationarity exploration.
>
> 4) In the actor component of this paper, we require the computation of score function $\mathbf{\psi}_{t,l}$ by Eq.(12) (see approximately in Line 388 in the second column). This is clearly **not** in the Pseudo code of (Zhou et al., 2024).
>
> 5) In Algorithm 1, it is required that $w^{i}_k=w^{i}_t$ for synchronization of the critic parameters (see the first column in Line 398-399). Note that this is **absent** in (Zhou et al. 2024).
>
> 6) Among the input of Algorithm 1 (Lines 380-381), our algorithm requires a weighted/exploration vector $\mathbf{p}$ in addition to standard input parameters for actor-critic with linear approximations.
>
> Again, we want to emphasize that the actor-critic algorithmic framework in the RL literature will share a lot of structural similarities, including (Zhou et al. (2014)). This also indicates the popularity of the actor-critic framework for solving RL problems. The similar notations are due to the convention in the literature for the ease of understanding.

---

> > ### Author Response · Authors · 2024-11-21
> > **Response to Reviewer P7fw (Continued)**
> >
> > > **Your Comment 2:** The theoretical result of WC-MOAC in Theorem 4 and Corollary 5 appear almost the same as the Theorem 5 and Corollary 6 in (Zhou et al., 2024).
> >
> > **Our Response:** Similar to most theoretical work on RL, in Theorem 4 of our paper, we proved the convergence to the stationarity under each given $\mathbf{p}$-exploration vector. The key differences in the convergence results compared to (Zhou et al., 2024) stem from the effect of the $\mathbf{p}$-vector. This weight vector affects the convergence in a fashion that is inversely proportional to $p_{\min}^2$, where $p_{\min}$ is minimum entry of the weight vector. What this **new** result entails is that the convergence rate of exploring the Pareto stationary front (also the Pareto front under convex settings) could be affected by $p_{\min}$. In other words, the convergence rate could potentially be slower when $p_{\min}$ is small.
> > Furthermore, in Corollary 5, the sample complexity of the Pareto stationarity front exploration is also characterized by $p_{\min}$ when designing learning rate $\eta_t$. Specifically, by carefully setting the learning rate $\eta_t=\frac{p^{2}_{\min}}{t^{2}}$, one can achieve a $\mathbf{p}$-independent sample complexity. In contrast, in (Zhou et al., 2024), there is no such Pareto stationarity front exploration in the convergence their result. In addition, the goal in (Zhou et al., 2024) is focused on converging to only one of potentially infinite Pareto Stationary point, which is a much weaker goal compared to ours.
> >
> > ________________
> > > **Your Comments 3 and 4:** The 1st paragraph of Introduction (Lines 34-46) appears to be paraphrased from the first two paragraphs of the Introduction of (Zhou et al., 2024). The 2nd paragraph of Introduction (Lines 47-64) appears to be paraphrased from the third paragraph of the Introduction of (Zhou et al., 2024).
> >
> > **Our Reponse:** Essentially Comment 3 and Comment 4 are referring to the same blocks of paragrahphs, so we address them together.
> > We want to point out the following aspects:
> >
> > **1) Clear Difference in Strutural Positioning and Wording:** In this paper, the paragraph is more narrative, introduced RL in a broad sense and moving to specific examples; Whereas in (Zhou et al. (2024)), the referred two paragraphs are more formal, which details the squential steps of RL process and transitioning the need for MORL.
> >
> > **2) Clear Difference in Level of Details:** Even though the passage in this paper is concise and general, with newer references for example generative AI motivated literature (Franceschelli & Musolesi, 2024); Whereas (Zhou et al. (2024)) goes into more details and (lack of) rigours of MORL, with details of RL.
> >
> > **3) Common Knowledge in the Field:** Many of the terms and ideas are established knowledge in RL. In order for our paper to be self-contained, it is necessary to restate these backgrounds to motivated our problem. We believe that **no** paper possesses exclusive rights to these "general knowledge" in RL, and our works strictly follows the rules of fair use.
> >
> > ________________
> >
> > >**Your comment 5:** The paragraphs about the “Key Contributions” (Lines 97-124) appear to directly follow the “Main Contributions” of the Introduction of (Zhou et al., 2024).
> >
> > **Our Response:** Here, we would like to compare "Key Contributions" in this paper with those in (Zhou et al., 2024) bullet-by-bullet:
> >
> > * In the first bullet of our "Key Contributions,"" we mentioned the impacts of the weight/exploration vector $\mathbf{p}$ on the sample complexity. More importantly, we proposed a completely different WC-MOAC algorithm, which systematically explores Pareto stionarity front assisted by the $\mathbf{p}$-vectors. These are clearly **not in** (Zhou et al.(2024)).
> >
> > * The second bullet in our "Key Contributions" addresses, even with such weight vector $\mathbf{p}$, it achieves sample complexity that is independent of $M$, where $M$ is the number of objectives.
> >
> > * Similarly, third bullet in our "Key Contributions" showed the benefit of momentum approach in the weighted-Chebyshev Pareto stationarity exploration.
> >
> > * Lastly, our empirical results strongly suggest the nature of exploration Pareto stationarity points by looking at Figure 1, which does *not* have any counterpart in (Zhou et al. (2024)). The key takeways from this paper is very different from the that of (Zhou et al. (2024)).

---

> > > ### Author Response · Authors · 2024-11-21
> > > **Response to Reviewer P7fw (Continued)**
> > >
> > > >**Your Comment 6:** In Section 3, the problem formulation about MOMDP (Lines 181-187) appears very similar to the second paragraph of Section 3.1 of (Zhou et al., 2024).
> > >
> > > **Our Response:** MDP is the **standard language** for defining RL problems, and MOMDP is a natural extension of MDP in the multi-objective setting. Similar to our response to your Comments 3--4, **no** paper owns an exclusive right to the MOMDP language for defining MORL problems.
> > > As a result, it shouldn't be a surprise to see similar, if not the entirely same, MOMDP language in all papers that study infinite horizon RL/MORL problems with ergodic property. To see the commonality of these MOMDP formulations, see the beginning of Section 3 of paper (Qiu et al (2021)), Section 2.1 of (Roijers et al (2018)), Section 2.1 of paper (Zhang et al (2018)) and Section 3.1 of (Hairi et al (2022)), just to name a few.
> > >
> > > Qiu, S., Yang, Z., Ye, J., and Wang, Z. On finite-time convergence of actor-critic algorithm. IEEE Journal on Selected Areas in Information Theory, 2(2):652–664, 2021.
> > >
> > > Roijers, D. M., Steckelmacher, D., and Nowe ́, A. Multi-objective reinforcement learning for the expected utility of the return. In Proceedings of the Adaptive and Learn- ing Agents workshop at FAIM, volume 2018, 2018.
> > >
> > > Hairi, F., Liu, J., and Lu, S. Finite-time convergence and sample complexity of multi-agent actor-critic reinforcement learning with average reward. In International Con- ference on Learning Representations, 2022.
> > >
> > > Zhang, K., Yang, Z., Liu, H., Zhang, T., and Basar, T. Fully decentralized multi-agent reinforcement learning with networked agents. In International Conference on Ma- chine Learning, pp. 5872–5881. PMLR, 2018.
> > >
> > > ________________
> > > >**Your Comment 7:** The part on “Learning Goal and Optimality in MORL” (Lines 200-215) appears almost the same as the “Problem Statement” in Section 3.1 of (Zhou et al., 2024). Moreover, even the footnote #3 in this paper almost goes verbatim compared to the footnote #1 in (Zhou et al., 2024).
> > >
> > > **Our Response:** Both our paper and (Zhou et al. (2024)) study infinite-horizon multi-objective reinforcement learning (MORL) problem either under accumulated discounted reward and average reward settings. We see similarities. However, the wordings are significantly disctint. More importantly, the solution approaches and goals in these two works are completely different (Pareto stationarity front exploration vs. identifying only a Pareto stationary point). Also, the footnotes in Section 3.1 in (Zhou et al., 2024) and the footnote #3 in this paper are common language that appear in many papers in the literature. Again, **no** paper owns an exclusive right to these common sentences.
> > >
> > > Moreover, we would like to protest the claim that just because studying the same MORL setting would consitute the so-called "plagiarism or dual submission." In academic research, a long line of works could all be dedicated to stuying a challenging problem setting or even a common problem (e.g., trying to solve or just make progress in proving/disproving a famous conjecture).
> > >
> > > ________________
> > >
> > > >**Your Comment 8:** The preliminaries about the policy gradient for MORAL (Lines 274-296) also largely resembles Section 3.2. Specifically, several sentences about Lemma 2 and Assumption 2 of this paper are exactly the same as those in Lemma 1 and Assumption 2 of (Zhou et al., 2024).
> > >
> > > **Our Response:** First of all, in our paper, we don't have a MORAL concept and we assume you are referring to MORL (multi-objective reinforcement learning). Second, it is simply **not** true that sentences in Lemma 2 of our paper are *"exactly the same"* as those in Lemma 1 of (Zhou et al. (2024)) and Assumption 2 with Assumption 2 in (Zhou et al. (2024)) for that matter.
> > >
> > > More importantly, Lemma 2 and Assumption 2 (correspondingly Lemma 1 and Assumption 2 in (Zhou et al., 2024)) are standard and basic concepts in the context of actor-critic framework and the policy gradient theorem for RL, which can be found in many classic papers in RL (e.g., (Sutton et al. (1999)). Moreover, we have also properly cited the sources in the subsequent remark after Lemma 2. Also, Assumption 2 is also a collection of standard assumptions in critic analysis with linear approximation in the literature, for which we have provided references for it. We never claimed that Lemma 1 and Assumption 2 are our own contributions.
> > >
> > > Richard S Sutton, David A McAllester, Satinder P Singh, Yishay Mansour, et al. Policy gradient methods for reinforcement learning with function approximation. In NIPs, volume 99, pp. 1057– 1063. Citeseer, 1999.
> > >
> > > ________________

---

> > > > ### Author Response · Authors · 2024-11-21
> > > > **Response to Reviewer P7fw (Continued)**
> > > >
> > > > >**Your Comment 9:** The description about Assumption 3 and Lemma 3 in this paper (Lines 424-437) appears to exactly follow the Assumption 3 and Lemma of (Zhou et al., 2024).
> > > >
> > > > **Our Reponse:** The Assumption 3 and Lemma 3 are standard assumptions widely used in RL and optimization in general. For example, Assumption 3 is listed as Assumption 1 and Proposition 1 in (Xu et al. (2020)) and references within, Assumption 5 of (Qiu et al. (2021)). Lemma 3 is Assumption 2 in (Xu et al (2020)) and Assumption 1 in (Qiu et al. (2021)). In fact, Assumption 3 (Lipschitz smoothness of the value function) is one of the most basic assumptions that establish the convergece of most policy gradient approaches.
> > > >
> > > > Qiu, S., Yang, Z., Ye, J., and Wang, Z. On finite-time convergence of actor-critic algorithm. IEEE Journal on Selected Areas in Information Theory, 2(2):652–664, 2021.
> > > >
> > > > Tengyu Xu, Zhe Wang, and Yingbin Liang. Improving sample complexity bounds for (natural) actor-critic algorithms. arXiv preprint arXiv:2004.12956, 2020.
> > > >
> > > > ________________
> > > > >**Your Comment 10 (Also Question 1):** One of my main technical concerns is the motivation for finding a Pareto-stationary policy (under the assumption that the state and action spaces are finite) in the specific context of MORL. Specifically, while it is indeed difficult to find the whole Pareto front in MORL, it is actually not hard to find one or some Pareto-optimal policies by adapting reducing MORL to single-objective RL and finding the convex coverage set (e.g., (Yang et al., 2019)). For example, based on (Chen and Magulari, 2022), one can use a policy-based method with off-policy TD learning (under linear function approximation) to find an epsilon-optimal solution for single-objective RL with a sample complexity of $\mathcal{O}(\epsilon^{-2})$.There are also several other recent works like (Lan 2021; Fatkhullin et al., 2023; Liu et al., 2020; Chen et al., 2022) that can find an epsilon-optimal policy with sample complexity guarantees. To adapt these results to MORL, one can use linear scalarization and thereby find one Pareto-optimal policy (specific to some preference vector). As a result, it remains not totally clear why it is theoretically appealing to design an algorithm for finding only a Pareto-stationary policy if we can already find Pareto-optimal policies (despite that Pareto-stationarity is indeed a widely adopted concept in the MOO literature).
> > > >
> > > > **Our Response:** It is indeed true that finding a Pareto optimal point is not hard and can be achieved via the linear scalarization approach. However, as stated in (Yang et al. (2019)), the CCS is only a subset of the Pareto frontier, containing the solutions on its outer convex boundary. Similarly, by Proposition 4.2 in (Qiu et al. (2024)), the linear scalarization (LS) approach cannot guarantee exploring the full Pareto front. When Pareto front is non-convex, it only provides limited exploration (more precisely, the convex hull of the Pareto front).
> > > >
> > > > Morever, from Proposition 4.7 and its proof on Page 42 in (Qiu et al. (2024)), the weighted-Chebyshev approach can guarantee exploring all Pareto-optimal solutions. This observation motivates us to design a weighted-Chebyshev-based Pareto front exploration approach for MOAC (i.e., WC-MOAC). Although our theoretical results only guarantees convergence to Pareto stationary points due to the fundamental intractability of the non-convex MORL setting, our empirical comparison results on the footprints of exploration for both LS in Figure 1(a) and weighted-Chebyshev approach in Figure 1(b) show that WC-based exploration indeed achieves a larger collection of Pareto-stationary solutions than that of the LS approach in Figure \(c).
> > > >
> > > > Shuang Qiu, Dake Zhang, Rui Yang, Boxiang Lyu, and Tong Zhang. Traversing pareto optimal policies: Provably efficient multi-objective reinforcement learning, 2024. URL https://arxiv.org/abs/2407.17466.
> > > >
> > > > ________________

---

> > > > > ### Author Response · Authors · 2024-11-21
> > > > > **Response to Reviewer P7fw (Continued)**
> > > > >
> > > > > >**Your Comment 11 (Also Question 2):** Another concern is the novelty in terms of algorithm and convergence analysis. Specifically, the WC-MOAC algorithm appears to be a direct application of the MOAC algorithm (Zhou et al. 2024) and also similar to the MOO algorithm CR-MOGM of (Zhou et al. 2022), which is the enhanced (stochastic) MGDA method (e.g., (Desideri, 2012; Liu and Vicente, 2021)) with the momentum update of the dual variable vector, to the setting of MORL (with the multi-objective critic learned by standard TD updates). Under a properly learned critic, then the stochastic multiple gradients can have a sufficiently low bias such that it enables similar convergence guarantees as in the general MOO. This is also shown in Theorem 11 (as a direct result of Lemma 10). As a result, the sample complexity and the convergence analysis of WC-MOAC essentially resemble those of CR-MOGM in (Zhou et al. 2022) for the general non-convex case, cf. Theorem 3 and Appendix E of (Zhou et al. 2022).
> > > > >
> > > > > **Our Response:** Due to the length of this comment, we organize our response in four parts:
> > > > >
> > > > > 1) **Direction Application of (Zhou et al. 2024):** It's simply **not** true to say that our work is a direct application of (Zhou et al. (2024)). On the contrary, our work is far more general than (Zhou et al. (2024)) because the goal in (Zhou et al. (2024)) is only limited to finding a mere Pareto Stationary point; while our work aims to systematically explore the Pareto stationarity front. From this perspective, our paper can be considered a general approach that includes the results in (Zhou et al. (2024)) as a special case. To see this, note that, when $u=0$ and $\mathbf{p}=(1,\cdots,1)^{\top}$ (for simplicity, ignoring the scaling) in our work, the results actually imply the those in (Zhou et al. (2024)). This is because Eq.(10) reduces to the $\min_{\lambda}||\mathbf{K_p}\lambda||^{2}$, which is equivalent to MGDA where the Quadratic programming is to solve $\min_{\lambda}||\mathbf{G_t}\lambda||^{2}$ where $\mathbf{G_t}=[\mathbf{g}^{i}_t,\cdots, \mathbf{g}^{M}_t]$ and $\mathbf{g}^{i}_t$ is defined in the second column of Line 394. In Algorithm 1, the update of $\mathbf{g}_t$ will reduce to $\mathbf{g}_t= \mathbf{G}_t\lambda_t$. In other words, the WC-approach proposed in this paper is a far more general approach than that of (Zhou et al. (2024)). Please also see our response to your Comment 2 for further details.
> > > > >
> > > > > 2) **Novelty in Algorithm and Analysis:** The novelty in the convergence analysis stems from analysis of involving Hadamard product by formulating the problem in a weighted-Chebyshev problem. In contrast, there is **no** such operand in (Zhou et al (2024)). In particular, unlike the pure MGDA approach where the $\mathbf{\lambda_t}$ is used to weigh multiple gradients of the objectives, due to the new WC component in our work, we need to construct a pseudo weight $q_t:=\frac{\lambda_t\odot \mathbf{p}}{\langle \lambda_t, \mathbf{p}\rangle}$ due to aforementioned Hadamard product and utilize properties of such pseudo weight. Please see our proofs in Lines 907-970 and Lines 1160-1210, where we carefully handled such pseudo weight, as comparison with naive MGDA. As a result of this analysis, we observed that $p_{\min}$ is a cruicial quantity in characterizing the stationarity convergence. As one can easily observe that coefficients in front of the cruicial terms are significantly different from (Zhou et al. (2024)). Just to name one example, the coefficient in front of $\mathcal{O}(1/T)$ term is $16L_J r_{\max}(1+\frac{2}{p_{\min}^{2}}\sum_{t=1}^{T}\eta_t)$ whereas $18L_Jr_{\max}(1+2\sum_{t=1}^{T}\eta_t)$. To claim that they are *"almost the same"*, it is unfair to say the least.
> > > > >
> > > > > 3) **Standard TD Updates:** We agree that Theorem 11 considers the standard TD learning. However, it has nothing to do with stochastic multiple gradient apporach as suggested by the reviewer. Specifically, in our work, TD learning is not designed to optimize the Mean Square Bellmann Error, where it may be possible to formulate the policy evaluation into such multiple gradient approach. However, this problem is beyond the scope of this paper, which is more appropriate to investigate in a separate paper. In this work, MGDA is incorporated into the actor component in our algorithmic design as we described in details at Step 2 of Section 4.2.

---

> > > > > > ### Author Response · Authors · 2024-11-21
> > > > > > **Response to Reviewer P7fw (Continued)**
> > > > > >
> > > > > > (continued response to Comment 11):
> > > > > >
> > > > > > 4) **Theorem 3 in (Zhou et al. (2022)):** Theorem 3 in (Zhou et al. (2022)) actually highlights the **new** contribution of our analysis since we don't have the $\sum_{i=1}^{m}|\hat{\lambda_k^{i}}-\lambda_{k-1}^{i}| $ coefficient terms in front of $O(\frac{1}{n})$, where $n$ is the same as $T$ in this work. The difference is that their convergence metric is $O(m)$ in the worst case because of the $\sum_{i=1}^{m}|\hat{\lambda_k^{i}}-\lambda_{k-1}^{i}|$ coefficient terms, where $m$ is the number of objectives in (Zhou et al.(2022)). Similarly, the subsequent terms in Theorem 3 of (Zhou et al. (2022)) are not only at least $O(m)$-dependent, but also depend on the accumulative learning rates $\sum_{k}\eta_k$ and momentum rates $\sum_{k}(1-\alpha_k)$. These key differences highlights novelty in our theoretical analaysis on pseudo weight $q_t:=\frac{\lambda_t\odot \mathbf{p}}{\langle \lambda_t, \mathbf{p}\rangle}$ term.
> > > > > >
> > > > > > ________________
> > > > > >
> > > > > > We hope our responses have addressed the reviewer's concerns on ethics and technical novelty, and we are happy to answer any further questions. If the concerns are addressed, we would highly appreciate a re-evaluation of our work.

---

> ### Comment · Reviewer_P7fw · 2024-12-03
>
> I sincerely thank the authors for the detailed response and for answering my technical questions. Below let us focus more on the Comments 1-8.
>
> - **Comment 1 (regarding the pseudo code of WC-MOAC and MOAC), Comment 2 (regarding the theoretical results), and Comments 3-4 (regarding the introduction)**: I can understand that both algorithms share the actor-critic architecture, and the two papers are both focused on the convergence to the stationarity. However, from a reviewer’s viewpoint, given the high similarity (in terms of design, paper structure, and even wording), it appears almost impossible to argue that this paper is completed independently from [Zhou et al., 2024] or the two are just concurrent works.  Moreover, as far as I understand, such similarity in both structure and wording shall be viewed as paraphrasing, which is not allowed if there is no due credit or acknowledgement given to the prior works (i.e., [Zhou et al., 2024]). Furthermore, as mentioned by Reviewer JPv7, without a proper comparison with or mention of [Zhou et al., 2024], the readers can be easily misguided by the overclaimed contributions.
>
> - **Comment 5 (regarding the Key Contributions)**: Just like the comment above, my concern about paraphrasing in this part still remains unsolved. Moreover, apart from the structure and wording, the novelty and contributions can be overestimated without properly mentioning [Zhou et al., 2024]. Specifically:
>
> > In the first bullet of our "Key Contributions,"" we mentioned the impacts of the weight/exploration vector on the sample complexity. More importantly, we proposed a completely different WC-MOAC algorithm, which systematically explores Pareto stionarity front assisted by the $\mathbf{p}$-vectors. These are clearly not in (Zhou et al.(2024)).
>
> Based on the current writing of the paper, the readers can get easily misled that WC-MOAC is a totally new algorithm in the MORL literature. However, under a comparison with MOAC, WC-MOAC can indeed appear somewhat incremental.
>
> > The second bullet in our "Key Contributions" addresses, even with such weight vector $\mathbf{p}$, it achieves sample complexity that is independent of $M$, where $M$ is the number of objectives.
>
> This is also one key feature that has been shown by MOAC. Then, this would not be a convincingly new feature for people to choose WC-MOAC over MOAC.
>
> > Similarly, the third bullet in our "Key Contributions" showed the benefit of the momentum approach in the weighted-Chebyshev Pareto stationarity exploration.
>
> As far as I know, the momentum-based design has already been proposed by MOAC (cf. Equation (10) in [Zhou et al., 2024]), and its benefit has been shown by the MOAC paper.
>
> > Lastly, our empirical results strongly suggest the nature of exploration Pareto stationarity points by looking at Figure 1, which does not have any counterpart in (Zhou et al. (2024)). The key takeaways from this paper is very different from that of (Zhou et al. (2024)).
>
> I can understand that the empirical results are shown to corroborate the nature of exploration of Pareto stationary points. That being said, if one directly compares the results of MC-MOAC and those of the MOAC in [Zhou et al., 2024] in terms of Like, Comment, Dislike, and WatchTime, the differences appear rather not significant. It is not immediately clear to me why the empirical results (cf. Table 1 and Figure 1) clearly supports the claim. While I understand in general that the works published within 3 months of submission are not necessarily needed in the comparison, a comparison with MOAC can serve as a really helpful ablation study and hence nicely support the claim of the paper. Please correct me if I missed anything.
>
>
> - **Comments 6-8 (regarding the writing of the problem formulation and preliminaries)**: I understand that MDP and MOMDP are standard problem formulations in RL. However, there are various ways to describe the same setting. As far as I know, it is a standard practice to describe the problem formulation that tailors to the need of each specific paper and hence shall be written in the authors’ own words. Again, as mentioned above, this similarity in both structure and wording shall be viewed as paraphrasing and is not allowed if there is no due credit or acknowledgement given to the prior works. Therefore, I would like to emphasize that “using the similar formulations” is not the issue, and the high similarity in terms of structure and wording is where the concerns arise. I hope that the authors can understand my concerns.

---

> > ### Author Response · Authors · 2024-12-03
> > **Response to Reviewer P7Fw's Follow-up Comments:**
> >
> > Thank you for your comments. Based on the comments, our reponses are as follows.
> > >**Your Follow-up Comment 1:** I sincerely thank the authors for the detailed response and for answering my technical questions. Below let us focus more on the Comments 1-8.
> >
> >
> > **Our Response:** We are glad to see that you have accepted our technical contributions and technical differences compared to (Zhou et al.(2024)) in our previous responses to your Comments 9-11.
> >
> > _____________
> > >**Your Follow-up Comment 2:** Comment 1 (regarding the pseudo code of WC-MOAC and MOAC), Comment 2 (regarding the theoretical results), and Comments 3-4 (regarding the introduction): I can understand that both algorithms share the actor-critic architecture, and the two papers are both focused on the convergence to the stationarity. However, from a reviewer’s viewpoint, given the high similarity (in terms of design, paper structure, and even wording), it appears almost impossible to argue that this paper is completed independently from [Zhou et al., 2024] or the two are just concurrent works. Moreover, as far as I understand, such similarity in both structure and wording shall be viewed as paraphrasing, which is not allowed if there is no due credit or acknowledgement given to the prior works (i.e., [Zhou et al., 2024]). Furthermore, as mentioned by Reviewer JPv7, without a proper comparison with or mention of [Zhou et al., 2024], the readers can be easily misguided by the overclaimed contributions.
> >
> > **Our Reponse:** In Section 2 on related work, we have specifically compared and contrasted our work to  (Zhou et al. (2024)). Specifically, we have stated that (Zhou et al. (2024)) falls into the no-preference category (See Line 139) and is not designed for Pareto stationarity front exploration. But to address the reviewer's concern, we will provide a much more thorough comparison to (Zhou et al.(2024)) and give more explicit credit to (Zhou et al. (2024)) in our revisions.
> >
> > ____________
> > >**Your Follow-up Comment 3:** Comment 5 (regarding the Key Contributions): Just like the comment above, my concern about paraphrasing in this part still remains unsolved. Moreover, apart from the structure and wording, the novelty and contributions can be overestimated without properly mentioning [Zhou et al., 2024]. Specifically:
> > Based on the current writing of the paper, the readers can get easily misled that WC-MOAC is a totally new algorithm in the MORL literature. However, under a comparison with MOAC, WC-MOAC can indeed appear somewhat incremental.
> >
> > **Our Response:** We understand that it is the reviewer's right to claim our work is "incremental". However, even if our work is indeed "incremental", which we respectfully disagree, it is fundamentally different from "plagiarism" and "dual submission". These harsh accusations are harmful to all authors' academic reputatation and even career-threating. Therefore, we sincerely hope the reviewer can remove the flag even if the reviewer still leans toward a rejection.
> > _____________
> > >**Your Follow-up Comment 4:**
> > >> The second bullet in our "Key Contributions" addresses, even with such weight vector $\mathbf{p}$, it achieves sample complexity that is independent of $M$, where $M$ is the number of objectives.
> > >
> > > This is also one key feature that has been shown by MOAC. Then, this would not be a convincingly new feature for people to choose WC-MOAC over MOAC.
> >
> > **Our Response:** In our revisions, we are happy to tone down our wording in the bullets of key contributions to important remark and cite (Zhou et al.(2024)). However, we do want emphasize that there does **not** exist such an issue of *"choosing WC-MOAC over MOAC or vice versa"*. This is because this paper studies the systematic exploration of the Pareto Stationay front, while (Zhou et al.(2024)) is only designed to guarantee the convergence to a Pareto stationary point. In other words, these two works aim at two completely different goals and not competing with each other. Therefore, it is important for WC-MOAC to establish/maintain an $M$-independence property in the **new problem** of systematically exploring Pareto stationarity front, which is **not** the goal of (Zhou et al. (2024)).
> > ________________

---

> > > ### Author Response · Authors · 2024-12-03
> > > **Response to Reviewer P7Fw's Follow-up Comments (Continued):**
> > >
> > > >**Your Follow-up Comment 6:**
> > > >> Lastly, our empirical results strongly suggest the nature of exploration Pareto stationarity points by looking at Figure 1, which does not have any counterpart in (Zhou et al. (2024)). The key takeaways from this paper is very different from that of (Zhou et al. (2024)).
> > > >
> > > > I can understand that the empirical results are shown to corroborate the nature of exploration of Pareto stationary points. That being said, if one directly compares the results of MC-MOAC and those of the MOAC in [Zhou et al., 2024] in terms of Like, Comment, Dislike, and WatchTime, the differences appear rather not significant. It is not immediately clear to me why the empirical results (cf. Table 1 and Figure 1) clearly supports the claim. While I understand in general that the works published within 3 months of submission are not necessarily needed in the comparison, a comparison with MOAC can serve as a really helpful ablation study and hence nicely support the claim of the paper. Please correct me if I missed anything.
> > >
> > > **Our Response:** Thank you for your question. In the following table, we provide a direct comparison of values of WC-MOAC in Our Table 1 and that of MOAC in (Zhou et al.(2024)) over the aforementioned 4 objectives. The 3rd-5th row of values are based on improvements over Behavior-Clone(BC) in Table 1 of our paper. We also remark that for dislike objective, the more decrease indicates the better performance.
> > > | Objective | Like($10^{-2}$) | Comment($10^{-3}$) | Dislike($10^{-4}$) | WatchTime |
> > > | -------- | ------- |------- |------- |------- |
> > > | MOAC | 1.312 | 3.266 | 1.486 | 1.307 |
> > > | WC-MOAC | 1.329 | 3.092 | 1.339 | 1.375 |
> > > | MOAC over BC | 6.57% | 1.27% | -35.5% | 1.71% |
> > > | WC-MOAC over BC | 7.96% | -4.12% | -41.88% | 7.00% |
> > > | WC-MOAC over MOAC | 1.39% | -5.39% | 6.38% | 5.29% |
> > >
> > > We can see that among the four objectives "Like," "Comment," "Dislike," and
> > > "WatchTime," 3 of them ("Like," "Dislike," and "WatchTime") favors WC-MOAC over MOAC and only "Comment" favors MOAC over WC-MOAC. More specifically, WC-MOAC improved over MOAC by 5.29% in watchtime, 6.38% in dislike and 1.39% in likes. WC-MOAC is worse than MOAC by 5.39% in the "Comment" objective. Overall, it shows WC-MOAC with the particular choice of $\mathbf{p}$ does find a Pareto solution that possesses 3:1 favorable objectives than MOAC.
> > >
> > > We would also like to point out that, sometimes, 1% increase doesn’t necessarily mean "not significant". On the contrary, it often has a significant implication in many real-world systems. For example, according to [1], "*the revenue from Kuaishou streaming business increased by 10.4% to RMB39.1 billion in 2023, from RMB35.4 billion in 2022, benefiting from consistent enrichment of content supply and continuous optimization of our live streaming ecosystem and algorithms*" [1]". With a 5.29% improvement on watchtime objective, it is a potentially significant improvement considering the population of user base and total revenue.
> > >
> > > [1] https://ir.kuaishou.com/news-releases/news-release-details/kuaishou-technology-announces-fourth-quarter-and-full-year-2023/
> > >
> > > Coming back to the main point of our paper, with the help of weight-vector $\mathbf{p}$, by exploring Pareto Stationary front, WC-MOAC enables finding a more preferable solution than MOAC, which merely guarantees finding a Pareto stationary point.
> > > _________________

---

> > > > ### Author Response · Authors · 2024-12-03
> > > > **Response to Reviewer P7Fw's Follow-up Comments (Continued):**
> > > >
> > > > > **Your Follow-up Comment 7:** Comments 6-8 (regarding the writing of the problem formulation and preliminaries): I understand that MDP and MOMDP are standard problem formulations in RL. However, there are various ways to describe the same setting. As far as I know, it is a standard practice to describe the problem formulation that tailors to the need of each specific paper and hence shall be written in the authors’ own words. Again, as mentioned above, this similarity in both structure and wording shall be viewed as paraphrasing and is not allowed if there is no due credit or acknowledgement given to the prior works. Therefore, I would like to emphasize that “using the similar formulations” is not the issue, and the high similarity in terms of structure and wording is where the concerns arise. I hope that the authors can understand my concerns.
> > > >
> > > > **Our Response:** In terms of formulations, we have already genuinely described them using our own words. In our humble opinion, this is strictly following academic fair use. Moreover, we want to mention such standard phrasing always brings similarity, as the intention is to precisely describe the same mathematical models that have been widely accepted in the literature. Additionally, we have provided many literature referencs that uses the same assumptions and similar preliminaries. In addition, MDP and MOMDP are standard concepts, not new contributions of any related work in this field. One may find such simialirites in Section 3 of paper (Qiu et al (2021)), Section 2.1 of (Roijers et al (2018)), Section 2.1 of paper (Zhang et al (2018)) and Section 3.1 of (Hairi et al (2022)). Also, one may find similar preliminaries in (Qiu et al (2021)) and (Xu et al.(2020)).
> > > >
> > > > Qiu, S., Yang, Z., Ye, J., and Wang, Z. On finite-time convergence of actor-critic algorithm. IEEE Journal on Selected Areas in Information Theory, 2(2):652–664, 2021.
> > > >
> > > > Roijers, D. M., Steckelmacher, D., and Nowe ́, A. Multi-objective reinforcement learning for the expected utility of the return. In Proceedings of the Adaptive and Learn- ing Agents workshop at FAIM, volume 2018, 2018.
> > > >
> > > > Hairi, F., Liu, J., and Lu, S. Finite-time convergence and sample complexity of multi-agent actor-critic reinforcement learning with average reward. In International Con- ference on Learning Representations, 2022.
> > > >
> > > > Zhang, K., Yang, Z., Liu, H., Zhang, T., and Basar, T. Fully decentralized multi-agent reinforcement learning with networked agents. In International Conference on Ma- chine Learning, pp. 5872–5881. PMLR, 2018.
> > > >
> > > > Tengyu Xu, Zhe Wang, and Yingbin Liang. Improving sample complexity bounds for (natural) actor-critic algorithms. arXiv preprint arXiv:2004.12956, 2020
> > > >
> > > > For example, in Section 2.1 of (Zhang et al.(2018)), the definition is *"A Markov decision processes is characterized by a quadruple $M = \langle S , A, P , r \rangle$, where S is a finite state space, A is a finite action space, $P (s′ | s, a) : \mathcal{S} × {A} × {S} → [0, 1]$ is a state transition probability" from state s to s′ determined by action a, and $R(s, a) : S × A → R$ is a reward function defined by $R(s, a) = \mathbb{E}[r_{t+1} | s_t = s, a_t = a]$, with $r_{t+1}$ being the instantaneous reward at time t. Policy of the agent is a mapping $\pi : \mathcal{S} × \mathcal{A} \rightarrow [0, 1]$, representing the probability of choosing action a at state s. The objective of the agent is to find the optimal policy that maximizes the expected time-average reward, notably, long-term return, which is given by $J(\pi)$:
> > > > $$J(π) = \lim_{T} \frac{1}{T}\sum_{t=0}^{T-1}\mathbb{E}(r_{t+1}) = \sum_{s\in\mathcal{S}}d_{\pi}(s)\sum_{a\in\mathcal{A}}\pi(s,a)R(s,a), $$
> > > > where $d_{\pi}(s) = \lim_{t} \mathbb{P}(s_t = s|\pi)$ is the stationary distribution of the Markov chain under policy $\pi$."*

---

> > > > > ### Author Response · Authors · 2024-12-03
> > > > > **Response to Reviewer P7Fw's Follow-up Comments (Continued):**
> > > > >
> > > > > (Continuing response to follow-up comment 7)
> > > > >
> > > > > Similarly in Section 3 of (Qiu et al.(2021)), it states *"The infinite-horizon average reward reinforcement learning problem [24], [25] is modeled as an average reward Markov Decision Process (MDP). Suppose that $\mathcal{S}$ and $\mathcal{A}$ are the finite state space and finite action space respectively. The policy $\pi$ is defined as a function that $\pi : \mathcal{A}×\mathcal{S} \rightarrow [0,1]$ such that $\pi(a|s)$ is the probability of choosing action $a \in \mathcal{A}$ at state $s \in \mathcal{S}$. From a practical perspective, the policy $\pi$ is usually parameterized by $\theta\in\Theta$ in a nonconvex form and then we denote the parameterized policy as $\pi_{\theta}$ . An agent takes $a ∼ \pi_{\theta}(· | s)$ at state s. Letting $P(s'| a, s)$ be the probability of an agent moving from state $s$ to state $s'$ with an action $a$, we can have a Markov transition probability induced by $\pi_{\theta}$ as$P^{\pi_{\theta}} (s′ | s) = \sum_{a\in\mathcal{A}} P(s′ | a, s)\pi_{\theta}(a | s)$, which is the probability of moving from state $s$ to state $s′$. At each time $τ$, we use a tuple $(s_τ,a_τ,s_{τ+1},r_{τ+1})$ to denote that an agent at state $s_τ$ chooses an action $a_τ$ and transitions to the next state $s_{τ+1}$ with a reward $r_{τ+1} :=r(s_τ,a_τ,s_{τ+1})$,where $r:S×A×S\rightarrow\mathbb{R}$ is a map to reward values. We assume $|r| ≤ r_{\max}$. Next, we make the following assumption on the policy $π_{\theta}$ and the probability $P(s′ | a, s)$.
> > > > > Assumption 1: We assume that the parameterization of $\pi_{\theta}$ and $P(s′|a,s)$ guarantee that the Markov chain decided by $P^{\pi_{\theta}} (s′ | s)$ for any $\theta\in\Theta$ is irreducible and aperiodic. Then, there can always exist a unique stationary distribution for any $\theta\in\Theta$, which is denoted as $d_{\pi_{\theta}}(s)$ for any state $s\in\mathcal{S}$."*
> > > > >
> > > > > It is clear from the above two examples of MDP definitions that there are significant wording overlaps in related work in the literature. Nonetheless, we are happy to adopt the suggestion of citing (Zhou et al.(2024)) in the formulations to indicate the similarity in problem setting.

---

### Meta-Review · Area_Chair_QjUf · 2024-12-21

**Metareview:**

This paper proposes the Weighted-Chebyshev Multi-Objective Actor-Critic (WC-MOAC) algorithm for Multi-Objective Reinforcement Learning (MORL). The authors aim to systematically explore the Pareto Stationary Front using an integration of Weighted-Chebyshev scalarization and momentum-based updates within the actor-critic framework. There are significant concerns regarding its originality and overlap with prior work, particularly with the paper by Zhou et al. (2024). These concerns include structural, algorithmic, and even textual similarities, which undermine the contribution of this work as a standalone advance.

This meta-review was reviewed by the senior area chair and the SAC confirms the text and decision.

**Additional Comments On Reviewer Discussion:**

During the rebuttal period, reviewers raised significant concerns regarding the overlap of the submission with Zhou et al. (2024), particularly in algorithmic design, theoretical guarantees, and textual content. Reviewers also emphasized the lack of novelty in the weighted-Chebyshev formulation, arguing it was a minor extension of existing methods. I read the paper and agree with Reviewers P7fw and JPv7 that this paper has concerning similarities to Zhou et al. (2024), even with text overlap flagging ethical concerns.

---

### Decision · Program_Chairs · 2025-01-22

Reject